# A transcriptomic and epigenomic cell atlas of the mouse primary motor cortex

Zizhen Yao[1,32], Hanqing Liu[2,32], Fangming Xie[3,32], Stephan Fischer[4,32], Ricky S. Adkins[5], Andrew I. Aldridge[2], Seth A. Ament[5], Anna Bartlett[2], M. Margarita Behrens[6], Koen Van den Berge[7,8], Darren Bertagnolli[1], Hector Roux de Bézieux[9], Tommaso Biancalani[10], A. Sina Booeshaghi[11], Héctor Corrada Bravo[12], Tamara Casper[1], Carlo Colantuoni[13,14,15], Jonathan Crabtree[5], Heather Creasy[5], Kirsten Crichton[1], Megan Crow[4], Nick Dee[1], Elizabeth L. Dougherty[10], Wayne I. Doyle[16], Sandrine Dudoit[7], Rongxin Fang[17], Victor Felix[5], Olivia Fong[1], Michelle Giglio[5], Jeff Goldy[1], Mike Hawrylycz[1], Brian R. Herb[5], Ronna Hertzano[5,18], Xiaomeng Hou[19], Qiwen Hu[20], Jayaram Kancherla[12], Matthew Kroll[1], Kanan Lathia[1], Yang Eric Li[21], Jacinta D. Lucero[6], Chongyuan Luo[2,22,23], Anup Mahurkar[5], Delissa McMillen[1], Naeem M. Nadaf[10], Joseph R. Nery[2], Thuc Nghi Nguyen[1], Sheng-Yong Niu[2], Vasilis Ntranos[24], Joshua Orvis[5], Julia K. Osteen[6], Thanh Pham[1], Antonio Pinto-Duarte[6], Olivier Poirion[19], Sebastian Preissl[19], Elizabeth Purdom[7], Christine Rimorin[1], Davide Risso[25], Angeline C. Rivkin[23], Kimberly Smith[1], Kelly Street[26], Josef Sulc[1], Valentine Svensson[11], Michael Tieu[1], Amy Torkelson[1], Herman Tung[1], Eeshit Dhaval Vaishnav[10], Charles R. Vanderburg[10], Cindy van Velthoven[1], Xinxin Wang[19,31], Owen R. White[5], Z. Josh Huang[27], Peter V. Kharchenko[20], Lior Pachter[11], John Ngai[28], Aviv Regev[10,29], Bosiljka Tasic[1], Joshua D. Welch[30], Jesse Gillis[4], Evan Z. Macosko[10], Bing Ren[19,21], Joseph R. Ecker[2,23], Hongkui Zeng[1 ✉] & Eran A. Mukamel[16 ✉]

Single-cell transcriptomics can provide quantitative molecular signatures for large, unbiased samples of the diverse cell types in the brain[1–3]. With the proliferation of multi-omics datasets, a major challenge is to validate and integrate results into a biological understanding of cell-type organization. Here we generated transcriptomes and epigenomes from more than 500,000 individual cells in the mouse primary motor cortex, a structure that has an evolutionarily conserved role in locomotion. We developed computational and statistical methods to integrate multimodal data and quantitatively validate cell-type reproducibility. The resulting reference atlas— containing over 56 neuronal cell types that are highly replicable across analysis methods, sequencing technologies and modalities—is a comprehensive molecular and genomic account of the diverse neuronal and non-neuronal cell types in the mouse primary motor cortex. The atlas includes a population of excitatory neurons that resemble pyramidal cells in layer 4 in other cortical regions[4]. We further discovered thousands of concordant marker genes and gene regulatory elements for these cell types. Our results highlight the complex molecular regulation of cell types in the brain and will directly enable the design of reagents to target specific cell types in the mouse primary motor cortex for functional analysis.

The cellular components of brain circuits are extraordinarily diverse[5,6]. Single-cell molecular assays, especially transcriptomic measurements by RNA sequencing (RNA-seq), have accelerated the discovery of cell types across brain regions and in diverse species[7]. Recent advances include single-cell transcriptomic datasets with more than $10^5$ individual cells, identifying hundreds of neuronal and non-neuronal cell types across the mouse nervous system[1–3]. As the number of profiled cells grows into the millions, a key question is whether these data will converge towards a comprehensive, coherent taxonomy. Although a comprehensive cell atlas should incorporate anatomical and physiological information, the high throughput of single-cell sequencing assays presents an opportunity for establishing a broad-based transcriptomic and epigenomic cell atlas. Molecular and genomic cell signatures will drive progress across modalities and help to obtain functional information.

Within the BRAIN Initiative Cell Census Network (BICCN), we aim to create an atlas of cell types across the brain of several mammalian species by integrating multiple single-cell omics approaches. We selected the primary motor cortex (MOp) (Extended Data Fig. 1a–d) as the starting point for our joint efforts owing to its relatively conserved structure and function across mammalian species. The MOp

lacks species-specific cellular structures, such as the whisker barrels in the rodent primary somatosensory cortex and the elaborate layer 4 (L4) with multiple sublayers in the primate primary visual cortex. Traditionally, the MOp is considered to lack a cytoarchitechtonically defined granular layer (L4), although neurons in the MOp with L4-like connectivity have been identified[4]. Our mouse MOp atlas is a case study of the expansive potential and the technical limitations of single-cell molecular methods for comprehensive brain-wide analysis of cell types.

Single-cell transcriptomics identifies cell-type marker genes and gene modules that shape functions such as the mode of synaptic communication[8]. Epigenomic measurements of DNA methylation and open chromatin provide signatures of gene regulation, including non-coding regulatory regions such as enhancers. Neurons acquire unique patterns of CG and non-CG DNA methylation during postnatal development[9,10] and have cell-type-specific open chromatin[11]. Together, transcription and epigenetic modifications establish attractors in a cell-state space that corresponds to cell types[12,13]. Here we integrated large-scale single-cell transcriptomic and epigenomic datasets to achieve a reference taxonomy for the adult mouse MOp.

## Multimodal molecular census of mouse MOp

We produced nine datasets, including seven single-cell or single-nucleus transcriptomic dataset (single-cell RNA-seq (scRNA-seq) and single-nucleus RNA-seq (snRNA-seq) using 10x v2, v3 and SMART-Seq v4; $n = 526,373$ high-quality cells), one single-nucleus DNA methylation dataset (snmC-seq2; $n = 9,872$) and one single-nucleus open chromatin dataset (single-nucleus assay for transposase-accessible chromatin using sequencing (snATAC-seq); $n = 81,196$) (Extended Data Fig. 1e, f, Supplementary Table 1). These span a range of technologies, assaying different numbers of cells, with different depths of sequence coverage per cell, and assessing different biological features (Fig. 1a). The datasets reflect the trade-off between the number of sequenced molecules per cell, which depends on cell size and the efficiency of RNA or DNA capture, and the total number of cells that can be assayed for a fixed total cost. Our datasets include single-nucleus transcriptomes from over 175,000 cells (using the 10x Chromium 3′ v3 platform), which captures a median of 3,100–12,700 unique molecular identifiers (UMIs) per cell. By contrast, full-length transcript sequencing using SMART-Seq v4 captured a greater number of unique molecular fragments per cell (1 million–2.1 million), but covered fewer cells (approximately 6,300 cells per dataset). Data on single-nucleus DNA methylation provided deep coverage of the epigenome per cell (median of 1.66 million unique sequenced DNA fragments, covering 6.2% of the genome) for a modest number of cells[9,14] (approximately 9,800 cells). Finally, snATAC-seq data scaled to over 81,000 cells but sampled fewer DNA fragments for individual cells (median of 3,778 unique fragments per cell; Supplementary Table 1)[11].

Subsampling RNA-seq datasets (Extended Data Fig. 2b, Supplementary Table 1) showed that scRNA-seq generally detects more genes per cell (up to approximately 7,100 median genes per cell for 10x and 10,000 for SMART) than snRNA-seq (up to approximately 4,000 for 10x and 5,800 for SMART). The 10x v3 platform detected 60–100% more genes than 10x v2. The number of genes detected per cell in the snRNA-seq 10x v3 B dataset (median of approximately 4,000 genes), using an improved nucleus isolation protocol[15] (Methods), was substantially higher than the other snRNA-seq datasets (1,700–3,500 genes) and was similar to the scRNA-seq 10x v3 dataset when compared at the same sequencing depth.

We created web resources to interactively access, explore, visualize and analyse the raw and processed datasets (Extended Data Fig. 1g, h).

## A consensus transcriptomic atlas of MOp

To establish a transcriptomic reference atlas of the mouse MOp, we jointly analysed seven scRNA-seq and snRNA-seq datasets. The datasets were mutually consistent, with strongly correlated expression of

cell-type marker genes (Extended Data Fig. 2a, d, e) despite different sensitivity to genes with low expression (Extended Data Fig. 2c). We used computational data integration (Methods) to jointly cluster and identify 116 cell types using all the datasets (Fig. 1b, c, Extended Data Fig. 2d, Supplementary Tables 2, 3). Cells and nuclei, assayed by each of the technologies and in each batch, grouped primarily by cell type and not by dataset (Fig. 1b). Residual systematic differences between nuclear and cellular RNA-seq assays were observed in some clusters as a gradient of transcriptomes from different datasets. We performed hierarchical clustering to uncover the relationships among types within each major cell class: GABAergic inhibitory neurons ($n = 59$ types), glutamatergic excitatory neurons ($n = 31$) and non-neurons ($n = 26$) (Fig. 1d). Six of the transcriptomic datasets used cell-sorting strategies to enrich neurons relative to non-neuronal cells, while the largest dataset (snRNA-seq 10x v3 B) represents an unbiased sample of both neuronal and non-neuronal cells. Despite these differences, the relative frequency of cell types was highly consistent across datasets after normalizing for the total sample of each major class (Supplementary Table 3). Most cell types (86 out of 116) were present in all of the datasets, whereas the rest were non-neuronal types that were under-sampled in many datasets or were extremely rare types (less than 0.01% of all cells).

To facilitate the use of these cell types by investigators, we adopted a nomenclature that incorporates multiple anatomical and molecular identifiers. For example, we identified four clusters of excitatory neurons (expressing *Slc17a7*, which encodes the vesicular glutamate transporter VGLUT1) that express a deep layer marker, *Fezf2*, as well as *Fam84b*, which is a unique marker of the pyramidal tract[3] or extratelencephalically- projecting neurons (ET)[16] (Fig. 1e). Thus, we labelled these neurons 'L5 ET 1–4'. We divided GABAergic neurons into five major subclasses based on marker genes: *Lamp5*, *Sncg* and *Vip*, which label cells derived from the caudal ganglionic eminence, and *Sst* and *Pvalb*, which label cells derived from the medial ganglionic eminence. Finer distinctions among GABAergic types are identified by secondary markers (for example, *Sst* and *Myh8*). Tables of cluster accession IDs and differentially expressed genes between every pair of cell types help to track the cell types and their underlying molecular evidence[17] (Supplementary Tables 3, 6).

We compared our MOp atlas with a large dataset of neurons from the mouse anterolateral motor cortex and the primary visual cortex assayed by scRNA-seq (SMART-Seq)[3] (Extended Data Fig. 3a). We found one-to-one matches between most of the 116 MOp cell types and the 102 cell types previously defined in the anterolateral motor cortex. Four types of L5 ET neurons correspond with three previously described deep layer excitatory neurons with distinct subcortical projection patterns to the thalamus and the medulla[18] (Extended Data Fig. 3b, c). These types, which were associated with distinct roles in movement planning and initiation, had consistent patterns of differential gene expression across the transcriptomic datasets (Extended Data Fig. 4).

The motor cortex is traditionally considered to lack a discernible L4 based on the absence of a clear cytoarchitectonic signature[19]. However, recent anatomical studies have identified a population of pyramidal cells located between L3 and L5, with hallmarks of L4 neurons including thalamic input and outputs to L4 and L2/3 (ref. [4]). We identified two intratelencephalically projecting (IT) clusters, containing over 99,000 cells, which express a combination of markers usually associated with L4 (ref. [20]), including *Cux2*, *Rspo1* and *Rorb* (both clusters), and those associated with L5, for example, *Fezf2* (one cluster) (Fig. 1e, Extended Data Fig. 5a). We confirmed the specificity of the expression of these genes in the MOp by in situ hybridization (Extended Data Fig. 5b). These cells represent a substantial fraction (18% or more) of all excitatory neurons in each dataset. Therefore, we labelled these clusters L4/5. Moreover, the localization of cells with these gene markers in middle layers is further supported by spatial transcriptomics[21].

Using our integrated dataset, we directly compared the nuclear and cytoplasmic transcriptomes of MOp cells. Both modalities can achieve

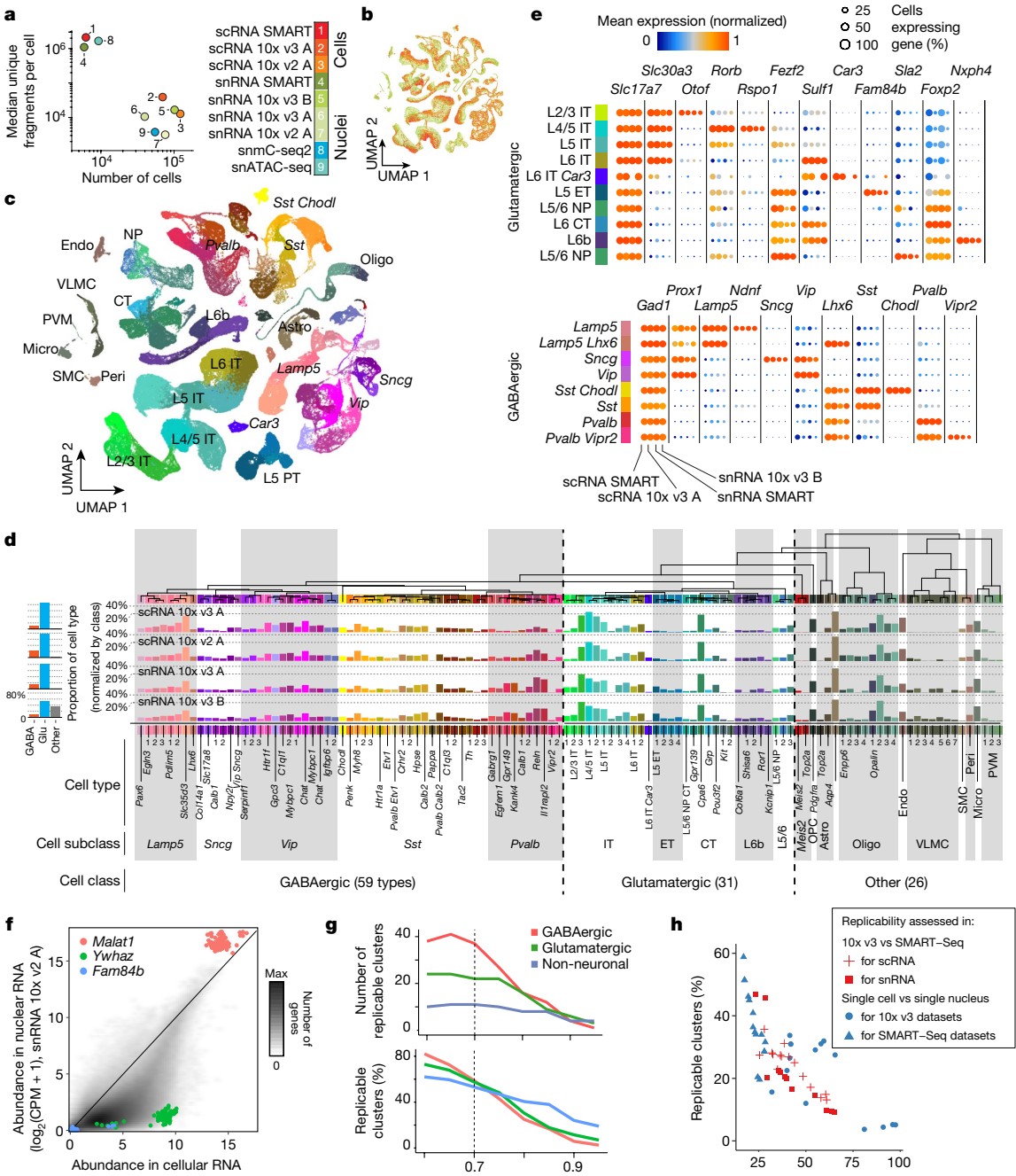

**Fig. 1 | Multi-platform transcriptomic taxonomy of the cell types in the MOp. a**, Key attributes of nine single-cell transcriptomic and epigenomic datasets from the mouse MOp. **b**, **c**, Two-dimensional projection (uniform manifold approximation and projection (UMAP)[40]) of cells and nuclei based on integrated analysis of seven transcriptomic (scRNA-seq and snRNA-seq) datasets. Cells and nuclei are coloured by dataset (**b**) using the colours shown in **a**, or by cell type (**c**). Non-neuronal cell types are depleted owing to the sampling strategy, which enriched neurons in all datasets except snRNA 10x v3 B. **d**, Dendrogram showing a hierarchical relationship among the consensus transcriptomic cell types and a proportion of cells of each type per dataset, normalized within major classes. Glu, glutamatergic. **e**, Expression of selected marker genes for excitatory (top) and inhibitory (bottom) cell classes, across

four platforms. **f**, Differential enrichment of transcripts in single cells versus single nuclei. The long non-coding RNA *Malat1* is enriched in nuclei. CPM, counts per million reads mapped. **g**, The number of replicable clusters across at least two of the seven scRNA-seq and snRNA-seq datasets as a function of the minimal MetaNeighbor score (AUROC). **h**, The trade-off between the number of clusters and replicability (the per cent of clusters with minimal MetaNeighbor replicability score). The major inhibitory neuron subclasses are *Lamp5*, *Sncg*, *Vip*, *Sst* and *Pvalb*. Astro, astrocyte; CT, corticothalamic; endo, endothelial; ET, extratelencephalically projecting; IT, intratelencephalically projecting; micro, microglial cell; NP, near-projecting; oligo, oligodendrocyte; OPC, oligodendrocyte precursor cell; peri, pericyte; PVM, perivascular macrophage; SMC, smooth muscle cell; VLMC, vascular leptomeningeal cell.

comparable clustering resolution (Extended Data Fig. 2d), as previously reported[22], but they provide distinct information about some cell types and transcripts. We found that the long non-coding RNA *Malat1* was enriched in snRNA-seq, consistent with its nuclear localization[23] (Fig. 1f,

Extended Data Fig. 2f). By contrast, mRNA of the protein-coding gene *Ywhaz* was strongly depleted from the nucleus.

We used MetaNeighbor to assess the cross-dataset replicability of clusters defined separately using each of the seven transcriptomic

datasets[24] (Supplementary Table 4). We found 70 clusters with a high replicability (area under receiver operating characteristic (AUROC) > 0.7 across at least two datasets) (Fig. 1g). Most clusters had reciprocal best matches across all datasets (Extended Data Fig. 8a). By comparing the results of three different widely used single-cell analysis packages[25–27], we found lower replicability for fine-grained partitions of cells into 30 or more clusters (Fig. 1h). These results highlight the importance of careful biologically informed cluster analyses.

## Combining transcriptomes and epigenomes

Regions of open chromatin and patterns of DNA methylation, including CG and non-CG methylation, are cell-type-specific signatures of neuronal identity and can be assayed in single nuclei[9,11]. We applied snmC-seq2 (ref. [14]) (9,876 cells) and snATAC-seq[28] (81,196 cells) assays to nuclei isolated from the same MOp samples. Independent analyses of each epigenomic dataset identified $n$ = 42 cell types using DNA methylation, and $n$ = 33 cell types using open chromatin (Extended Data Fig. 6a–d, Supplementary Table 4). Marker genes for major cell classes had corresponding patterns of cell-type-specific depletion of non-CG methylation (low mCH; Extended Data Fig. 6b) and open chromatin in the gene body (Extended Data Fig. 6d).

We integrated eight transcriptomic and epigenomic datasets using two computational methods (linked inference of genomic experimental relationships (LIGER)[29] and SingleCellFusion[30]) to produce a unified, multimodal cell census (Fig. 2a–c, Extended Data Figs. 6e–j, 7a, b, Supplementary Table 5). We reasoned that cells of the same type measured in each modality can be identified based on correlated gene-centric features. Gene expression is negatively correlated with gene body non-CG methylation[9] and positively related to the gene body and promoter ATAC-seq read density[31]. Although distal regulatory elements (for example, enhancers) were not used for dataset integration, they were subsequently analysed at the level of integrated cell types.

By combining cells from integrated clusters into pseudo-bulk tracks, we obtained base-resolution epigenomic and transcriptomic information (Fig. 2f, g) (https://brainome.ucsd.edu/BICCN_MOp). To illustrate, we highlight the locus of *Tac1*, which encodes a precursor of the neuropeptide substance P and marks a subset of interneurons derived from the medial ganglionic eminence[32]. We confirmed *Tac1* mRNA expression in parvalbumin-expressing neurons marked by *Reln* and *Calb1*. We further observed accessible chromatin and low DNA methylation at CG sites within the body of the *Tac1* gene and at a location approximately 24 kb upstream of the transcription start site (Fig. 2f).

Both computational integration methods (LIGER and SingleCellFusion) identified 56 cell types, which showed a high degree of concordance between the methods and with the transcriptome-based consensus clusters (Extended Data Fig. 7a–d). Indeed, integrated analysis identified more cell types than the single-modality analysis of each epigenomic dataset, while largely concurring with the independent clusters (Extended Data Fig. 7b). Integration revealed notable examples of cross-modal cell-type-specific signatures. For example, *Tshz2* is a specific marker of L5 near-projecting excitatory neurons, with low DNA methylation (mCG and mCH), open chromatin and strong cell-type-specific expression (Fig. 2d, e, g). The close correspondence between transcriptomic and epigenomic signatures at *Tshz2*, and at 35 markers of other cell types, was evident across each of the datasets (Fig. 2d). Importantly, these pseudo-bulk tracks include data, such as CG methylation and intergenic snATAC-seq signals, that were not used for the multimodal computational integration.

In addition to concordant cross-modal signals, we also found loci where transcriptomic and epigenomic data diverged. For instance, at *Lhx9*, we found high DNA methylation in L6b excitatory neurons, with little or no methylation in any other cell type (Fig. 2g, Extended Data Fig. 7f). Despite this cell-type-specific DNA methylation, we found no expression of *Lhx9* RNA in any cell type and no significant enrichment of ATAC-seq reads. *Lhx9* has been implicated in early developmental patterning of the caudal forebrain and may be transcriptionally silenced in the adult, potentially through Polycomb-mediated repression[33]. Other regulators of neural development, such as *Pax6* and *Dlx1/2*, have a similar epigenetic profile with cell-type-specific hypermethylation. This pattern may represent a vestigial epigenetic signature of embryonic development[34].

## Cell-type-specific epigenomic marks

Epigenomic data identify potential regulatory regions, such as distal enhancers, marked by open chromatin and low DNA methylation (mCG). These modalities have complementary technical characteristics, such as the number of cells assayed (higher for open chromatin) and the genomic coverage per cell (higher for DNA methylation; Fig. 1a). We first defined differentially methylated regions (DMRs) and chromatin accessibility peaks independently, identifying over 1.3 million DMRs covering 225 Mb (8.3% of the genome) and 300,000 accessible regions (170 Mb) (Fig. 3a, b). In each cell type, a large fraction of accessible regions (28–89%) overlapped hypomethylated DMRs (Fig. 3a). By contrast, many DMRs did not overlap accessibility peaks (Fig. 3b). In some cases, these DMRs coincided with broad open chromatin regions, such as whole gene bodies, which had no narrow ATAC peaks.

By downsampling data from two abundant cell types (L2/3 IT and L6 CT neurons), we found that the number of detectable accessibility peaks was saturated after sampling around 1,000 cells (Fig. 3c). By contrast, the number of DMRs reached a plateau after sampling 200–300 cells (Fig. 3d). Furthermore, the number of significantly enriched transcription factor motifs increased with the number of cells (Fig. 3e); although for L6 CT neurons, it reached a plateau of approximately five key motif families after sampling around 100 cells.

Combining both epigenomic datasets, we identified 250,000 putative enhancers with fine resolution[35] (Supplementary Table 7). Putative enhancers were often found in distal regions, at least 2 kb from the nearest transcription start site (Fig. 3h, i). Sequence motifs of several transcription factor families were enriched in each cell type (Fig. 3f), such as *Rfx* motifs in L2/3 neurons. Using the transcriptomic data, we found that *Rfx3*, but not other *Rfx* family members, was specifically enriched in L2/3 neurons and had low methylation and accessible chromatin in the gene body as well as approximately 15 kb upstream of the *Rfx3* promoter (Fig. 3g). These data suggest a key role for *Rfx3* in L2/3 neurons.

## Reproducible cell types across datasets

Different molecular modalities, sampling strategies, sequencing technologies and computational analysis procedures can lead to divergent estimates of the total number of cell types. We used systematic cross-dataset analyses to assess the statistical and biological reproducibility of cell types and constrain the range of plausible numbers of cell types based on current single-cell sequencing data.

We first addressed the effect of the number of sampled cells on the resolution of the cell atlas, by downsampling each dataset followed by clustering analysis with a fixed resolution parameter (Fig. 4a). The number of detected neuronal cell types (clusters) increased logarithmically with cell number, with relatively few additional clusters detected after sampling approximately 80,000 cells or nuclei. Notably, the dependence of the number of clusters on the number of sampled cells was similar for all modalities and datasets, showing that the number of sampled cells is a key determinant of cluster resolution.

Any dataset can be divided into increasingly fine-grained clusters, yet they may not reflect biologically meaningful or reproducible cell-type distinctions. We used cross-validation to objectively measure the generalizability of cluster-based descriptions of the data (Extended Data Fig. 8b). We first used within-dataset cross-validation, dividing the features (genes or genomic bins) into clustering and validation sets. After clustering all cells using the clustering feature set, we split the cells into training and test sets. We used the training cells to learn the validation set features

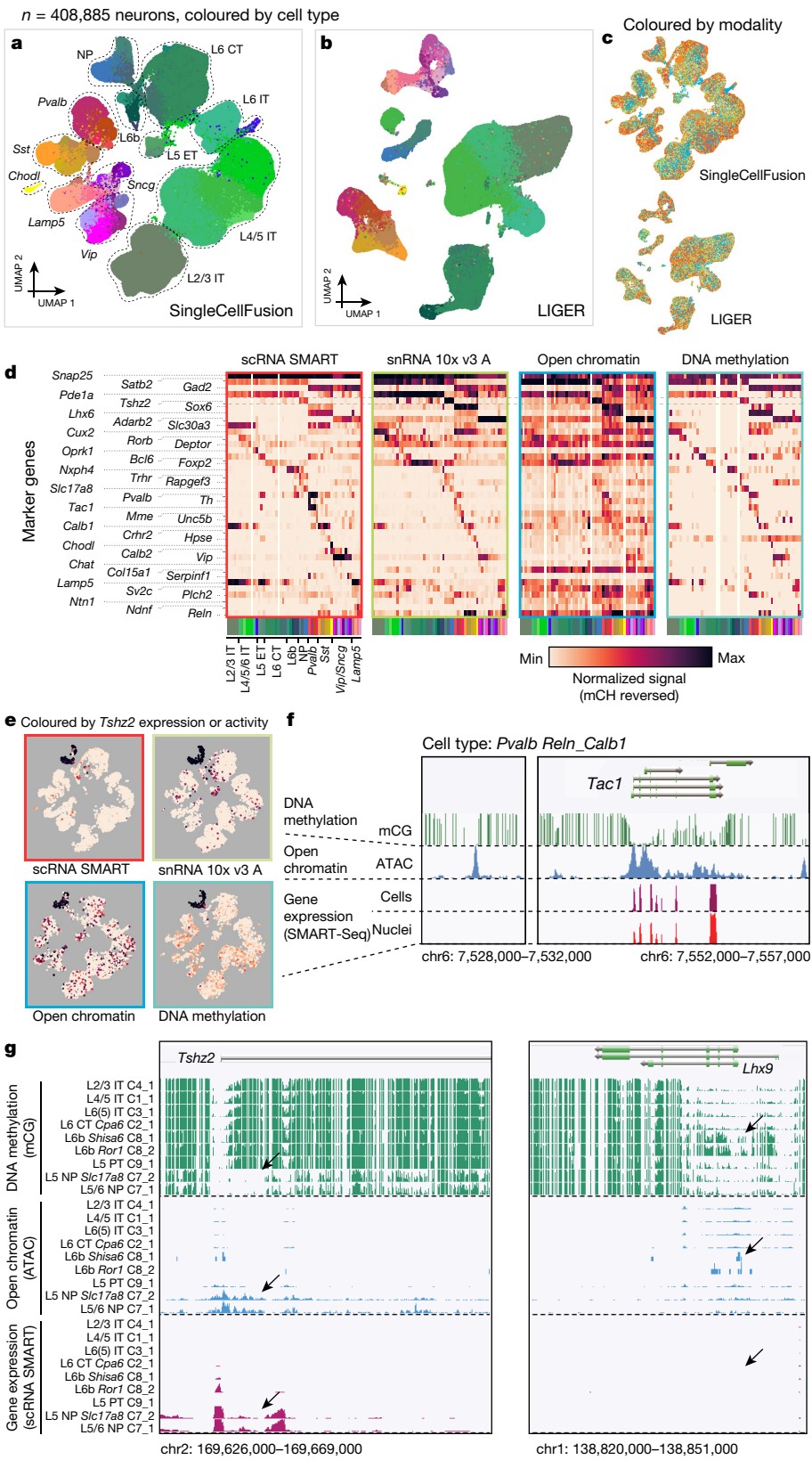

**Fig. 2 | Integration of epigenomes and transcriptomes. a, b,** Two-dimensional projection (UMAP) of more than 400,000 individual cells and nuclei from eight transcriptomic and epigenomic datasets (excluding snRNA 10x v2 A), integrated using SingleCellFusion (**a**) or LIGER (**b**). Cells are coloured by joint clustering assignments from the respective integration method. **c,** UMAP projection with cells coloured by dataset (colour scheme as in Fig. 1a). **d,** Selected marker genes across data modalities. The grey dashed box highlights the gene *Tshz2*. **e,** UMAP embeddings coloured by the level of mRNA expression, accessibility or DNA methylation at *Tshz2*. **f,** Browser view of the *Tac1* locus comparing four datasets with base-resolution transcriptomic and epigenomic information for one cell type: *Pvalb Reln_Calb1*. See https://bit.ly/2Hhb0VY. Chr6, chromosome 6. **g,** Browser showing excitatory cell-type tracks. *Tshz2* consistently marks L5 NP cell types across data modalities, while *Lhx9* has a unique epigenetic signature in L6b cell types in DNA methylation only. The black arrows show cell-type-specific demethylation, open chromatin or gene expression. See https:bit.ly/3bABMX2 for *Tshz2* and https://bit.ly/2HioFMv for *Lhx9*.

for each cluster. Finally, we compared the validation set features with the held-out data for test cells to measure the mean squared error. We applied this procedure to each dataset with a range of clustering resolutions, resulting in a U-shaped cross-validation curve for the test set error as a function of the number of clusters (Fig. 4b, Extended Data Fig. 8c, d). The location of the minimum mean squared error is an estimate of the number of reliable clusters. Finally, we repeated this cross-validation procedure for each dataset in combination with systematic downsampling (Fig. 4c).

All of the datasets (except snRNA SMART-Seq) supported approximately 100 or more cell types when a sufficient number of cells was sampled. The number of cells required to achieve this resolution was larger for snATAC-seq (with few reads per cell) than for RNA-seq or

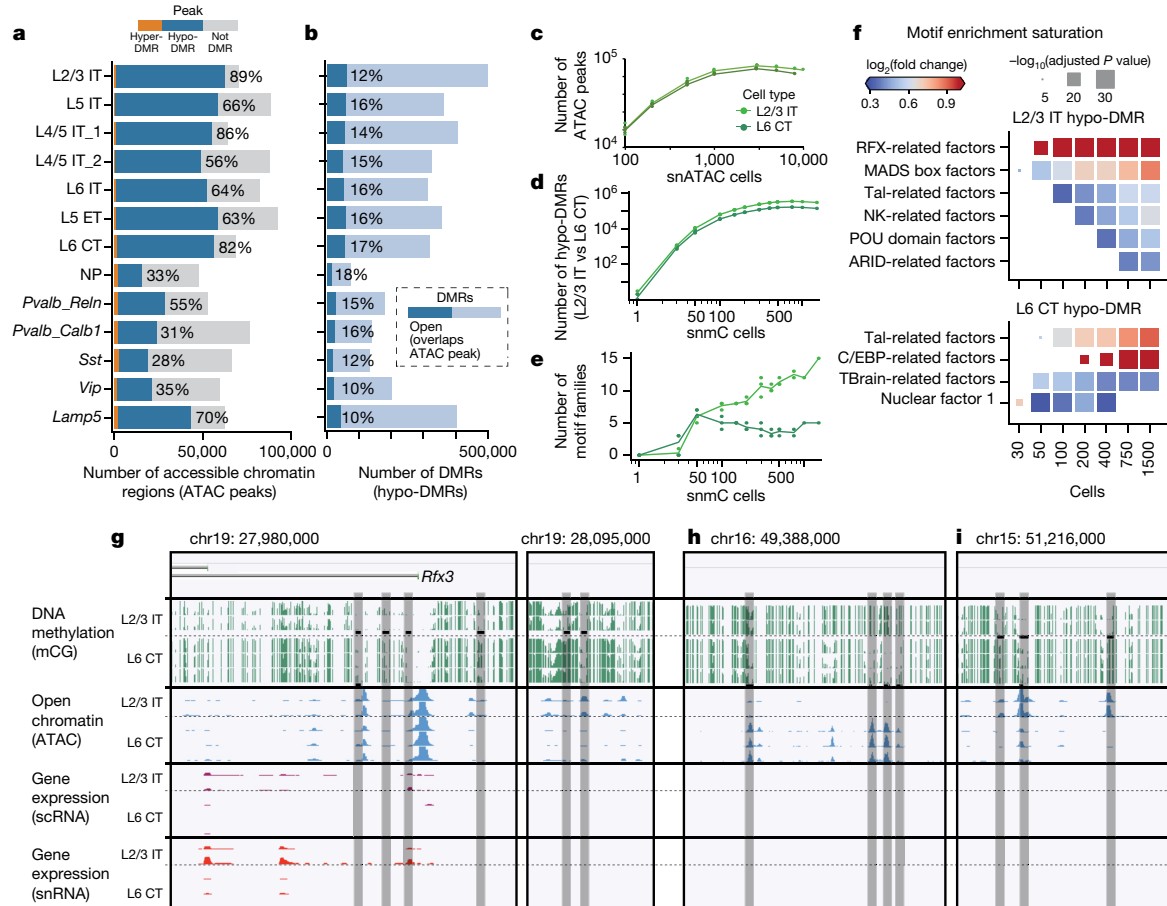

**Fig. 3 | Epigenomic signatures of regulatory elements in the MOp.**
**a**, **b**, Regulatory regions were identified in each cell type using DMRs
($n = 1,302,403$) (**a**) and open chromatin regions (ATAC peaks; $n = 316,788$) (**b**)
in multimodal integrated clusters. **c**, **d**, Saturation analysis for two excitatory
subclasses shows the number of regulatory regions detected as a function
of sampled cells. **e**, Saturation analysis of the number of transcription
factor DNA-binding sequence motifs enriched in DMRs of each cell type.
**f**, Enrichment of motifs for selected transcription factor families as a function
of the number of cells sampled. **g**, Browser views of loci containing cell-type-
specific regulatory elements (grey highlighted regions). The *Rfx3* gene is
differentially expressed in L2/3 neurons and has an enhancer specific to L2/3
located approximately 15 kb upstream of the promoter region. **h**, **i**, Examples of
intergenic regions with accessibility and demethylation specific to L6 CT (**h**) or
L2/3 (**i**) neurons. The black bars indicate predicted regulatory regions.

snmC-seq2. This observation is consistent with the relative sparseness
of the snATAC-seq data. We further found that scRNA-seq and snRNA-seq
datasets with the largest numbers of cells could support very high cluster
resolution with up to approximately 600 clusters. Our cross-validation
analysis shows that these fine-grained clusters capture genuine transcrip-
tomic structure, which is correlated and replicable across cells and across
genomic features. However, at least some of this structure probably cor-
responds to continuous variation within discrete cell types, rather than
discrete cell-type categories[36]. Moreover, the cross-validation analysis
shows no sharp error minimum at a particular value of the number of
clusters. Instead, the U-shaped cross-validation curve has a broad basin
covering a range of plausible values (Fig. 4b, Extended Data Fig. 8c, d).

To more stringently test the reproducibility of cell types, we per-
formed cross-dataset cross-validation (Extended Data Fig. 8b). This
procedure uses a randomly chosen half of genomic features to perform
data integration and joint analysis of eight datasets using SingleCellFu-
sion. Next, we used the joint cluster labels to perform cross-validation
in each dataset, as in the within-dataset procedure above. This analysis
supported a maximum resolution of approximately 100 clusters when
testing using the scRNA SMART-Seq data (Fig. 4d).

As an alternative to joint analysis of multiple datasets, which could
potentially discern spurious correlations owing to computational data
integration, we also took a more stringent approach to cross-validation.
Using the independent cluster analysis of each dataset, we performed

MetaNeighbor analysis to assess the replicability of clusters[24]. We
found that the median replicability score for all clusters was high
(AUROC > 0.8) for integrated analyses with coarse resolution (less
than 50 clusters, level 1 analyses; Fig. 4e). The more fine-grained joint
analyses (level 2; 50–120 clusters) were also largely supported by
MetaNeighbor, but with a lower median replicability score around 0.7.
Notably, we found a high degree of consistency in the results of joint
cluster analysis when using different computational methods (Fig. 4f).

Finally, we explored whether cell-type signatures in the MOp were
stable across different scRNA-seq and snRNA-seq platforms. Using four
RNA-seq datasets (scRNA SMART, snRNA SMART, scRNA 10x v3 A and snRNA
10x v3 A), we performed clustering on a network of samples (Conos[37]) to
link cells across datasets and determine joint clusters. We compared the
clustering results based on inter-platform network connections only ver-
sus results that also included connections across datasets of the same
platform (Extended Data Fig. 8e). Most neuron types, except parvalbumin-
expressing interneurons and L6 CT, had only a modest difference in cluster
stability using both approaches (Fig. 4g) and a low level of inter-platform
divergence in their cell-type transcriptomic signatures (Fig. 4h).

## Discussion

Our MOp cell atlas represents the most comprehensive, integrated
collection of single-cell transcriptomic and epigenomic datasets for a

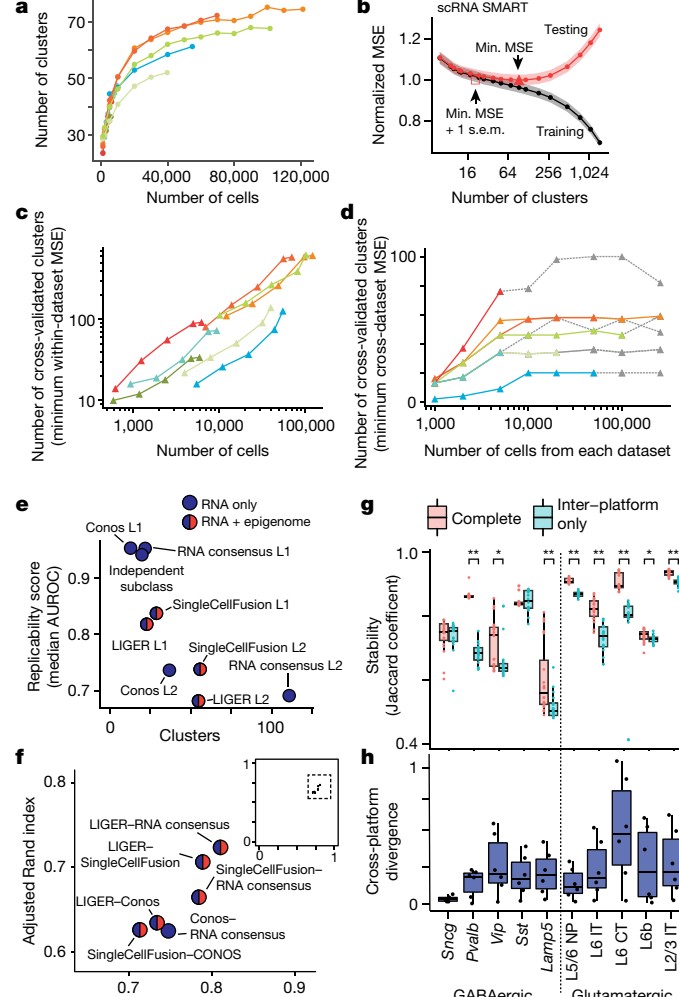

**Fig. 4 | Robustness and reproducibility of cell types within and across datasets. a**, The number of clusters estimated for each dataset after sampling a fraction of the total cells (Leiden clustering, resolution $r = 6$; colour scheme as in Fig. 1a). **b**, Mean squared error (MSE) as a function of the number of clusters for scRNA SMART-Seq. The minimum MSE and the minimum MSE +1 s.e.m. define a range of optimal cluster resolutions. The shaded region shows the s.e.m. derived from cross-validation with $n = 5$ random data partitions.
**c**, **d**, The number of clusters estimated by within-dataset (**c**) or across-dataset (**d**) cross-validation ($n = 5$ data partitions) (colour scheme as in Fig. 1a). For cross-dataset comparison (**d**), the number of clusters is based on the minimum test MSE for one dataset after joint multimodal clustering. The grey dotted lines are shown when the number of cells ($x$ axis) exceeds the dataset size, in which case all of the cells from the corresponding modality were used.
**e**, The trade-off between the number of clusters and replicability (median MetaNeighbor AUROC) of consensus clustering methods applied at various resolutions. **f**, Agreement between consensus clustering results using different computational procedures. Inset: zoomed-out view showing that all methods have high cluster purity and adjusted Rand index. **g**, Transcriptomic platform consistency is assessed by cross-dataset cluster stability analysis (Conos) using complete networks, and using inter-platform edges only. Glutamatergic and Pvalb subclasses have reduced stability in inter-platform comparison. Data points show $n = 20$ independent random samples, each containing 95% of the total cells. **h**, Cross-platform expression divergence (Jensen–Shannon) for major cell subclasses. The box-and-whisker plots (**g**, **h**) show the median, the interquartile range (25–75th percentile), and the smaller of the data range (minimum to maximum) or the 1.5 times the interquartile range. *False discovery rate (FDR) ≤ 0.05, **FDR ≤ 0.0001, Wilcoxon rank-sum test, Benjamini–Hochberg correction.

single brain region to date. We generated a high-resolution consensus transcriptomic cell-type taxonomy that integrates seven scRNA-seq and snRNA-seq datasets collected from the MOp with six experimental methods. Our transcriptomic taxonomy is highly consistent with a previously published transcriptomic cell census from the primary visual and the anterolateral motor cortices based on SMART-Seq alone[3]. We found that gene expression profiles were largely consistent across methodologies, while providing complementary information about particular genes such as nucleus-enriched transcripts. The MOp atlas demonstrates the power of a two-pronged strategy that combines broad sampling of diverse cell types (for example, 10x with a large number of cells and shallow sequencing) with deep sequencing (for example, SMART-Seq) to precisely characterize gene expression profiles for each cell type. This strategy should guide future cell census efforts, by the BICCN and others, at the scale of whole brains and in other species.

We further demonstrated multimodal integration of transcriptomic (scRNA-seq and snRNA-seq), DNA methylation (snmC-seq2) and chromatin accessibility (snATAC-seq) datasets using two computational methods (SingleCellFusion and LIGER). It is possible to directly establish links between molecular modalities through simultaneous measurement of multiple signatures in the same cell[38]. However, multimodal single-cell assays remain challenging and often provide lower depth or resolution of data in each modality than single-modality assays. Moreover, it is important to show that data collected from different animals, across different laboratories and using different experimental platforms and assays, nevertheless can be integrated within a unified cell-type atlas. By correlating mRNA transcripts, gene body methylation and accessibility peaks, we showed that different types of data can be integrated without forfeiting the resolution of more than 50 fine-grained neuron types. Integrative analysis of transcriptional and epigenetic signatures of cell identity will enable the development of tools based on cell-type-specific enhancers for cell targeting and manipulation.

Our data provide new insights into the molecular architecture of cell types in the MOp. *Tac1*, encoding the neuropeptide substance P precursor, marks a subset of parvalbumin-expressing cells and is strongly upregulated in the rodent MOp following motor learning[32,39]. We found that *Tac1* is expressed in two subtypes of MOp interneurons (*Pvalb_Calb1* and *Pvalb_Reln*), and our epigenomic data identified a cell-type-specific enhancer approximately 24 kb upstream of the gene promoter. We provide new evidence that the MOp has an excitatory neuron population that expresses markers of L4 thalamic-recipient neurons, including *Cux2*, *Rspo1* and *Rorb*[4]. The laminar distribution of these cells has been confirmed by in situ hybridization of these marker genes and in a parallel study by MERFISH[21]. This discovery revises the traditional understanding of the MOp as an agranular cortex lacking L4. We also found multiple types of L5 ET neurons that align with recently described populations with distinct subcortical projection targets[18]. Moreover, we identified networks of gene expression regulatory elements, marked by overlapping regions of open chromatin and cell-type-specific demethylation, that have sequence motifs that identify the key transcriptional regulators. For example, by combining epigenetic and gene expression data, we identified *Rfx3* as a candidate factor for L2/3 IT cells. We also identified genes with non-canonical regulatory signatures, such as enrichment of mCG in *Lhx9*, specifically in L6b excitatory cells.

We took advantage of the unprecedented diversity of large-scale datasets, generated in a coordinated manner from the mouse MOp, to critically evaluate the robustness and reliability of the cell-type taxonomies obtained by clustering molecular datasets. Our cross-validation analysis of individual datasets and multimodal integration objectively constrains the range of cluster resolutions supported by the data without overfitting. Rather than supporting a single, definitive number of cell types in the mouse MOp, our studies instead point to a range of cluster resolutions spanning from approximately 30 to 116 cell types that are supported by the data. Indeed, discrete cell-type categories

may be an inappropriate description at a fine-grained level of analysis, in which the molecular profiles of cells vary along a continuum.

By integrating nine large-scale single-cell transcriptomic and epigenomic datasets, we have comprehensively classified and annotated the diversity of cell types in the adult mouse MOp. Our study demonstrates general procedures for objective cross-dataset comparison and statistical reproducibility analysis, as well as standards and best practices that can be adopted for future large-scale studies. Together with complementary BICCN datasets from spatial transcriptomics, connectivity and physiology, as well as cross-species comparative studies, our results help to establish a multifaceted understanding of brain cell diversity. Targeted studies of individual cell types, taking advantage of the transcriptional and epigenetic signatures described here, will define their functional roles and significance in the context of neural circuits and behaviour. Integrative analyses will be essential to make progress towards understanding the organizing principles of cell types in the brain through their molecular genetic signatures.

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

¹Allen Institute for Brain Science, Seattle, WA, USA. ²Genomic Analysis Laboratory, The Salk Institute for Biological Studies, La Jolla, CA, USA. ³Department of Physics, University of California, San Diego, La Jolla, CA, USA. ⁴Stanley Institute for Cognitive Genomics, Cold Spring Harbor Laboratory, Cold Spring Harbor, NY, USA. ⁵Institute for Genome Sciences, University of Maryland School of Medicine, Baltimore, MD, USA. ⁶Computational Neurobiology Laboratory, The Salk Institute for Biological Studies, La Jolla, CA, USA. ⁷Department of Statistics, University of California, Berkeley, Berkeley, CA, USA. ⁸Department of Applied Mathematics, Computer Science and Statistics, Ghent University, Gent, Belgium. ⁹Division of Biostatistics, School of Public Health, University of California, Berkeley, Berkeley, CA, USA. ¹⁰Broad Institute of MIT and Harvard, Cambridge, MA, USA. ¹¹California Institute of Technology, Pasadena, CA, USA. ¹²Center for Bioinformatics and Computational Biology, University of Maryland, College Park, College Park, MD, USA. ¹³Johns Hopkins School of Medicine, Department of Neurology, Baltimore, MD, USA. ¹⁴Johns Hopkins School of Medicine, Department of Neuroscience, Baltimore, MD, USA. ¹⁵University of Maryland School of Medicine, Institute for Genome Sciences, Baltimore, MD, USA. ¹⁶Department of Cognitive Science, University of California, San Diego, La Jolla, CA, USA. ¹⁷Bioinformatics and Systems Biology Graduate Program, University of California, San Diego, San Diego, CA, USA. ¹⁸Department of Otorhinolaryngology, Anatomy and Neurobiology, University of Maryland School of Medicine, Baltimore, MD, USA. ¹⁹Center for Epigenomics, Department of Cellular and Molecular Medicine, University of California, San Diego School of Medicine, La Jolla, CA, USA. ²⁰Department of Biomedical Informatics, Harvard Medical School, Boston, MA, USA. ²¹Ludwig Institute for Cancer Research, La Jolla, CA, USA. ²²Department of Human Genetics, University of California, Los Angeles, Los Angeles, CA, USA. ²³Howard Hughes Medical Institute, The Salk Institute for Biological Studies, La Jolla, CA, USA. ²⁴University of California, San Francisco, San Francisco, CA, USA. ²⁵Department of Statistical Sciences, University of Padova, Padova, Italy. ²⁶Department of Data Sciences, Dana-Farber Cancer Institute, Boston, MA, USA. ²⁷Cold Spring Harbor Laboratory, Cold Spring Harbor, NY, USA. ²⁸Department of Molecular and Cell Biology, University of California, Berkeley, Berkeley, CA, USA. ²⁹Howard Hughes Medical Institute, Department of Biology, MIT, Cambridge, MA, USA. ³⁰Department of Computational Medicine and Bioinformatics, University of Michigan, Ann Arbor, MI, USA. ³¹Present address: McDonnell Genome Institute, Washington University School of Medicine, St Louis, MO, USA. ³²These authors contributed equally: Zizhen Yao, Hanqing Liu, Fangming Xie, Stephan Fischer. ✉e-mail: HongkuiZ@alleninstitute.org; emukamel@ucsd.edu

# Methods

## Tissue collection and isolation of cells or nuclei (RNA-seq at the Allen Institute)

The following methods apply to the following transcriptomic datasets generated at the Allen Institute: scRNA SMART, scRNA 10x v3 A, scRNA 10x v2 A, snRNA SMART, snRNA 10x v3 A and snRNA 10x v2 A.

**Mouse breeding and husbandry.** All procedures were carried out in accordance with the Institutional Animal Care and Use Committee protocols at the Allen Institute for Brain Science. Mice were provided food and water ad libitum and were maintained on a regular 12-h day/night cycle at no more than five adult mice per cage. Ambient temperature was set to 72 °F and relative humidity was set to 40%. All rooms were on 12/12-h light/dark cycle. For this study, we enriched for neurons by using *Snap25-IRES2-Cre* mice[41] (MGI: J:220523) crossed to *Ai14* (ref. [42]) (MGI: J:220523), which were maintained on the C57BL/6J background (RRID: IMSR_JAX:000664). Mice were euthanized at 53–59 days of postnatal age. Tissue was collected from both males and females (scRNA SMART, snRNA SMART, scRNA 10x v3 A and snRNA 10x v2 A), only males (scRNA 10x v2 A) or only females (snRNA 10x v3 A).

**Single-cell isolation.** We isolated single cells by adapting previously described procedures[3,43]. The brain was dissected, submerged in artificial cerebrospinal fluid (ACSF)[3], embedded in 2% agarose, and sliced into 250-μm (SMART-Seq) or 350-μm (10x Genomics) coronal sections on a Compresstome (Precisionary Instruments). The Allen Mouse Brain Common Coordinate Framework version 3 (CCFv3; RRID: SCR_002978)[44] ontology was used to define the MOp for dissections (Extended Data Fig. 1b).

For SMART-Seq, the MOp was microdissected from the slices and dissociated into single cells with 1 mg/ml pronase (P6911-1G, Sigma) and processed as previously described[3]. For 10x Genomics, tissue pieces were digested with 30 U/ml papain (PAP2, Worthington) in ACSF for 30 min at 30 °C. Enzymatic digestion was quenched by exchanging the papain solution three times with quenching buffer (ACSF with 1% FBS and 0.2% BSA). The tissue pieces in the quenching buffer were triturated through a fire-polished pipette with a 600-μm diameter opening approximately 20 times. The solution was allowed to settle and supernatant containing single cells was transferred to a new tube. Fresh quenching buffer was added to the settled tissue pieces, and trituration and supernatant transfer were repeated using 300-μm and 150-μm fire-polished pipettes. The single-cell suspension was passed through a 70-μm filter into a 15-ml conical tube with 500 μl of high BSA buffer (ACSF with 1% FBS and 1% BSA) at the bottom to help cushion the cells during centrifugation at 100*g* in a swinging bucket centrifuge for 10 min. The supernatant was discarded, and the cell pellet was resuspended in a quenching buffer.

All cells were collected by fluorescence-activated cell sorting (FACS; BD Aria II; RRID: SCR_018091) using a 130-μm nozzle. Cells were prepared for sorting by passing the suspension through a 70-μm filter and adding DAPI (to the final concentration of 2 ng/ml). The sorting strategy was as previously described[3], with most cells collected using the tdTomato-positive label. For SMART-Seq, single cells were sorted into individual wells of eight-well PCR strips containing lysis buffer from the SMART-Seq v4 Ultra Low Input RNA Kit for Sequencing (634894, Takara) with RNase inhibitor (0.17 U/μl), immediately frozen on dry ice and stored at −80 °C. For 10x Genomics, 30,000 cells were sorted within 10 min into a tube containing 500 μl of quenching buffer. Each aliquot of 30,000 sorted cells was gently layered on top of 200 μl of high BSA buffer and immediately centrifuged at 230*g* for 10 min in a swinging bucket centrifuge. The supernatant was removed and 35 μl of buffer was left behind, in which the cell pellet was resuspended. The cell concentration was quantified and immediately loaded onto the 10x Genomics Chromium controller.

## Tissue collection and nuclei isolation (RNA-seq at the Broad Institute)

These methods apply to the snRNA 10x v3 B dataset, generated at the Broad Institute.

**Animal housing.** Mice were group housed with a 12-h light/dark schedule and allowed to acclimate to their housing environment for 2 weeks after arrival. Ambient temperature was set to 70 ± 2 °F and relative humidity was set to 40 ± 10%. All rooms are on 12/12-h light/dark cycle. All procedures involving animals at the Massachusetts Institute of Technology were conducted in accordance with the US National Institutes of Health Guide for the Care and Use of Laboratory Animals under protocol number 1115-111-18 and approved by the Massachusetts Institute of Technology Committee on Animal Care. All procedures involving animals at the Broad Institute were conducted in accordance with the US National Institutes of Health Guide for the Care and Use of Laboratory Animals under protocol number 0120-09-16. Samples were collected from both male and female mice.

**Brain preparation before 10x nuclei sequencing.** At 60 days of age, C57BL/6J mice were anaesthetized by administration of isoflurane in a gas chamber flowing 3% isoflurane for 1 min. Anaesthesia was confirmed by checking for a negative tail pinch response. Mice were moved to a dissection tray and anaesthesia was prolonged via a nose cone flowing 3% isoflurane for the duration of the procedure. Transcardial perfusions were performed with ice-cold pH 7.4 HEPES buffer containing 110 mM NaCl, 10 mM HEPES, 25 mM glucose, 75 mM sucrose, 7.5 mM MgCl₂ and 2.5 mM KCl to remove blood from the brain and other organs sampled. The brain was removed immediately and frozen for 3 min in liquid nitrogen vapour and moved to −80 °C for long-term storage. A detailed protocol is available at protocols.io[15].

**Generation of MOp nuclei profiles.** Frozen mouse brains were securely mounted by the cerebellum onto cryostat chucks with OCT embedding compound such that the entire anterior half, including the MOp, was left exposed and thermally unperturbed. Dissection of 500-μm anterior–posterior spans of the MOp (Extended Data Fig. 1c) was performed by hand in the cryostat using an ophthalmic microscalpel (P-715, Feather safety Razor) precooled to −20 °C and donning 4× surgical loupes. Each excised tissue dissectate was placed into a precooled 0.25-ml PCR tube using precooled forceps and stored at −80 °C. To assess dissection accuracy, 10-μm coronal sections were taken at each 500-μm anterior–posterior dissection junction and imaged following Nissl staining. Nuclei were extracted from these frozen tissue dissectates using gentle, detergent-based dissociation, according to a protocol[45] adapted from one generously provided by the McCarroll laboratory, and loaded into the 10x Chromium v3 system. Reverse transcription and library generation were performed according to the manufacturer's protocol.

This 10x v3 snRNA-seq protocol resulted in a higher number of genes recovered than other snRNA-seq methods. We believe that there are three reasons for this, and that the summation of benefits imparted by the combination of these accounts for the outcome.

First, mouse brains were perfused with a solution emulating ACSF and then rapidly frozen over liquid nitrogen vapour in such a way that RNA integrity was highly preserved. The resulting bioanalyzer RIN scores of the starting brain tissues were routinely 9.8. Storage of the brains before dissection was at −80 °C in the presence of a hydration sink of 1 ml of OCT compound pre-frozen into the bottom of a 5-ml storage tube. This prevents sublimation and subsequent desiccation-dependent RNA fragmentation.

Second, we performed expeditious sample processing. We have a well-trained group of technicians who processed the mouse brain (as above), and then perform the dissociation, FACS and 10x processing

(as below) in one continuous protocol without pauses. For example, each mouse was perfused and ready for dissection within minutes (10 min), and we limited our sample size to six mice so that no sample was waiting to move through the process.

Third, the frozen tissue snRNA-seq protocol incorporates two main features that we believe are important to quality because they prevent the nuclei from 'leaking' valuable signal and simultaneously contaminating the barcoded nuclei mixture with exogenous RNA signal. The first feature was a very low level of centrifugation, which we have found to cause both loss of signal and increased exogenous signal. The second feature was the inclusion of an excipient reagent, BASF Kollidon VA-64, as per the McCarroll laboratory protocol[46].

### Tissue collection and isolation of nuclei for epigenomic samples

The following methods apply to the snmC-seq2 and snATAC-seq datasets generated at the Salk Institute and the University of California, San Diego.

**Tissue preparation for nuclei production.** Procedures involving animals at the Salk Institute were conducted in accordance with the US National Institutes of Health Guide for the Care and Use of Laboratory Animals under protocol number 18-00006 and approved by the Institutional Animal Care and Use Committee. Male C57BL/6J mice were purchased from Jackson laboratories at 8 weeks of age and maintained in the Salk animal barrier facility on 12-h dark/light cycles with controlled temperature (20–22 °C) and humidity (30–70%), and food ad libitum for 1 week before dissection.

Brains were extracted from 56 to 63-day-old mice and immediately sectioned into 0.6-mm coronal sections, starting at the frontal pole, in ice-cold dissection media[9]. The MOp was dissected from slices two to five along the anterior–posterior axis according to the Allen Brain reference Atlas (Extended Data Fig. 1d). Slices were kept in ice-cold dissection media during dissection and immediately frozen in dry ice for subsequent pooling and nuclei production. For nuclei isolation, dissected regions of the MOp from 15 to 23 mice were pooled for each biological replicate, and two replicates were processed for each region. Nuclei were isolated by flow cytometry as described in previous studies[9,10]. In brief, nuclei were produced by homogenization in sucrose buffer as previously described[9], and the nuclei pellet produced was divided into two aliquots. One aliquot underwent sucrose gradient purification and NeuN labelling (snmC-seq2), and the second aliquot went directly to tagmentation (snATAC-seq).

**Bisulfite conversion and library preparation for snmC-seq2.** Detailed methods for bisulfite conversion and library preparation are previously described for snmC-seq2 (ref. [14]), and the protocol is available on protocols.io[47]. The snmC-seq2 libraries were sequenced using an Illumina Novaseq 6000 instrument (RRID: SCR_016387) with S4 flowcells and 150-bp paired-end mode.

**snATAC-seq data generation.** Combinatorial barcoding snATAC-seq was performed as previously described[28,48]. Isolated brain nuclei were pelleted with a swinging bucket centrifuge (500g for 5 min at 4 °C; 5920R, Eppendorf). Nuclei pellets were resuspended in 1 ml nuclei permeabilization buffer (5% BSA, 0.2% IGEPAL-CA630, 1 mM dithiothreitol and cOmplete, EDTA-free protease inhibitor cocktail (Roche) in PBS) and pelleted again (500g for 5 min at 4 °C; 5920R, Eppendorf; RRID: SCR_018092). Nuclei were resuspended in 500 μl high-salt tagmentation buffer (36.3 mM Tris-acetate (pH 7.8), 72.6 mM potassium-acetate, 11 mM Mg-acetate and 17.6% DMF) and counted using a haemocytometer. Concentration was adjusted to 4,500 nuclei per 9 μl, and 4,500 nuclei were dispensed into each well of a 96-well plate. For tagmentation, 1 μl of barcoded Tn5 transposomes[48] were added using BenchSmart 96 (Mettler Toledo; RRID: SCR_018093), mixed five times and incubated for 60 min at 37 °C with shaking (500 r.p.m.). To inhibit the Tn5 reaction, 10

μl of 40 mM EDTA was added to each well with BenchSmart 96 (Mettler Toledo) and the plate was incubated at 37 °C for 15 min with shaking (500 r.p.m.). Next, 20 μl 2× sort buffer (2% BSA and 2 mM EDTA in PBS) were added using BenchSmart 96 (Mettler Toledo). All wells were combined into a FACS tube and stained with 3 μM Draq7 (Cell Signaling). Using a SH800 (Sony), 40 nuclei were sorted per well into eight 96-well plates (a total of 768 wells) containing 10.5 μl EB (25 pmol primer i7, 25 pmol primer i5 and 200 ng BSA (Sigma)). Preparation of sort plates and all downstream pipetting steps were performed on a Biomek i7 Automated Workstation (Beckman Coulter; RRID: SCR_018094). After the addition of 1 μl 0.2% SDS, samples were incubated at 55 °C for 7 min with shaking (500 r.p.m.). Triton-X (12.5%; 1 μl) was added to each well to quench the SDS. Next, 12.5 μl NEBNext High-Fidelity 2× PCR Master Mix (NEB) was added and samples were PCR-amplified (72 °C for 5 min, 98 °C for 30 s (98 °C for 10 s, 63 °C for 30 s and 72 °C or 60 s) ×12 cycles, held at 12 °C). After PCR, all wells were combined. Libraries were purified according to the MinElute PCR Purification Kit manual (Qiagen) using a vacuum manifold (QIAvac 24 plus, Qiagen) and size selection was performed with SPRI Beads (0.55× and 1.5×; Beckmann Coulter). Libraries were purified one more time with SPRI Beads (1.5×, Beckmann Coulter). Libraries were quantified using a Qubit fluorimeter (Life Technologies; RRID: SCR_018095), and the nucleosomal pattern was verified using a Tapestation (High Sensitivity D1000, Agilent). The library was sequenced on a HiSeq2500 sequencer (Illumina; RRID: SCR_016383) using custom sequencing primers, 25% spike-in library and the following read lengths: 50 + 43 + 37 + 50 (Read1 + Index1 + Index2 + Read2)[11].

### Genomic library preparation, sequencing and data processing

**scRNA-seq and snRNA-seq (Allen Institute).** For SMART-Seq processing, we performed the procedures with positive and negative controls as previously described[3]. The SMART-Seq v4 Ultra Low Input RNA Kit for Sequencing (634894, Takara) was used to reverse transcribe poly(A) RNA and amplify full-length cDNA. Samples were amplified for 18 cycles in eight-well strips, in sets of 12–24 strips at a time. All samples proceeded through Nextera XT DNA Library Preparation (FC-131-1096, Illumina) using Nextera XT Index Kit V2 (FC-131-2001, Illumina) and a custom index set (Integrated DNA Technologies). Nextera XT DNA Library preparation was performed according to the manufacturer's instructions, with a modification to reduce the volumes of all reagents and cDNA input to 0.4× or 0.5× of the original protocol.

For 10x v2 processing, we used the Chromium Single Cell 3′ Reagent Kit v2 (120237, 10x Genomics). We followed the manufacturer's instructions for cell capture, barcoding, reverse transcription, cDNA amplification and library construction. We targeted a sequencing depth of 60,000 reads per cell.

For 10x v3 processing, we used the Chromium Single Cell 3′ Reagent Kit v3 (1000075, 10x Genomics). We followed the manufacturer's instructions for cell capture, barcoding, reverse transcription, cDNA amplification and library construction. We targeted a sequencing depth of 120,000 reads per cell.

**RNA-seq data processing and quality control (Allen Institute).** Processing of SMART-Seq v4 libraries was performed as previously described[3]. Briefly, libraries were sequenced on an Illumina HiSeq2500 platform (paired-end with read lengths of 50 bp), and Illumina sequencing reads were aligned to GRCm38.p3 (mm10) using a RefSeq annotation gff file retrieved from the NCBI on 18 January 2016 (https://www.ncbi.nlm.nih.gov/genome/annotation_euk/all/). Sequence alignment was performed using STAR v2.5.3[49]. PCR duplicates were masked and removed using STAR option 'bamRemoveDuplicates'. Only uniquely aligned reads were used for gene quantification. Gene counts were computed using the R GenomicAlignments package (RRID: SCR_018096)[50] and the summarizeOverlaps function in 'IntersectionNotEmpty' mode for exonic and intronic regions separately. For the SMART-Seq v4 dataset, we only used exonic regions for gene quantification. Cells that

met any one of the following criteria were removed: <100,000 total reads, <1,000 detected genes (CPM > 0), <75% of reads aligned to the genome or CG dinucleotide odds ratio > 0.5. Cells were classified into broad classes of excitatory, inhibitory and non-neuronal based on known markers, and cells with ambiguous identities were removed as doublets[3].

10x v2 and 10x v3 libraries were sequenced on Illumina NovaSeq 6000 (RRID: SCR_016387), and sequencing reads were aligned to the mouse pre-mRNA reference transcriptome (mm10) using the 10x Genomics CellRanger pipeline (version 3.0.0; RRID: SCR_017344) with default parameters. Cells were classified into broad classes of excitatory, inhibitory and non-neuronal based on known markers. Low-quality cells that fit the following criteria were filtered from clustering analysis. Different filtering criteria were used for neurons and non-neuronal cells as neurons are bigger than non-neuronal cells and contain more transcripts. For scRNA datasets, we excluded neurons with fewer than 2,000 detected genes and non-neuronal cells with fewer than 1,000 detected genes; for snRNA datasets, we excluded neurons with fewer than 1,000 detected genes and non-neuronal cells with fewer than 500 detected genes. Doublets were identified using a modified version of the DoubletFinder algorithm[51] and removed when the doublet score was greater than 0.3.

**Chromatin accessibility (snATAC-seq) data pre-processing (UCSD).** Paired-end sequencing reads were demultiplexed and aligned to the mm10 reference genome using bwa[52]. After alignment, we converted paired-end reads into fragments and for each fragment, we checked the following attributes: (1) mapping quality score MAPQ; (2) whether two ends are appropriately paired according to the alignment flag information; and (3) fragment length. We only keep the properly paired fragments whose MAPQ (–min-mapq) is greater than 30 with fragment length less than 1,000 bp (–max-flen). Because the reads have been sorted based on the names, fragments belonging to the same cell (or barcode) are naturally grouped together, which allows for removing PCR duplicates. After alignment and filtration, we used Snaptools (https://github.com/r3fang/SnapTools; RRID: SCR_018097) to generate a snap-format file that contains metadata, cell-by-bin count matrices of various resolutions and cell-by-peak count matrices.

**Filtering cells by transcription start site enrichment and unique fragments.** The method for calculating enrichment at the transcription start site (TSS) was adapted from a previously described method[53]. TSS positions were obtained from the GENCODE database (RRID: SCR_014966). Briefly, Tn5-corrected insertions were aggregated ±2,000 bp relative (TSS strand-corrected) to each unique TSS genome-wide. Then, this profile was normalized to the mean accessibility ±1,900–2,000 bp from the TSS and smoothed every 11 bp. The maximum of the smoothed profile was taken as the TSS enrichment. We excluded any single cells that had fewer than 1,000 unique fragments or a TSS enrichment of less than 10 for any sample sets.

**Doublet removal.** After filtering out low-quality nuclei, we used Scrublet (RRID: SCR_018098)[54] to remove potential doublets for every sample set. Cell-by-peak count matrices were used as input, with default parameters.

**Preprocessing of the DNA methylation (snmC-seq2) data (Salk Institute)**
**Mapping and feature count pipeline for snmC-seq2.** We implemented a versatile mapping pipeline (cemba-data.rtfd.io) for all the single-cell methylome-based technologies developed by our group[9,14,30]. The main steps of this pipeline included: (1) demultiplexing FASTQ files into single-cell files; (2) reads-level quality control; (3) mapping; (4) BAM file processing and quality control; and (5) final molecular profile generation. The details of the five steps for snmC-seq2 have been

previously described[14]. We mapped all the reads onto the mouse mm10 genome. After mapping, we calculated the methyl-cytosine counts and the total cytosine counts in two sets of genome regions for each cell: the non-overlapping 100-kb bins tiling the mm10 genome, which was used for methylation-based clustering analysis, and gene body regions ± 2 kb, which was used for cluster annotation and cross-modality integration.

**Quality control and cell filtering.** We filtered the cells based on five quality metrics: (1) the rate of bisulfite non-conversion as estimated by the rate of methylation at CCC positions (mCCC) < 0.03 (the mCCC rate reliably estimates the upper bound of the bisulfite non-conversion rate[9]); (2) the overall mCG rate > 0.5; (3) the overall mCH rate < 0.2; (4) the total final reads (combining R1 and R2) > 500,000; and (5) the total mapping rate (using Bismark[55]) > 0.5.

**Preprocessing and clustering.** The clustering steps of snmC-seq2 data were previously described[30]. In brief, we calculated the posterior mCH and mCG rate based on beta-binomial distribution for the non-overlapping 100-kb bins matrix. We then selected the top 3,000 highly variable features to perform principal components analysis (PCA) and find dominant principal components for mCH and mCG separately. We concatenate principal components from both methylation types together to construct a $k$-nearest neighbour (KNN) graph, and ran the Leiden community detection algorithm[56] repeatedly to get the consensus clustering results. The stopping criteria of clustering considered the number of marker genes, the accuracy of the reproducible supervised model based on the cluster assignments and the minimum cluster size. We performed the clustering in two iterations to get the major types and fine-grained types for comparison with other modalities in further integration.

## Computational analysis
**Estimation of library size.** For estimate of library size, see Extended Data Fig. 1e. For each dataset, we estimated the total library size, that is, the number of unique RNA or DNA fragments ($F$), based on the rate of duplicate sequence reads. The number of unique mapped reads is $N_{unique} = F(1 - \text{Bin}[0|S, 1/F]) = F[1 - (1 - 1/F)^S]$, where $S$ is the total number of sequenced reads. Using this equation, we numerically solved for $F$ using the median values of $S$, $N_{unique}$.

## Transcriptome analysis
**Clustering individual datasets.** For transcriptomic analysis, see Fig. 1. Clustering for each scRNA-seq and snRNA-seq dataset was performed independently using the R package scrattch.hicat[3] (RRID: SCR_018099; available at https://github.com/AllenInstitute/scrattch.hicat). This package supports iterative clustering by making successively finer splits while ensuring all pairs of clusters, even at the finest level, are separable by stringent differential gene expression criteria[3]. For the scRNA 10x datasets, we used q1.th = 0.4, q.diff.th = 0.7, de.score.th = 150 and min.cells = 10. For the snRNA 10x datasets, we used q1.th = 0.3, q.diff.th = 0.7, de.score.th = 100 and min.cells = 10. For the scRNA SMART datasets, we used q1.th = 0.5, q.diff.th = 0.7, de.score.th = 150 and min.cells = 4. For the snRNA SMART dataset, we used q1.th = 0.4, q.diff.th = 0.7, de.score.th = 100 and min.cells = 4. We further performed consensus clustering by repeating iterative clustering on a subsample of 80% of cells, resampled 100 times, followed by final clustering based on the co-clustering probability matrix. Using this procedure, we could fine-tune cluster boundaries as well as assess cluster uncertainty.

Next, we removed low-quality and doublet-driven clusters. We performed differential gene expression analysis between every pair of clusters within each subclass. If any cluster had ≤2 upregulated genes (fold change > 2, FDR < 0.01, with additional dataset-specific parameters listed in the previous paragraph) than another cluster, and had a substantially lower average number of detected genes per cell, we flagged the cluster as low quality and removed it from further analysis. Next,

if the upregulated genes between any two clusters within a subclass were predominantly marker genes for a different subclass, and one of the clusters had a significantly higher average of genes detected per cell and UMI count, we flagged the cluster as a potential doublet cluster and removed it from further analysis. These criteria led to the exclusion of 8.3% of all cells, the vast majority of which came from the two 10x v3 datasets (scRNA 10x v3 A and snRNA 10x v3 B). While the 10x v3 platform boosts the gene detection for good cells, it does the same to damaged cells or debris, leading to an increased number of clusters that were excluded for these datasets.

**Joint clustering of multiple transcriptome datasets.** To provide a consensus cell-type taxonomy across all transcriptomic datasets, we developed an integrative clustering analysis across multiple data modalities. This procedure is available via the harmonize function of the scrattch.hicat package. Unlike Seurat/CCA[57], which aims to find aligned common reduced dimensions across multiple datasets, this method directly builds a common adjacency graph using the cells from all datasets, and then applies the Louvain community detection algorithm[58]. We extended the cluster merging algorithm in the scrattch.hicat package to ensure that all clusters can be separated by conserved differentially expressed genes across platforms. The i_harmonize function, similar to the iter_clust function in the single-dataset clustering pipeline, applies integrative clustering across datasets iteratively while ensuring that all the clusters at each iteration are separable by conserved differentially expressed genes.

To build a common adjacency matrix incorporating samples from all the datasets, we first chose a subset of datasets that we used as 'reference datasets'. For this study, we used the 10x v2 single-cell dataset from the Allen Institute (scRNA 10x v2 A) and the 10x v3 single-nucleus dataset from the Broad Institute (snRNA 10x v3 B) as the reference datasets, as both are large datasets that provide comprehensive cell-type coverage and relatively sensitive gene detection.

The key steps of the pipeline are outlined: (1) perform single-dataset clustering (Methods described above). (2) Select the anchor cells for each reference dataset. For each reference dataset (scRNA 10x v2 A or snRNA 10x v3 B), we randomly sampled up to $\max\left(100, \frac{5,000}{\text{no.ofclusters}}\right)$ anchor cells per cluster to normalize coverage for each cell type. This is the only step that uses the dataset-specific clustering information. (3) Select highly variable genes. Highly variable gene selection and dimensionality reduction by PCA were performed using the scrattch.hicat package. We removed principal components with a Pearson correlation coefficient of more than 0.7 with $\log_2(N_{\text{genes}})$. This step was implemented to mitigate the effect of cell or nucleus quality on gene expression variability, and to select only biologically relevant principal components. For each remaining principal component, $Z$-scores were calculated for gene loadings. The top 100 genes with an absolute $Z$-score greater than 2 were selected as highly variable genes. The highly variable genes from each reference dataset were combined. (4) Compute KNNs. For each cell in each query dataset, we computed its KNNs ($k=15$) among anchor cells in each reference dataset (scRNA 10x v2 A or snRNA 10x v3 B), based on the highly variable genes selected above. The RANN package was used to compute KNN based on the Euclidean distance when the query and reference dataset was the same. To compute nearest neighbours across datasets, we used correlation as a similarity metric. (5) Compute the Jaccard similarity. For every pair of cells from all datasets, we computed their Jaccard similarity, defined as the ratio of the number of shared KNNs (among all anchors cells from all the reference datasets) divided by the number of combined KNNs. (6) Perform Louvain clustering. (7) Merge clusters. To ensure that every pair of clusters are separable by conserved differentially expressed genes across all datasets, for each cluster, we first identified the top three most similar clusters. For each pair of such closely related clusters, we computed the differentially expressed genes in each dataset. We focus on the conserved differentially expressed genes that are

significant in at least one dataset, while also having more than twofold change in the same direction in all but one datasets. We then computed the overall statistical significance based on such conserved differentially expressed genes for each dataset independently. If any of the datasets passed our differentially expressed gene criteria described in the 'clustering' section, the pair of clusters remained separated; otherwise they were merged. Differentially expressed genes were recomputed for the merged clusters, and the process was repeated until all clusters were separable by the conserved differentially expressed genes criteria. If one cluster had fewer than the minimal number of cells in a dataset (4 cells for SMART-Seq and 10 cells for 10x), then this dataset was not used for differentially expressed gene computation for all pairs involving the given cluster. This step allows detection of unique clusters absent in some platforms. (8) Iterative clustering. Repeat steps 1–6 for cells within each cluster to gain finer-resolution clusters until no more clusters can be found. (9) Final compilation and merging of clusters. Concatenate all the clusters from all of the iterative clustering steps and perform the final merging as described in step 6.

**Marker gene selection.** For each pair of clusters, we computed the conserved differentially expressed genes, that is, those which are significantly differentially expressed in at least one dataset, with a twofold or more change in expression in the same direction among 70% of datasets. To allow computation of differentially expressed genes involving cell types only present in a subset of datasets, only the datasets with enough cells (based on min.cells parameter) for both cell types under comparison were used. We selected the top 50 genes in each direction. After pooling genes from all pairwise comparisons, we identified a total of 3,792 marker genes (Supplementary Table 6).

**Imputation.** To facilitate direct comparison, we projected gene expression of all datasets to the space of a given reference dataset. To do that, we leveraged the KNN matrices computed during the iterative joint clustering step to adjust the expression values for systematic differences between datasets. During each iteration of the joint clustering, for cells in each dataset, we used the average gene expression of their KNNs among the anchor cells from the reference dataset as the adjusted expression in the reference space. At the top-level clustering, we imputed the expression for all genes. For each subsequent iteration, we only imputed the expression of the high-variance genes and the conserved differentially expressed genes for the clusters defined in that iteration. We used this iterative approach for imputation because the nearest neighbours based on the genes chosen at the top level may not reflect the distinction between the finer types, and the imputed values for the differentially expressed genes that define the finer types consequently are not accurate based on these nearest neighbours. Therefore, we deferred imputation of the differentially expressed genes between the finer types to the iteration when these types were defined. This method is provided in the impute_knn_global function in the scrattch.hicat package[3]. We imputed the gene expression matrix for both reference datasets used in the integrative clustering.

**Building a cell-type taxonomy tree.** We first computed the average adjusted expression of marker genes for each cluster. This average was computed using each of the two reference datasets (scRNA 10x v2 A and snRNA 10x v3 B). Then, the two matrices were concatenated. We constructed a hierarchy (tree) using the build_dend_harmonize function in the scrattch.hicat package[3].

**Dimensionality reduction by UMAP.** We performed PCA based on imputed gene expression matrices of 3,792 marker genes using the 10x single-nucleus dataset from the Broad Institute as the reference, and selected the top 50 principal components (93% variance explained). We removed principal components with Pearson correlation coefficient > 0.6 with the $\log_2(N_{\text{genes}})$ to reduce bias related to the number of

detected genes. UMAP was used to embed the cells in two dimensions with parameters nn.neighbours = 25 and md = 0.3 (ref. [40]).

## MetaNeighbor analysis

For the MetaNeighbor analysis, see Fig. 1g. To quantify replicability of clusters across the seven transcriptomic datasets, we applied a modified version of unsupervised MetaNeighbor (RRID: SCR_016727)[24]. MetaNeighbor uses a neighbour voting algorithm and a cross-dataset validation scheme to quantify cluster similarity across multiple datasets. It requires a set of unnormalized datasets, a set of cluster labels and a set of highly variable genes. We used the raw count data for all cells passing the quality control criteria for the seven single-cell transcriptomic datasets, as well as the labels obtained through independent clustering (Supplementary Table 5). We used the variableGenes procedure in MetaNeighbor to select 310 highly variable genes that were detected as highly variable across all datasets.

We defined replicable clusters in a two-step procedure: first, we quantified the similarity between clusters across datasets, then we extracted groups of highly similar clusters, or 'meta-clusters'. We used the MetaNeighborUS function to obtain an initial similarity matrix between clusters. By default, cluster similarity is quantified as a one-vs-all AUROC: given a training cluster (in one dataset), we asked how similar cells from a test cluster (in another dataset) were to training cells, compared to all other cells in the test dataset. To make cluster matching more stringent, we transformed the one-vs-all AUROC matrix into a one-vs-best AUROC matrix: instead of ranking test cells among all cells from the test dataset, we only compared them to cells from the best-matching cluster. This modification ensured that only the best match had an AUROC > 0.5, facilitating identification of reciprocal best hits. For interpretability and computational efficiency, we adopted the following convention: the best-matching AUROC of a cluster was obtained by comparing it to the second best-matching cluster, the second best AUROC of a cluster was obtained by computing 1 − AUROC of the best-matching cluster, and all other clusters obtained an AUROC of 0, as we were only interested in finding best matches. To extract meta-clusters, we interpreted the one-vs-best AUROC as a graph where nodes are clusters and edges connect nodes if they are reciprocal best hits. We define meta-clusters as connected components in this graph. We can obtain more robust meta-clusters by requiring that best hits exceed some AUROC threshold. In practice, we noted that one-vs-best AUROC > 0.7 offered a good balance between the number of meta-clusters and reproducibility strength.

For scalability, we modified MetaNeighbor in the following ways. In the MetaNeighborUS function, we removed the rank standardization of the cell–cell similarity network (by setting the parameter fast_version to TRUE) and the node degree normalization of the neighbour voting, enabling analytical simplifications of the neighbour voting procedure. The variableGenes procedure was applied to a random subset of 50,000 cells for datasets exceeding that size.

MetaNeighbor analysis further allowed us to examine the consistency of computational clustering procedures (Fig. 1h). We ran three widely used single-cell analysis packages[25-27] to generate a fine-grained clustering of each dataset. These cluster analyses were not optimized or manually curated; instead, we used 'off-the-shelf' computational procedures to test the robustness of the results from a relatively straightforward and automated analysis. These clusters are thus expected to be less biologically meaningful and robust than more customized procedures, such as our reference clustering that incorporates analysis of differential expression to validate the biological reality of cell types. Using the three off-the-shelf cluster analyses, we created a sequence of increasingly coarse-grained clusterings by iteratively merging pairs of clusters chosen to maximize the consistency across computational methods (ARI-merging). Finally, at each level of resolution, we used MetaNeighbor to calculate the number of clusters that were highly replicable (AUROC > 0.7) across datasets. The result of this analysis

showed that fine partitions of the data with more than 30–50 clusters have limited replicability.

**Cluster analysis for snmC-seq2.** For cluster analysis for snmC-seq2, see Extended Data Fig. 6a, b. We concatenated principal components from both methylation types (CG and CH) together, and used these to construct a KNN graph followed by Leiden community detection[56]. We repeated the cluster analysis several times to get consensus clustering results. The stopping criteria of clustering considered the number of marker genes, the accuracy of the reproducible supervised model based on the cluster assignments and the minimum cluster size. We performed the clustering in two iterations to get major types and fine-grained cell types for comparison with other modalities in further integration.

Two-dimensional embedding using t-distributed stochastic neighbour embedding[59] (t-SNE; perplexity = 30) was calculated based on the top principal components using the implementation from the scanpy package[60].

**Cluster analysis for snATAC-seq.** For cluster analysis for snATAC-seq, see Extended Data Fig. 6c, d. We used the snapATAC pipeline[48] to identify cell clusters with binarized cell-by-bin matrix in 5-kb resolution as the input. Cell clusters were annotated to cell type by checking chromatin accessibility along the body of marker genes. Then, another round of clustering was performed on medial ganglionic eminence (MGE)-derived and caudal ganglionic eminence (CGE)-derived inhibitory GABAergic interneurons, to identify sub-cell types.

## Multimodality integration

For multimodality integration, see Fig. 2.

**Computational data integration with LIGER.** We used LIGER (RRID: SCR_018100) to integrate the single-cell transcriptomic and epigenomic data as previously described[29], with one modification. We used the optimizeALS function in the LIGER package to perform joint factorization on all datasets except methylation (seven RNA datasets and one ATAC dataset) to infer shared ($W$) and dataset-specific ($V_i$) metagene factors and cell factor loadings ($H_i$). We then used the resulting $W$ to calculate cell factor loadings ($H_i$) for the methylation data using the solveNNLS function in the LIGER package. We found that this strategy yielded better integration than jointly factorizing all eight datasets, possibly because the inverse relationship and massive size imbalance of datasets between methylation and all other datasets complicates the learning of shared metagenes. Our analysis used only the cells annotated by each data-generating group as passing quality control. We did not perform any data imputation or smoothing, but simply normalized and scaled the raw cell-by-gene count matrices from each dataset using the normalize and scaleNotCenter functions in the LIGER package. We next used the quantileAlignSNF function with default settings to perform quantile normalization of cell factor matrices ($H_i$) from all eight datasets. Finally, we performed Louvain clustering on the normalized cell factor matrices ($H_i$) to obtain joint clusters. We performed two rounds of integration and joint clustering; in the first round, we separately integrated all neurons across datasets and all glia across datasets. We then performed a second round of integration and clustering separately for each of the four neuronal subclasses: excitatory IT neurons, excitatory non-IT neurons, MGE interneurons and CGE interneurons. We used $k = 40$ factors for the non-neuron analysis, $k = 30$ for the first-round neuron analysis and $k = 20$ for all of the second-round analyses.

**Computational integration with SingleCellFusion.** SingleCellFusion[30] is designed to robustly integrate DNA methylation, ATAC-seq and/or RNA-seq data. We applied SingleCellFusion iteratively to integrate all neurons from eight datasets (Supplementary Table 1) and jointly call

cell clusters. To integrate both the broad and fine-grained cell types, we performed three rounds of integration. For every cell cluster generated in the previous round, it was further split into smaller clusters by reapplying SCF on cells in that cluster only. In the first round, we ran SCF on all neurons from 8 datasets and got 10 broad neuronal clusters. Rounds two and three generates 29 clusters and 56 more fine-grained clusters, respectively (Supplementary Table 3).

The procedure comprised four major steps: preprocessing, within-modality smoothing, cross-modality imputation, and clustering and visualization. (1) For the preprocessing step, we defined a gene-by-cell feature matrix for each dataset. Droplet-based RNA-seq features (10x) were $\log_{10}(CPM + 1)$ normalized; full-length RNA-seq (SMART-Seq) features were $\log_{10}(TPM + 1)$ normalized. snATAC-seq data were represented by read counts within the gene body, normalized by $\log_{10}(RPM + 1)$, where CPM stands for counts per million reads mapped (counts normalized), TPM stands for transcripts per million reads mapped (length normalized) and RPM stands for reads per million reads mapped (length normalized), respectively. DNA methylation data are represented by the mean gene body mCH level, normalized by the global (genome-wide) mean mCH level for each cell. For each dataset, we only used high-quality cells (passed quality control) and highly variable genes ($n = 4,000-6,300$) for further analysis. To select highly variable genes, for RNA-seq and ATAC-seq datasets, we first removed genes that were expressed in fewer than 1% of cells. We then divided the remaining genes into 10 bins according to their mean expression across cells (CPM). For each bin, except for the one with the most expression, we selected the top 30% of genes with the most expression dispersion (variance/mean) as the highly variable genes. For the DNA methylation dataset, we first selected genes that had more than 20 cytosine coverage in more than 95% of cells, then divided the remaining genes into 10 bins according to their mean normalized mCH level – raw mCH level normalized by the global mCH for each cell. For each bin, we selected the top 30% of genes with the most variance as the highly variable genes. (2) For the within-modality smoothing step, to reduce the sparsity and noise of feature matrices, we shared information among cells with similar profiles using data diffusion. The procedure is adapted from ref. [61] and described in detail in ref. [30]. Here we exactly followed ref. [30] with $[ndim = 50, k = 30, ka = 5]$ for all datasets, and $[P = 0.7]$ for RNA-seq datasets, $[P = 0.9]$ for the DNA methylation dataset and $[P = 0.1]$ for the ATAC-seq dataset. (3) For the cross-modality imputation by restricted k-partners (RKP) step, to integrate all eight datasets, we impute the scRNA 10x v2 A gene features for cells in all seven other datasets. The imputation was done pairwise between the scRNA 10x v2 A dataset and each of the other datasets. For each pairwise imputation, we followed the procedure described in ref. [30] with 20 RKP and relaxation parameter 3 $[k = 20, z = 3]$. Instead of using Euclidean distance in a low-dimensional space, we used the (flipped) Spearman correlation coefficient across genes that were highly variable in both datasets as the distance metric between cells in two different modalities. (4) For the clustering and visualization step, we started from a cell-by-feature matrix, where cells included all cells from eight datasets and features were highly variable genes of the scRNA 10x v2 A dataset. We reduced the dimensionality of features into the top 50 principal components. Next, we performed UMAP embedding[40] on the principal component matrix ($n\_neighbours = 60$, $min\_dist = 0.5$). Finally, we performed Leiden clustering on the KNN graph (symmetrized, unweighted) generated from the final principal component matrix (Euclidean distance, $k = 30$, resolution = 0.1).

For Extended Data Fig. 7e, we created the embedding of the cluster centroids using the imputed scRNA 10x v2 A gene features ($\log_{10}(CPM + 1)$) for all cells from the eight different datasets generated from SingleCellFusion integration. Clusters are defined by individual dataset clusterings and by the joint clustering with SingleCellFusion. Cluster centroids were calculated by the mean imputed scRNA 10x v2 A gene profiles across cells. After getting a gene-by-cluster matrix, we applied PCA to reduce to 50 feature dimensions, followed by applying a UMAP embedding with $min\_dist = 0.7$ and $n\_neighbours = 10$.

For Fig. 2e, to compare molecular signals across data modalities, all signals were normalized to $[0, 1]$. This was achieved by first getting molecular signals by dataset-specific normalization (step 1), followed by a linear transformation (step 2). In step 1, for SMART-Seq datasets, we show $\log_{10}(TPM + 1)$; for 10x RNA-seq datasets, we show $\log_{10}(CPM + 1)$; for the ATAC-seq dataset, we show $\log_{10}(RPM + 1)$ normalized gene body counts; and for DNA methylation, we show gene body mCH normalized by global mCH level of each cell. For step 2, we applied a linear transformation to map the range of the signal to $[0, 1]$. For datasets other than DNA methylation, we applied the following formula:

$$x_{\text{normalized}} = \frac{x - x_{\min}}{x_{\max} - x_{\min}}$$

where $x$ is the dataset-specific gene-level signal for a cell, $x_{\min}$ and $x_{\max}$ are defined as the bottom two percentile and the top two percentile of $x$ across all cells, respectively. For the DNA methylation dataset, we applied the following formula:

$$x_{\text{normalized}} = 1 - \frac{x - x_{\min}}{x_{\max} - x_{\min}},$$

with which signals were still mapped to $[0, 1]$ but flipped—a high signal on the plot means a low level of DNA methylation. We did this to align DNA methylation signals with gene expression and open chromatin signals, because DNA methylation is a repressive marker of gene expression and negatively correlates with it. In these formulas, $x_{\min}$ and $x_{\max}$ are defined as the bottom 2 percentile and the top 50 percentile of $x$ across all cells, respectively.

For Fig. 2d, for each gene, cell-level signals were normalized the same way as described in step 1 of Fig. 2e. Cluster-level signals are the mean cell-level signals across cells in clusters. After getting gene-by-cluster matrices this way, for non-DNA methylation datasets, the matrices were further normalized by the maximum of each cluster (column); for DNA methylation datasets, no further normalization was done, as they were already normalized by cell.

For Extended Data Fig. 7g, h, the heat maps show pairwise Spearman correlation coefficients between the centroids of cells from each cell type (SingleCellFusion) and each dataset, using the gene expression levels ($\log_{10}(CPM + 1)$; measured or imputed by SingleCellFusion) of the scRNA 10x v2 A dataset as features. Centroid-level profiles were computed as the average of cell-level profiles across cells from the same cell type and the same dataset. The row and column orderings were the same, generated by a hierarchical clustering on the above-defined centroid-level features with average linkage and Euclidean distance. Extended Data Fig. 7g shows the correlations between broad-level joint clusterings (10 subclasses; SingleCellFusion L0) (Supplementary Table 8); Extended Data Fig. 7h shows those between fine-level joint clusterings (56 clusters in total; not all are shown; SingleCellFusion L2) (Supplementary Table 8) for four example broad-level subclasses (MGE, CGE, L2/3 IT and L4/5 IT).

For the agreement metric in Extended Data Fig. 7c, we calculated dataset agreement metrics as described in the LIGER paper[29]. In brief, we performed dimensionality reduction using either non-negative matrix factorization (NMF; for LIGER) or PCA (for SingleCellFusion) and built a KNN graph for each individual dataset. Then, we built a KNN graph using the joint latent space from either LIGER or SingleCellFusion and calculated what fraction of the nearest neighbours from individual datasets were still nearest neighbours in the joint space. This metric assesses how well the joint latent space preserves the structure of each individual dataset. An agreement metric close to 0 indicates poor preservation of structure from individual datasets, while an agreement metric close to 1 ideally preserves the structure.

For the alignment metric in Extended Data Fig. 7d, we calculated dataset alignment metrics as described in the LIGER[29] and Seurat[57] papers, except that we first downsampled cells so that the cluster proportions and the total number of cells were identical across all datasets. Next, we built a KNN graph using the joint latent space from either LIGER or SingleCellFusion and calculated what fraction of the nearest neighbours around each point came from each dataset. We then normalized the metric to be between 0 (no alignment) and 1 (perfect mixing of datasets). This metric assesses how well the joint latent space aligns the datasets. Note that maximizing alignment and maximizing agreement are competing objectives. For example, it is possible to trivially maximize alignment by randomly mixing cells from all datasets according to a spherical Gaussian distribution; conversely, one could trivially maximize agreement by simply assigning non-overlapping latent representations to all datasets. However, methods must balance these competing objectives to score highly on both alignment and agreement metrics.

For Extended Data Fig. 7f, to get cluster-level gene signals, we first got normalized cell-level signals the same way as step 1 of Fig. 2e, followed by taking the mean cell-level signals across cells in clusters.

### Analysis of enhancers

**Epigenome cluster level.** On the basis of the cell–cell integration in Fig. 2, to have enough whole-genome coverage of each cell type, we further merged the co-clusters into a higher level to increase the coverage of each cluster, which we termed as the epigenome cluster level.

**DMR calling.** For DMR calling in the snmC-seq2 data, we merged single-cell ALLC files into the pseudo-bulk level for each cluster, and then used the methylpy[62] DMRfind function to calculate mCG DMRs across all clusters. The base call of each paired CpG site was added up before analysis. In brief, the methylpy function used a permutation-based root mean square test of goodness-of-fit to identify differentially methylated sites simultaneously across all samples, and then merged the differentially methylated sites within 250 bp into DMRs. Hypo-DMRs and hyper-DMRs were then assigned to each sample by examining the residue of observed counts from the expected counts. We also filtered the DMRs by requiring that the maximum difference of mCG rate between clusters was larger than 0.3.

**snATAC peak calling.** We called peaks according to the ENCODE ATAC-seq pipeline (https://www.encodeproject.org/atac-seq/). For every cell cluster, we combined all properly paired reads to generate a pseudo-bulk ATAC-seq dataset for individual biological replicates. In addition, we generated two pseudo-replicates, each of which included half of the reads from each biological replicate. We called peaks independently for each of these four datasets, as well as for a pool of the data from both biological replicates. Peak calling was performed on the Tn5-corrected single-base insertions using MACS2[63] (RRID: SCR_013291) with parameters: −shift −75 −extsize 150 −nomodel −call-summits −SPMR −keep-dup all −q 0.01. We extended peak summits by 250 bp on either side to a final width of 501 bp for merging and downstream analysis. To generate a list of reproducible peaks, we kept peaks that (1) were detected in the pooled dataset and overlapped 50% or more of the peak length with a peak in both individual biological replicates, or (2) were detected in the pooled dataset and overlapped 50% or more of the peak length with a peak in both pseudo-replicates.

To account for differences in performance of MACS2 based on read depth and/or the number of nuclei in individual clusters, we converted MACS2 peak scores ($-\log_{10}(q$ value)) to score per million (SPM)[64] and kept peaks with SPM > 2. We only kept reproducible peaks on chromosomes 1–19 and both sex chromosomes, and filtered ENCODE mm10 blacklist regions[65] (http://mitra.stanford.edu/kundaje/akundaje/release/blacklists/mm10-mouse/mm10.blacklist.bed.gz). Finally, since snATAC-seq data are relatively sparse, we selected only elements that

were identified as open chromatin in a significant fraction of the cells in each cluster. To this end, we defined a set of background regions, matching the number of peak regions for each cell type, by randomly selecting regions from the genome while excluding accessible sites from the ENCODE registry of cis-regulatory elements (https://screen.encodeproject.org/). We calculated the fraction of nuclei for each cell type that had ATAC fragments mapping to the background regions. Next, we fitted a zero-inflated beta model and empirically identified a significance threshold of FDR < 0.01 to filter potential false-positive peaks. Peak regions with FDR < 0.01 in at least one of the clusters were included in the downstream analysis.

We used 'bedtools intersect' with the '-wa -u' parameter to calculate DMR and ATAC peak overlaps[66] (RRID: SCR_006646).

**Saturation analysis.** To investigate the efficiency of regulatory element identification in terms of cell number in the epigenomic data, we did a saturation analysis using the two most abundant cell types: the L2/3 IT and the L6 CT excitatory neurons. The total reads assigned to these two cell types were comparable to bulk-seq. We subsampled a different number of cells without replacement in each cluster three times when we had enough cells, and used cells from each replicate separately when possible. In the last group, we used all of the cells for each cell type as a maximum reference. For methylome data, we called DMRs between L2/3 IT and L6 CT within each cell number group. Peaks were called for each cell-type group.

**REPTILE enhancer prediction.** We performed enhancer prediction using the REPTILE[35] algorithm. The REPTILE is a random-forest-based supervised method that incorporates different sources of epigenomic profiles with base-level DNA methylation data to learn and then distinguish the epigenomic signatures of enhancers and genomic background. We trained the model in a similar way as in previous studies[35,67] using CG methylation, chromatin accessibility of each epigenome cluster and mouse embryonic stem cells. The model was first trained on mouse embryonic stem cell data and then predicted a quantitative score that we termed enhancer score for the DMR of each cell type. The positives were 2-kb regions centred at the summits of the top 5,000 EP300 peaks in mouse embryonic stem cells. Negatives included randomly chosen 5,000 promoters and 30,000 2-kb genomic bins. The bins have no overlap with any positives or promoters[67]. Methylation and chromatin accessibility profiles in bigwig format for mouse embryonic stem cells were from the mouse ENCODE project[67]. The mCG rate bigwig file was generated from cell-type-merged ALLC files using the software ALLCOOLS (https://lhqing.github.io/ALLCools). For chromatin accessibility of each cell type, we merged all fragments from snATAC-seq cells that were assigned to this cell type in the integration analysis and used 'deeptools bamcoverage' to generate CPM-normalized bigwig files. The bin size for all bigwig files was 50 bp.

**Motif enrichment analysis.** We used 724 motif position weight matrices (PWMs) from the JASPAR 2020 CORE vertebrates database[68], where each motif was able to assign corresponding mouse transcription factor genes. For each set of REPTILE-predicted enhancers, we standardized the region length into centre ± 250 bp and used the FIMO tool from the MEME suite[69] to scan the motifs in each enhancer with log odds $P$ value < $10^{-6}$ as the threshold of the motif hit. To calculate motif enrichment, we used the adult non-neuronal mouse tissue DMRs[70] as background regions. We subtracted enhancers in the region set from the background and then scanned the motifs in background regions using the same approach. We then used Fisher's exact test to find motifs enriched in the region set and the Benjamini–Hochberg procedure to correct multiple tests. Transcription factors with significant motif enrichment were grouped by TFClass[71] classification. Genes within the same group shared very similar motifs.

## Cluster validation analysis
For cluster validation analysis, see Fig. 4.

**Downsampling analysis of cluster number.** For downsampling analysis of cluster number, see Fig. 4a–d.

Preprocessing was done in the same way as described in the section 'Computational integration with SingleCellFusion'. After preprocessing, we obtained a gene-by-cell feature matrix for each dataset. Only neuronal cells passing quality control (Supplementary Table 1) and highly variable genes for each dataset were included.

**Clustering.** Clustering (Fig. 4a) required three steps. We first reduced feature dimensions by PCA ($n = 50$). We then built a KNN graph ($k = 30$) between cells using the Euclidean distance in the principal component space. We finally applied the Leiden clustering algorithm with a fixed-resolution parameter ($r = 6$). For each dataset, we report the number of clusters as a function of the number of cells randomly downsampled from the full dataset. Error bars show the s.e.m. of ($n = 10$) rounds of downsampling.

**Clustering with within-dataset cross-validation.** This analysis (Fig. 4c) aimed to estimate the optimal number of clusters of a dataset, by testing which clustering granularity best preserves the gene-level features of cells. For a given dataset, a gene-by-cell matrix, we first randomly split gene features into two sets, for clustering and validation, respectively. To avoid any potential linkage, the split was done by separating chromosomes into two sets, such that genes from the same chromosomes were always in the same set. We then performed Leiden clustering (as described in the methods related to Fig. 4a) on all cells using the clustering feature set only, with different clustering resolutions. After clustering, every cell in the dataset received a cluster label. We next randomly separated those cells into training and testing sets. Using training-set cells, we trained a supervised model to predict the validation set gene features based on cluster assignments. The model was trained by minimizing the MSE between the model prediction and the data. This is equivalent to predicting the gene features of a cell as its cluster centroid. Finally, we evaluated the model performance by calculating the MSE for the cells in the test set. This is equivalent to estimating the mean squared distance between individual cells in the test set and the cluster centroid calculated using the training set. As a function of the number of clusters (by varying the resolution parameter in Leiden clustering), we observed a U-shaped curve of the MSE. The minimum point of the curve represents the most plausible clustering resolution. Applying this scheme to each dataset and different downsampling levels of cells, we report in Fig. 4c the number of clusters as a function of the number of cells, for each dataset. For robustness, random splitting of gene features was repeated $n = 5$ times; random splitting of cells was repeated $n = 5$ times with $k = 5$-fold cross-validation each time.

**Clustering with cross-dataset cross-validation.** Extending the within-dataset clustering cross-validation scheme used in Fig. 4c, we developed a cross-dataset cross-validation method (Fig. 4d), by combining the previously described within-dataset cross-validation method with a joint clustering method: SingleCellFusion. First, similar to within-dataset cross-validation, we randomly split gene features into clustering and validation sets for all datasets. We then generated integrated clusterings across data modalities by applying SingleCellFusion on all cells and on half of the gene features (the clustering feature set). After clustering, we estimated the MSE of clustering on the validation feature set as described above for each dataset on its own. Applying this scheme to different downsampling levels of cells, we report in Fig. 4d the number of clusters as a function of the number of cells from each dataset.

**Integrated analyses: trade-off between replicability and resolution and cluster consistency.** For integrated analyses, see Fig. 4e, f. We collected the clusters obtained with the four integrative clustering methods described previously (Conos, LIGER, RNA consensus clustering from Fig. 1 and SingleCellFusion), as well as the 'subclass' level from the independent clustering of the RNA datasets. Each integrative method returned clusters at two granularity levels. We named the coarser level of clustering L1 and the finer level of clustering L2 clusters. We focused our analyses on the neuron clusters of the transcriptomic data, as we wished to investigate the agreement of neuron cluster hierarchies.

To quantify replicability, we used the same modified version of Meta-Neighbor, the same datasets and the same variable genes as defined above (see 'MetaNeighbor analysis'). We used the one-vs-best AUROC to obtain cluster similarity scores, then computed an average AUROC score per integrated cluster (averaged over every pair of datasets in which the cluster is present). For every method, we reported the median AUROC across integrated clusters as the final reproducibility score. To quantify the overall similarity of the clustering results, we computed the adjusted Rand index. When necessary, we restricted the adjusted Rand index computation to the intersection of labelled cells (the intersection being recomputed for every pair of methods).

**Conos analysis.** To evaluate the extent to which different cell subpopulations were supported by different platforms, we assessed the difference in the ability to recover the corresponding cell with and without within-platform comparisons. The clustering of cells was performed using Conos[37] (Fig. 4g, h), using walktrap community detection to identify hierarchical cell populations. The stability of the hierarchical clusters was estimated as follows: 20 random cell subsampling rounds were performed, each sampling 95% of cells from each dataset, and repeating the walktrap hierarchical clustering procedure. For each node in the original walktrap tree, we evaluated stability as a minimum of specificity and sensitivity relative to the ensemble of subsampled trees by finding the best-matching subtree. To evaluate the ability to recover subpopulations based on cross-platform comparisons only, we removed within-platform edges (those connecting datasets generated by the same platform) in the joint graph (generated by Conos). In this way, the subpopulation was detected only based on mapping to the other platform. The modified approach facilitates grouping of cell populations that are common in the different platforms, as it removes the platform-specific information in the joint graph.

To assess the similarity of the expression profiles detected by different platforms for a given cell type (Fig. 4h), we used Jensen–Shannon divergence to assess the overall similarity of gene expression patterns between the four RNA-seq platforms (scRNA 10x v3 A, snRNA 10x v3 A, scRNA SMART and snRNA SMART). Specifically, 1,000 cells were sampled from each cell type for each platform. If the number of cells from a cell type was fewer than 1,000 cells, sampling with replacement was performed. Cell types that accounted for less than 1% (fewer than 300 cells) in any specific platform were omitted. The molecules detected for each gene were then aggregated across all sampled cells for each cell type in each platform. The counts were normalized by the total number of molecules for each cell type or platform, and Jensen–Shannon divergence was calculated.

## Reporting summary
Further information on research design is available in the Nature Research Reporting Summary linked to this paper.

## Data availability
The BICCN MOp data (RRID: SCR_015820) can be accessed via the NeMO archive (RRID: SCR_016152) at: https://assets.nemoarchive. org/dat-ch1nqb7. Visualization and analysis resources can be found

at: NeMO analytics (https://nemoanalytics.org/), Genome browser (https://brainome.ucsd.edu/BICCN_MOp) and Epiviz browser (https://epiviz.nemoanalytics.org/biccn_mop).

## Code availability

The codes used for data analysis: scratch.hicat (hierarchical, iterative clustering for analysis of transcriptomics) for RNA clustering (https://github.com/AllenInstitute/scratch.hicat); SnapTools for ATAC-seq analysis (https://github.com/r3fang/SnapTools); YAP (Yet Another Pipeline) and ALLCools for DNA methylation (snmC-seq2) mapping and cluster-level aggregation (https://github.com/lhqing/cemba_data; documentation: cemba-data.rtfd.io; https://lhqing.github.io/ALL-Cools); MetaNeighbor for cluster reproducibility analysis (https://github.com/gillislab/MetaNeighbor-BICCN); LIGER for multimodal integration, embedding and clustering (https://github.com/welch-lab/liger); SingleCellFusion for multimodal integration, embedding and clustering (https://github.com/mukamel-lab/SingleCellFusion); Conos for cluster reproducibility analysis (https://github.com/kharchenkolab/conos); STAR v2.5.3 for RNA-seq alignment[49]; and Bismark for DNA methylation (snmC-seq2) alignment[55].

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

**Acknowledgements** We are grateful to A. Bandrowski and Y. Yao for their insightful comments. This work was funded by the NIH BRAIN Initiative (U19MH114830 to H.Z., U19MH121282 to J.R.E., U19MH114821 to Z.J.H., R24MH114788 to O.R.W., U24MH114827 to M.H., R24MH114815 to R.H. and O.R.W., and NIH NIDCD DC013817 to R.H.), the Hearing Restoration project Hearing Health Foundation (R.H.) and the NIH NIGMS (GM114267 to H.C.B.).

**Author contributions** A.R., A.T., B.T., C.R., C.R.V., D.B., D.M., E.L.D., E.Z.M., H.T., H.Z., J. Goldy, J.S., K.C., K.L., K. Smith, M.K., M.T., N.D., N.M.N., O.F., T.C., T.N.N. and T.P. contributed to RNA data generation. A.B., A.C.R., A.I.A., A.P.-D., C.L., H.L., J.D.L., J.K.O., J.R.E., J.R.N., M.M.B., S.-Y.N. and Y.E.L. contributed to DNA methylation (snmC-seq2) data generation. A.P.-D., B.R., J.D.L., J.K.O., M.M.B., S.P., X.H., X.W. and Y.E.L. contributed to snATAC data generation. A.M., B.R.H., C.C., C.v.V., E.A.M., F.X., H.C., H.C.B., J.C., J. Goldy, J.K., J.O., M.G., M.H., O.R.W., R.F., R.H., R.S.A., S.A.A., S.-Y.N., V.F., W.I.D. and Z.Y. contributed to data archive/infrastructure. A.R., A.S.B., B.T., D.R., E.A.M., E.D.V., E.P., E.Z.M., F.X., H.L., H.R.d.B., H.Z., J.D.W., J. Goldy, J. Gillis, J.O., K. Smith, K. Street, K.V.d.B., L.P., M.C., O.F., O.P., P.V.K., Q.H., R.F., S.D., S.F., S.-Y.N., T.B., V.N., V.S., W.I.D., Y.E.L. and Z.Y. contributed to data analysis. A.R., B.R., B.T., C.L., E.A.M., E.D.V., E.Z.M., F.X., H.L., H.Z., J.D.W., J. Gillis, J.N., M.C., M.M.B., P.V.K., Q.H., R.F., S.F., T.B., Y.E.L. and Z.Y. contributed to data interpretation. A.S.B., E.A.M., F.X., H.L., H.Z., J.D.W., J. Gillis, L.P., M.C., Q.H., S.F., Z.J.H. and Z.Y. contributed to writing the manuscript.

**Competing interests** B.R. is a shareholder of Arima Genomics, Inc. P.V.K. serves on the Scientific Advisory Board to Celsius Therapeutics, Inc. A.R. is an equity holder and founder of Celsius Therapeutics, an equity holder in Immunitas, and a Scientific Advisory Board member to Syros Pharmaceuticals, Neogene Therapeutics, Asimov and Thermo Fisher Scientific.

**Additional information**
**Correspondence and requests for materials** should be addressed to H.Z. or E.A.M.

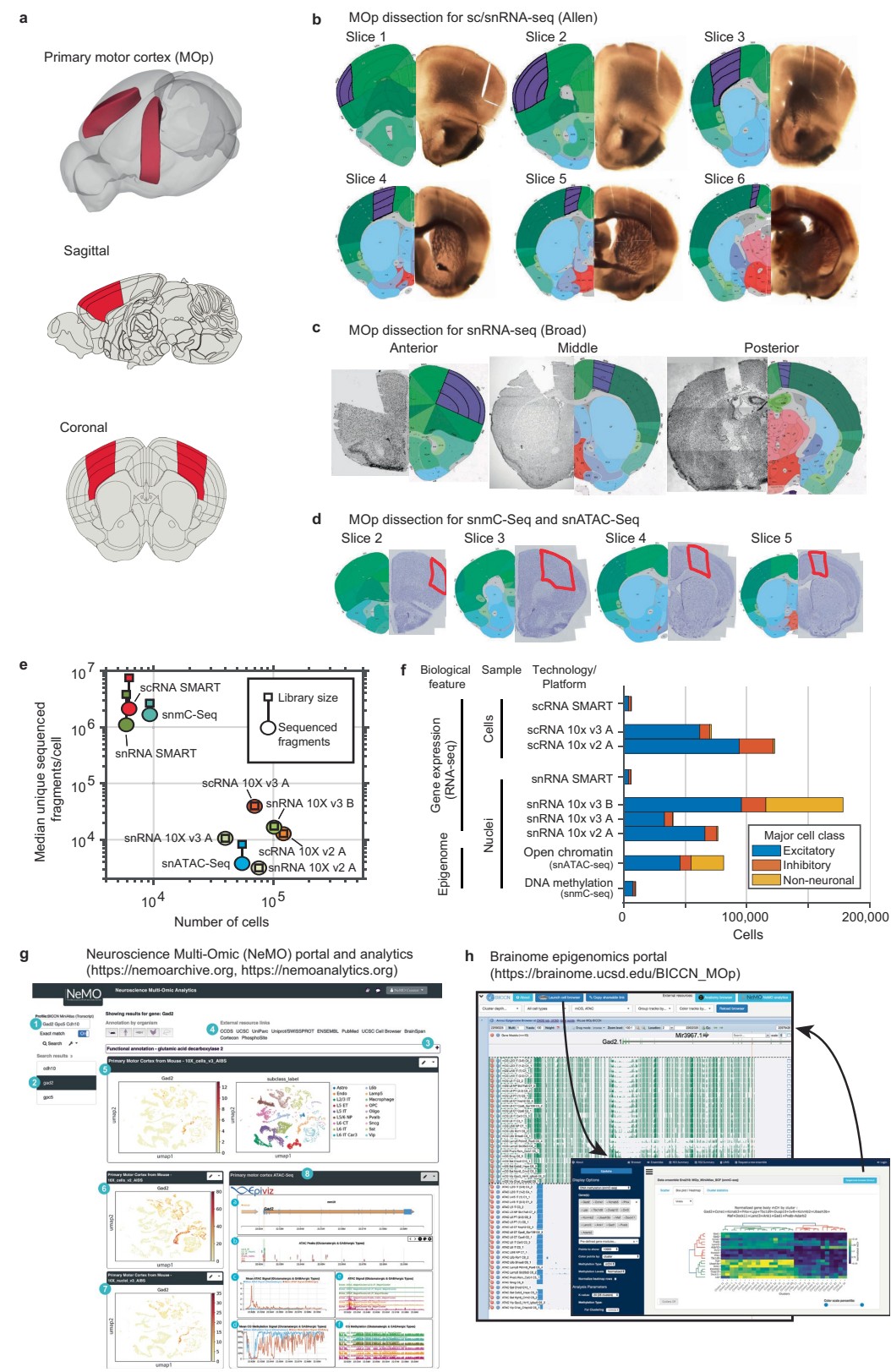

**Extended Data Fig. 1** | See next page for caption.

**Extended Data Fig. 1 | A multimodal molecular cell-type atlas of the MOp.**
**a**, Anatomical location of the mouse MOp in the Allen Mouse Brain Common Coordinate Framework (CCFv3) in 3D and in representative sagittal and coronal sections. **b**–**d**, Documentation of MOp samples collected at the Allen Institute (**b**), the Broad Institute (**c**) and the Salk Institute (**d**). Each panel shows a diagram of coronal brain slices and dissected regions for transcriptomic (scRNA-seq and snRNA-seq) and epigenomic (snATAC and snmC-seq2) data samples based on the Allen Mouse Brain Common Coordinate Framework (CCF). Nissl-stained images in **d** show the posterior face of tissue slices (600 μm thickness).
**e**, Number of cells and median number of unique sequenced DNA or RNA fragments per cell in each of the nine single-cell transcriptomic and epigenomic datasets. The squares show the extrapolated total library size based on the sequence duplication rate. **f**, Number of cells in each of the major cell classes (glutamatergic excitatory, GABAergic inhibitory neurons and non-neurons) of each dataset. Differences in cell-type sampling strategy, including the use of cell sorting to enrich neurons, affect the relative number of neurons and non-neuronal cells. Datasets include cells from the following numbers of mice (Supplementary Table 1): scRNA SMART: $n = 28$ male, 17 female; scRNA 10x v3 A: $n = 3$ male, 3 female; scRNA 10x v2 A: $n = 3$ male; snRNA SMART: $n = 8$ male, 2 female; snRNA 10x v3 B: $n = 5$ male, 6 female; snRNA 10x v2: $n = 2$ male, 1 female; snRNA 10x v3 A: $n = 1$ female; snmC-seq2 and snATAC-seq: $n = 2$ replicates, each pooled from 6 to 30 male mice. **g**, NeMO Analytics (nemoanalytics.org) visualization and analysis environment for the BICCN mouse molecular mini-atlas. Screenshot of NeMO Analytics showing multi-omic results for glutamate decarboxylase 2 (*Gad2*), a marker gene in inhibitory neurons. The web portal has the following features: (1) search box for gene names; (2) indicator of the gene viewed; (3) expandable species-specific functional annotation; (4) link-outs to additional resources for the selected gene; (5–7) interactive visualizations of each BICCN dataset, displayed in a 'standalone' box showing gene expression and cell clustering on integrated UMAP coordinates. Additional data exploration options for each of the datasets are available via the drop-down menu at the upper right corner of the NeMO Analytics dataset titles. (8) An embedded Epiviz interactive workspace to visualize scATAC-seq and sncMethyl-seq datasets in a linear browser view (8a), here showing the average ATAC and % CG methylation at the *Gad2* locus (8c, 8d) as well as in each major cluster of glutamatergic and GABAergic neurons (8b, 8e, 8f). Epigenomic data are also available at http://epiviz.nemoanalytics.org/biccn_mop, and instructions for setting up and extending the Epiviz workspaces are available at http://github.com/epiviz/miniatlas. **h**, Brainome epigenomics portal (https://brainome.ucsd.edu/BICCN_MOp). The portal shows single-base resolution epigenomic and transcriptomic data (snmC-seq2, snATAC-seq, scRNA-seq and snRNA-seq) using the AnnoJ browser. Drop-down menus allow the user to select groups of cells (for example, excitatory, inhibitory and MGE-derived, among others), modalities (mCG, mCA, ATAC, scRNA, snRNA and enhancers) and display options. A Cell Browser allows visualization of scatter plots and heat maps of groups of genes across data modalities.

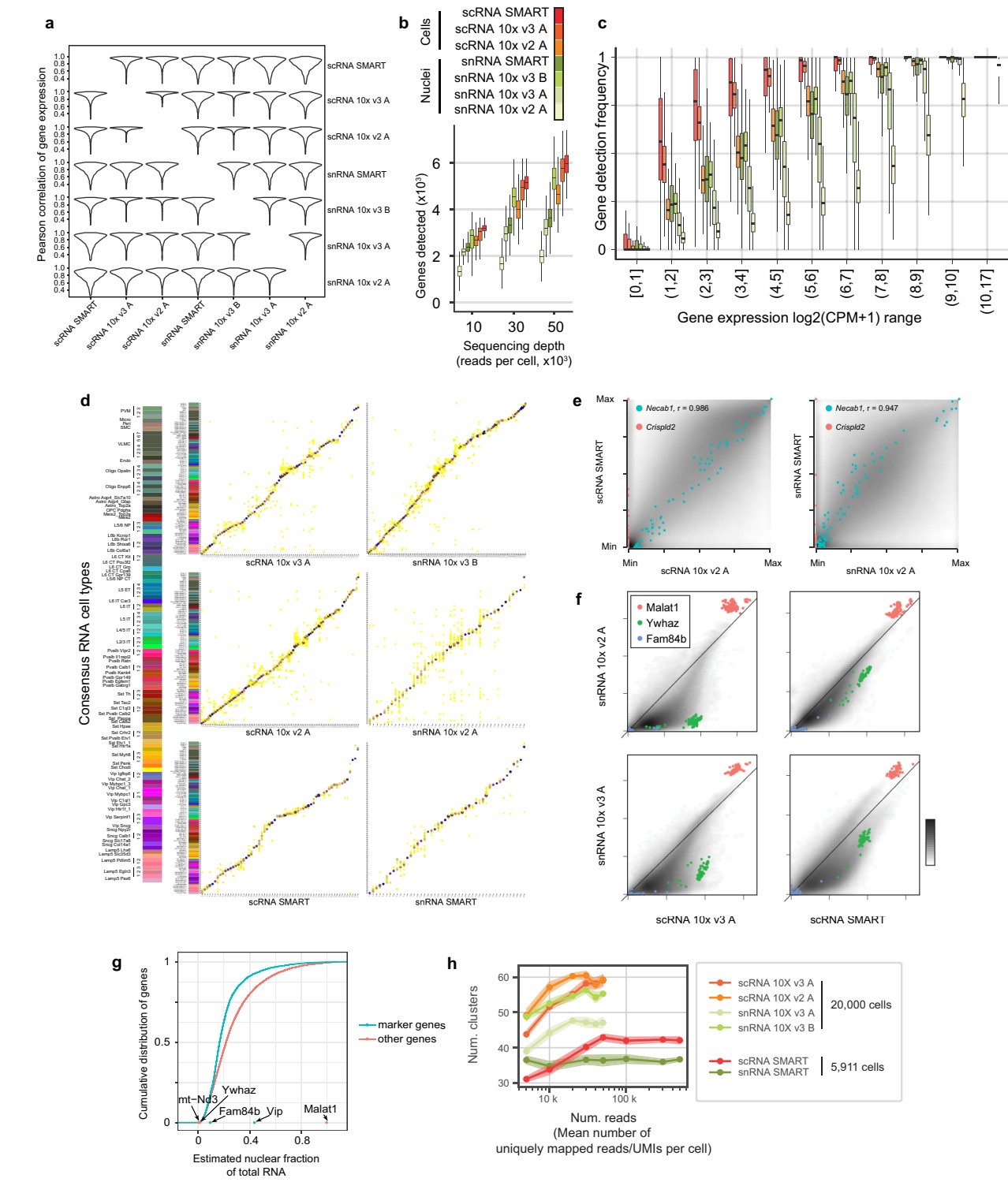

**Extended Data Fig. 2** | See next page for caption.

**Extended Data Fig. 2 | Cluster membership and gene expression consistency across scRNA-seq and snRNA-seq datasets. a**, Pearson correlation of gene expression of 3,792 cell-type-specific marker genes across cell types between every pair of datasets. Each violin plot shows the distribution of correlation values for all genes between a pair of datasets. Most genes have highly conserved gene expression patterns at the cell-type level among all datasets (average correlation of 0.856 across all pairs of comparisons). The most consistent datasets are scRNA 10x v2 and v3 (average correlation of 0.95), while snRNA 10x v3 B is also highly similar to both scRNA 10x v2 and v3 datasets. Overall, we found the differences between single-cell and single-nucleus datasets to be more significant than SMART-Seq versus 10x platform differences. **b**, Number of genes detected per cell or nucleus by each transcriptomic assay as a function of sequencing depth, as determined by downsampling analysis ($n = 79$ independent biological samples; see Supplementary Table 1). **c**, Gene detection frequency (sensitivity) at each gene expression range for each dataset ($n = 79$ independent biological samples; see Supplementary Table 1). Expression of all genes in each cell type was binned based on the average logCPM in scRNA 10x v2 and snRNA 10x v3 B datasets. Single-cell datasets overall have higher sensitivity for gene expression than single-nucleus datasets, with the exception of the snRNA 10x v3 B dataset, which was more sensitive than the scRNA 10x v2 A dataset. For weakly expressed genes, the gene detection frequency can vary dramatically between datasets. For these genes, scRNA SMART was the most sensitive, followed by 10x v3 datasets, all of which showed very robust gene detection. Note that sequencing depth was not considered for this analysis. For **b**, **c**, box-and-whisker plots show the median, the interquartile range (IQR) (25–75th percentile), and the whiskers show the smaller of the data range (minimum to maximum) or 1.5 times the IQR. **d**, Comparisons between clustering analysis of individual datasets with the consensus clusters derived from seven transcriptomic datasets. The size of the dot indicates the number of overlapping cells, and the colour of the dot indicates the Jaccard index (number of cells in intersection/number of cells in union) between the independent and joint clusters. **e**, Comparison of the relative gene expression of marker genes across all cell types between corresponding SMART-Seq and 10x v2 datasets. To compare gene expression directly between SMART-Seq and 10x datasets, which differ in experimental platforms, gene expression quantification software and gene annotation reference, for each gene, we normalized the average $\log_2(\text{CPM} + 1)$ values at the cluster level in the range [0,1] by subtracting the minimum value and then dividing by the maximum value for that gene. The smooth scatter plot corresponds to the normalized gene expression for all marker genes across all types in two datasets, with their overall Pearson correlation (across all marker genes and cell types) highlighted. **f**, Differential enrichment of transcripts in single cells ($x$ axis) versus single nuclei ($y$ axis) across four platforms. Non-coding RNAs such as *Malat1* are enriched in nuclei. **g**, Distribution of the estimated nuclear localization fraction for all mRNAs based on comparison of the snRNA and scRNA 10x v2 datasets[22]. To calibrate the differences among cell types, we sampled the same number of cells in each cluster for both datasets, and aggregated all the cells for estimation. We plot the empirical cumulative density function for the marker genes and all other genes separately. The fraction of nuclear mRNAs for five selected genes are shown along the $x$ axis. As expected, mitochondrial genes such as mt-*Nd3* have almost no nuclear localization, whereas *Vip* is significantly enriched in the nucleus. A selected set of 3,792 cell-type-specific marker genes (see Methods section 'Marker gene selection') have a lower nuclear fraction relative to the other genes (median 16.6%, compared with 21.9% for non-marker genes). **h**, Cluster resolution analysis, showing the number of clusters identified in each transcriptomic dataset with a fixed cluster procedure and resolution ($r = 6$) as a function of the number of sequenced reads, and using the same number of cells for each of the 10x or SMART-Seq datasets. The shaded region shows the s.e.m. from cross-validation with $n = 5$ independent data partitions.

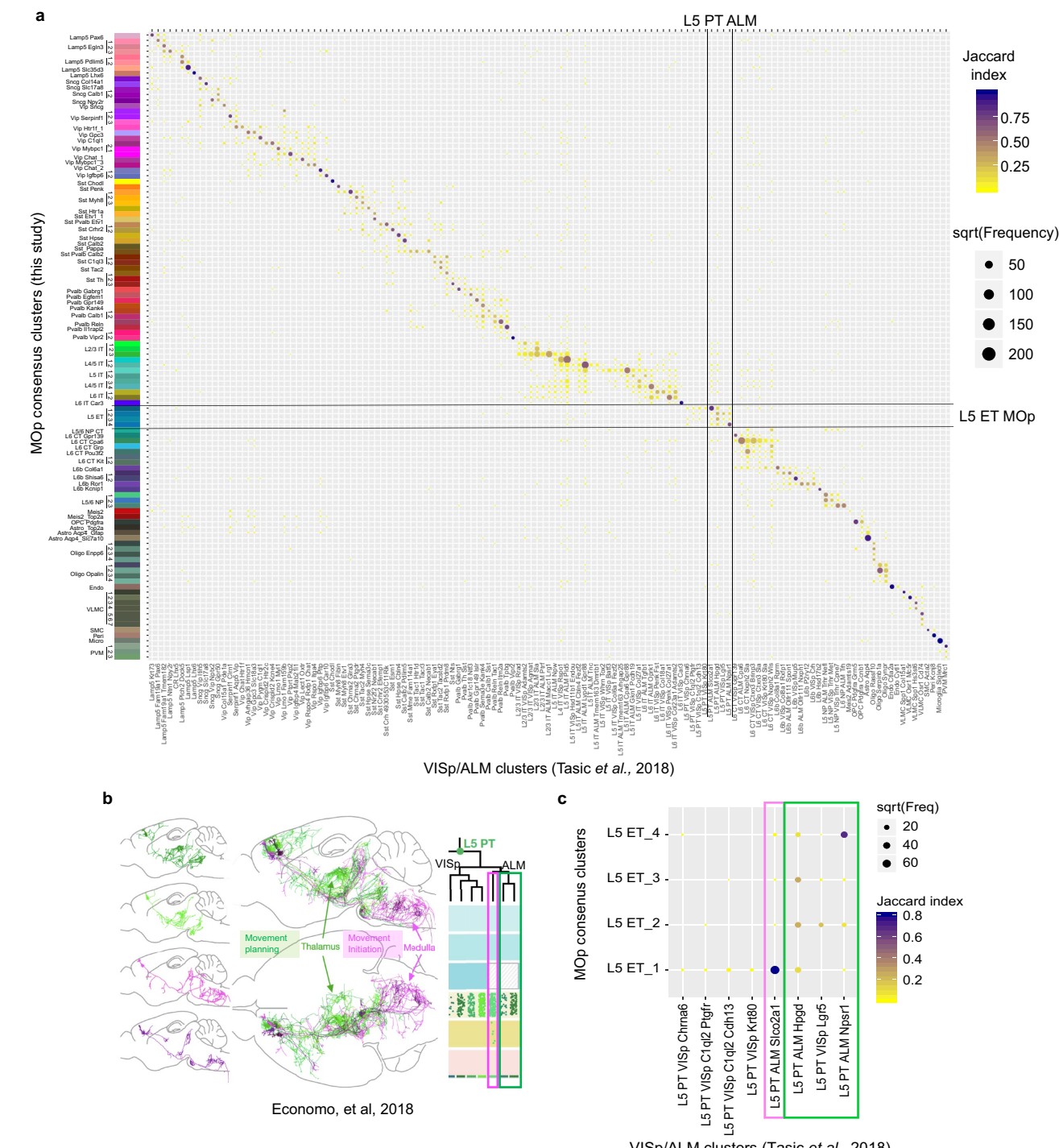

**Extended Data Fig. 3 | Correspondence between the MOp consensus RNA-seq cell-type taxonomy and previously published VISp/ALM cell-type taxonomy[3]. a**, Cells from all scRNA and snRNA MOp datasets were mapped to the most correlated VISp/ALM cell types based on VISp/ALM cell-type markers. The size of the dots indicates the number of overlapping cells, and the colour indicates the Jaccard index (number of cells in intersection/number of cells in union). MOp L5 ET types are mapped predominantly to L5 pyramidal tract (PT) ALM types in the VISp/ALM study. Note that we have adopted the nomenclature 'extratelencephalically projecting (ET)' for these neurons, instead of the

previously used 'pyramidal tract (PT)', owing to the fact that not all of these neurons project to the pyramidal tract leading to the spinal cord. **b**, Three L5 PT ALM types can be divided into two groups with distinct projection patterns. Cells in the pink group project to the medulla and have been functionally associated with movement initiation, while the cells in the green group project to the thalamus, associated with movement planning. Adapted from Economo et al. (2018)[18]. **c**, Enlarged view of the correspondence between MOp L5 ET types and VISp/ALM L5 PT types. Two subsets of medulla-projecting (pink) and thalamus-projecting (green) L5 PT cells are highlighted.

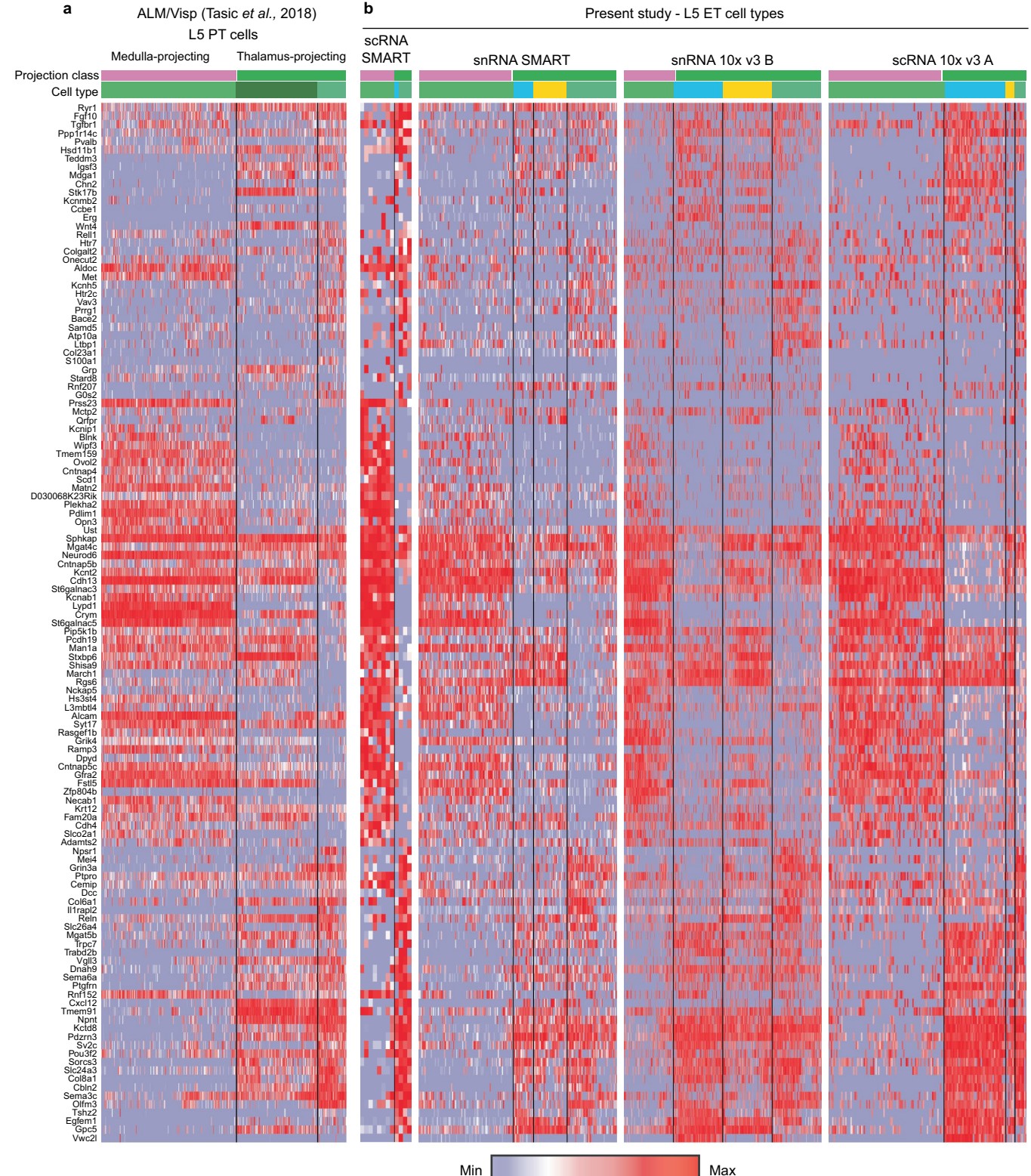

**Extended Data Fig. 4 | Marker genes for L5 ET cell types. a**, Heat map showing expression of a combination of marker genes of L5 PT ALM types in a previously published dataset[3], and marker genes for MOp L5 ET types. The coloured bars on the top indicate the cell type and projection class. **b**, Heat map for MOp L5 ET types in multiple scRNA and snRNA datasets using the same marker genes in the same order as in **a**. Cell types are divided into pink and green groups based on correspondence in Extended Data Fig. 3c.

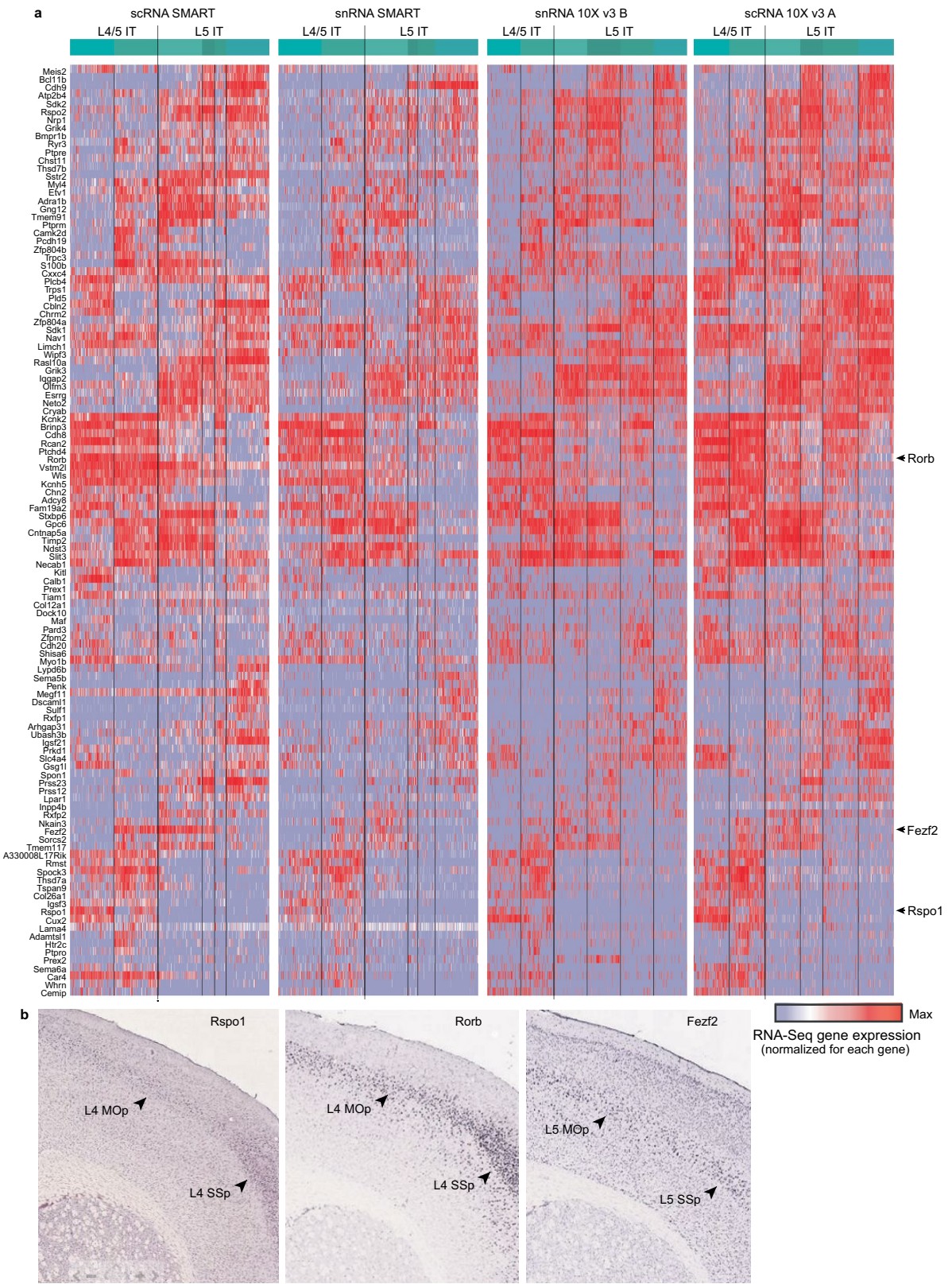

**Extended Data Fig. 5 | Marker genes for L4/5 IT and L5 IT cell types.**
**a**, Heat map of marker genes for MOp L4/5 IT and L5 IT types in multiple scRNA and snRNA datasets. **b**, In situ hybridization (ISH) showing validation of L4 marker genes (*Rspo1* and *Rorb*) and L5 (*Fezf2*) in the mouse MOp. Note that *Rorb* labels both L4 and a subset of L5 neurons.

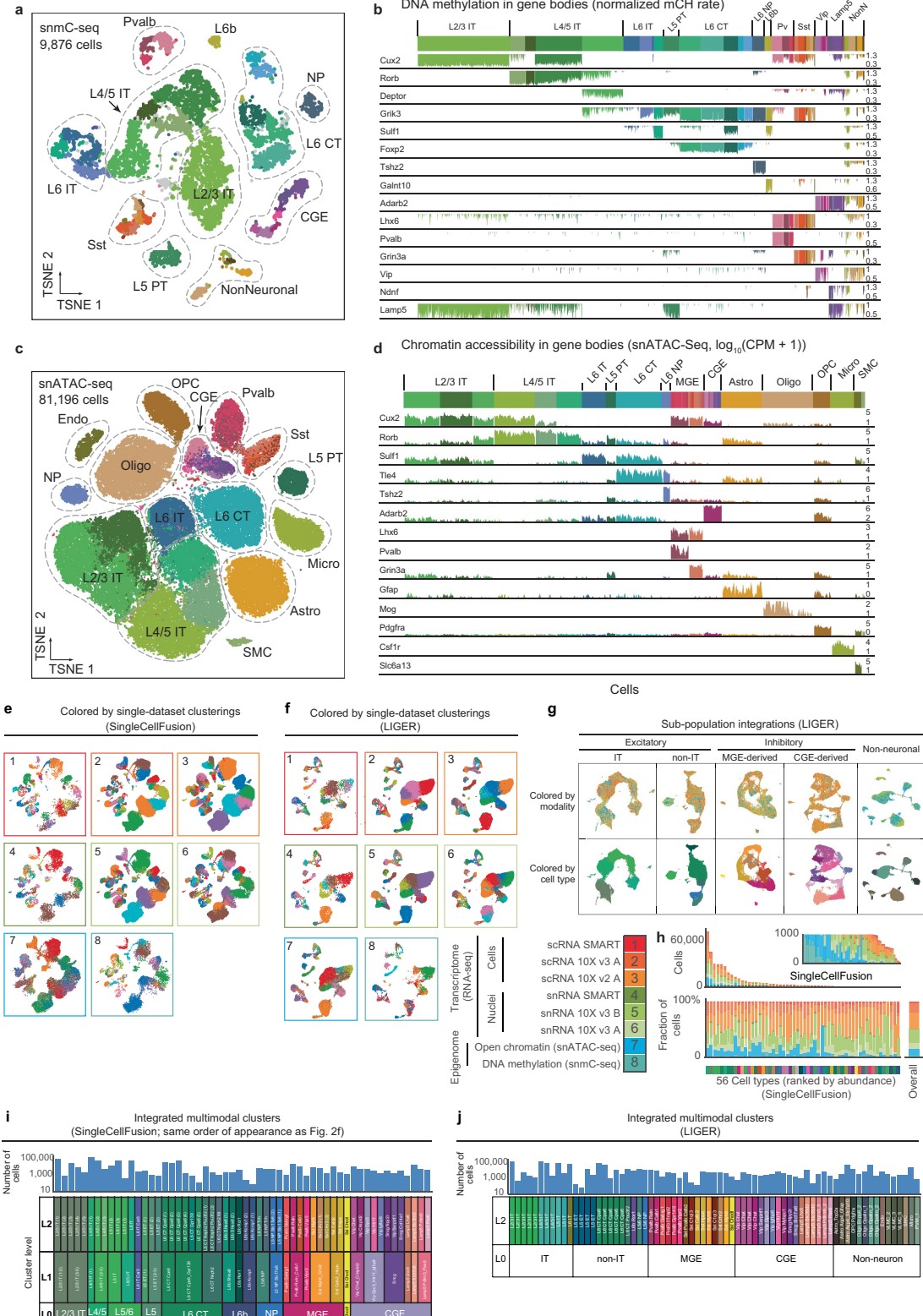

**Extended Data Fig. 6** | See next page for caption.

**Extended Data Fig. 6 | Epigenomic cell types and multimodal integration.**
**a**, Cell-type clusters from single-nucleus methyl-C-Seq (snmC-seq2 (refs. [9,14]))
for 9,876 MOp nuclei are represented in a two-dimensional projection. Labels
indicate broad cell types; the colours show finest cluster resolution. **b**, Non-CG
DNA methylation level (normalized mCH) for each cell at gene bodies of
markers of major cell types. Actively expressed genes have low mCH, indicated
by the coloured bars extending downward. Highly methylated (repressed)
genes appear white in this plot. **c**, Two-dimensional projection of cell-type
clusters from snATAC-seq[11] profiles for 81,196 cells. **d**, Gene body chromatin
accessibility (total snATAC-seq read density, log(CPM + 1)) for marker genes.
For **b** and **d**, each bar represents one cell. The abbreviations of cell type are as in
Fig. 2. CGE/MGE, caudal/medial ganglionic eminence-derived inhibitory cells.
**e**, **f**, Integrated, multimodal UMAP embeddings (SingleCellFusion (**e**); LIGER
(**f**)) coloured by the clusters assigned in separate analysis of each dataset. Each
panel shows the cells from a single dataset. **g**, Integrated analysis of major cell
classes by LIGER. Cells in each of the five cell classes are separately integrated,
illustrating fine-grained resolution of integrated data. **h**, Number of cells in
each of 56 multimodality cell types (SingleCellFusion; L2), ranked by cluster
size. **i**, **j**, Number of cells for 56 integrated clusters (SingleCellFusion L2 (**i**);
LIGER L2 (**j**)), as well as the corresponding coarser clusters (L1, L0). Cluster
order and colour scheme are as shown in Fig. 2.

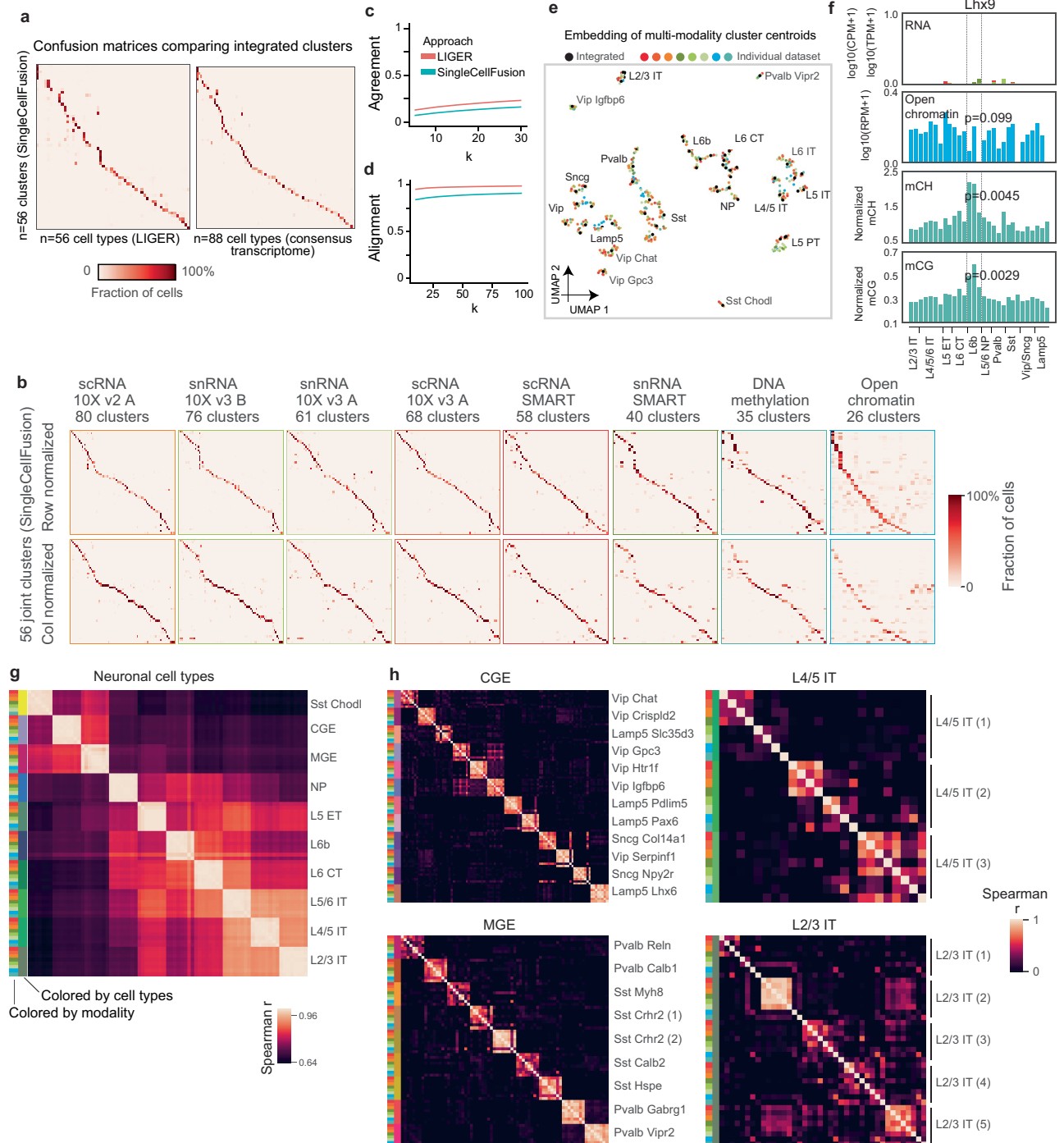

**Extended Data Fig. 7 | Validation of multimodal integration of transcriptomic and epigenomic data. a**, Confusion matrices comparing integrated clusters generated by SingleCellFusion versus clusters generated by LIGER (left), and comparing SingleCellFusion versus consensus transcriptomic taxonomy (right). **b**, Confusion matrix comparing integrated clusters (SingleCellFusion L2) with single-modality clustering for every dataset. **c**, **d**, Agreement and alignment metrics[29] characterize the fidelity of the joint low-dimensional embedding for LIGER and SingleCellFusion. Agreement measures the fraction of KNNs for each dataset that are still nearest neighbours in the low-dimensional embedding. A high value of the agreement metric thus indicates preservation of each dataset's internal structure in the joint embedding. Alignment measures the mixing of datasets in the joint

low-dimensional space, and is a normalized measure of the mean number of KNNs that come from each of the datasets. **e**, Embedding of multimodality cluster centroids. The black dots are cluster centroids of integrated clusters (SingleCellFusion); coloured dots are cluster centroids of individual datasets. **f**, Molecular signatures at the gene body of *Lhx9*, a developmentally expressed transcription factor, across cell types (*n* = 29; SingleCellFusion L1). We found enrichment of mCG and mCH in L6b neurons with no corresponding RNA or ATAC-seq signal. **g**, Spearman correlation matrix for cluster centroid gene expression (measured or imputed) across major cell subclasses for each dataset (SingleCellFusion L0). **h**, Correlation for subsets of inhibitory (CGE and MGE) and excitatory (L4/5 IT and L2/3 IT) neuron types using fine-grained integrated clusters (SingleCellFusion L2).

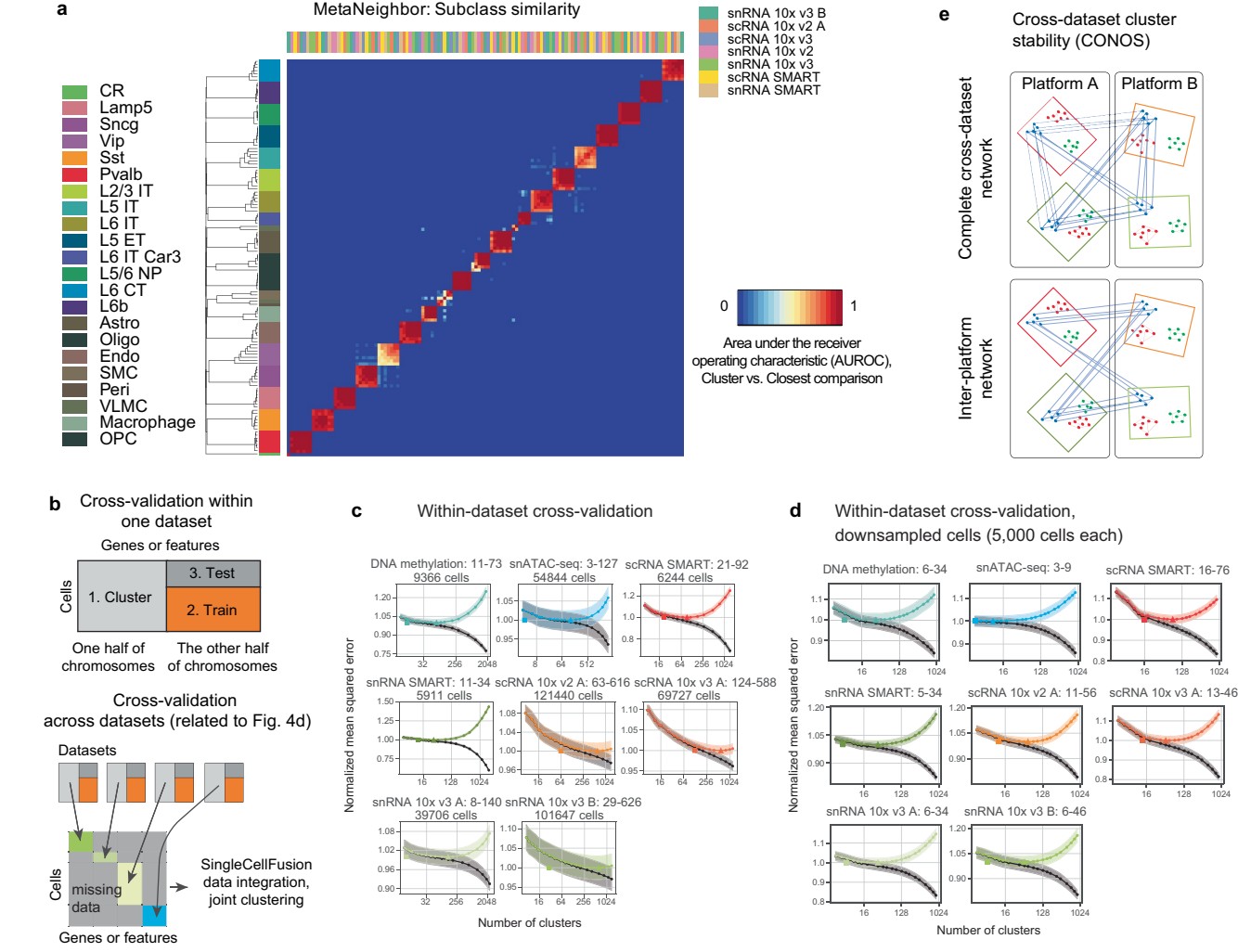

**Extended Data Fig. 8 | MetaNeighbor and cross-validation analysis of cluster reproducibility. a**, Heat map showing replicability scores (MetaNeighbor AUROC) at the subclass level of the independent clusterings of seven RNA-seq datasets. High AUROC indicates that the cell-type labels in one dataset can be reliably predicted based on the nearest neighbours of those cells in another dataset, together with the independent cluster analysis of that dataset. **b**, Scheme for within-dataset and across-dataset cross-validation. **c**, **d** Within-dataset cross-validation analysis for each dataset, either using the full set of cells (**c**) or using a random sample of 5,000 cells (**d**). In each plot, the black curve shows training error, while the coloured U-shaped curve shows the test set error, with a minimum at the cluster resolution that balances over-fitting and under-fitting. The shaded region shows the s.e.m. based on cross-validation with $n = 5$ data partitions. **e**, Transcriptomic platform consistency is assessed by cross-dataset cluster stability analysis (Conos[37]).

# nature research

| | |
|---|---|

# Reporting Summary

Nature Research wishes to improve the reproducibility of the work that we publish. This form provides structure for consistency and transparency in reporting. For further information on Nature Research policies, see Authors & Referees and the Editorial Policy Checklist.

## Statistics

For all statistical analyses, confirm that the following items are present in the figure legend, table legend, main text, or Methods section.

| n/a | Confirmed | |
|---|---|---|
| ☐ | ☒ | The exact sample size (*n*) for each experimental group/condition, given as a discrete number and unit of measurement |
| ☐ | ☒ | A statement on whether measurements were taken from distinct samples or whether the same sample was measured repeatedly |
| ☐ | ☒ | The statistical test(s) used AND whether they are one- or two-sided *Only common tests should be described solely by name; describe more complex techniques in the Methods section.* |
| ☐ | ☒ | A description of all covariates tested |
| ☐ | ☒ | A description of any assumptions or corrections, such as tests of normality and adjustment for multiple comparisons |
| ☐ | ☒ | A full description of the statistical parameters including central tendency (e.g. means) or other basic estimates (e.g. regression coefficient) AND variation (e.g. standard deviation) or associated estimates of uncertainty (e.g. confidence intervals) |
| ☐ | ☒ | For null hypothesis testing, the test statistic (e.g. *F*, *t*, *r*) with confidence intervals, effect sizes, degrees of freedom and *P* value noted *Give P values as exact values whenever suitable.* |
| ☒ | ☐ | For Bayesian analysis, information on the choice of priors and Markov chain Monte Carlo settings |
| ☒ | ☐ | For hierarchical and complex designs, identification of the appropriate level for tests and full reporting of outcomes |
| ☐ | ☒ | Estimates of effect sizes (e.g. Cohen's *d*, Pearson's *r*), indicating how they were calculated |

*Our web collection on statistics for biologists contains articles on many of the points above.*

## Software and code

Policy information about availability of computer code

| Data collection | Software for mapping and analysis of transcriptomic and epigenomic datasets is fully described in the Methods.<br><br>SMART-Seq v4 (SSv4) Ultra Low Input RNA Kit for Sequencing (Takara Cat# 634894) |
|---|---|
| Data analysis | Software for mapping and analysis of transcriptomic and epigenomic datasets is fully described in the Methods<br><br>STAR v2.5.3<br>https://github.com/epiviz/miniatlas,<br>https://github.com/AllenInstitute/scrattch.hicat,<br>https://github.com/r3fang/SnapTools,<br>https://github.com/lhqing/cemba_data (documentation: cemba-data.rtfd.io)<br>https://github.com/gillislab/MetaNeighbor-BICCN,<br>https://github.com/welch-lab/liger,<br>https://github.com/mukamel-lab/SingleCellFusion<br>http://epiviz.nemoanalytics.org/biccn_mop,<br>CONOS(Barkas et al. 2018): https://github.com/kharchenkolab/conos<br>Bismark (Krueger and Andrews, 2011) |

For manuscripts utilizing custom algorithms or software that are central to the research but not yet described in published literature, software must be made available to editors/reviewers. We strongly encourage code deposition in a community repository (e.g. GitHub). See the Nature Research guidelines for submitting code & software for further information.

## Data

Policy information about [availability of data](availability of data)

All manuscripts must include a [data availability statement](data availability statement). This statement should provide the following information, where applicable:

- Accession codes, unique identifiers, or web links for publicly available datasets
- A list of figures that have associated raw data
- A description of any restrictions on data availability

The BICCN MOp data (RRID:SCR_015820) can be accessed via the NeMO archive (RRID:SCR_002001) at accession: https://assets.nemoarchive.org/dat-ch1nqb7. Visualization and analysis resources: NeMO analytics: https://nemoanalytics.org/, Genome browser: https://brainome.ucsd.edu/annoj/BICCN_MOp/, Epiviz browser: https://epiviz.nemoanalytics.org/biccn_mop.

# Field-specific reporting

Please select the one below that is the best fit for your research. If you are not sure, read the appropriate sections before making your selection.

☒ Life sciences ☐ Behavioural & social sciences ☐ Ecological, evolutionary & environmental sciences

For a reference copy of the document with all sections, see [nature.com/documents/nr-reporting-summary-flat.pdf](nature.com/documents/nr-reporting-summary-flat.pdf)

# Life sciences study design

All studies must disclose on these points even when the disclosure is negative.

| | |
|---|---|
| Sample size | Our study focuses on differences between major brain cell populations, which are highly conserved across individual mice. Sample size (number of animals) was determined by the experimental requirements for collection of sufficient tissue for each assay.

For RNA-seq data, we collected tissue from a total of 45 animals for SMART-seq cells, 10 for SMART-seq nuclei, and 3~12 for various 10x platforms. We used relatively larger number of animals for SMART-seq due to use of specific cre-lines and layer specific dissections. Due to highly sensitive gene detection provided by the SMART-seq platform, we collected about ~6000 cells for both cells/nuclei, targeting at least 5 cells for any cell types with abundance more than 0.1%. Our previous studies suggest that highly distinct cell types can be detected with just 5 cells, while more subtle cell type differences can be detected with 20~50 cells. For 10x, we collected at least 100,000 cells for each platform, targeting at least 100 cells for any cell types with abundance more than 0.1%.

For collection of single nuclei by FACS for the epigenomic datasets (snmC-seq and snATAC-seq), we collected tissue from 6-23 individual mice for each sub-region of MOp (different anterior-posterior levels). Larger numbers of mice were used for relatively small subregions, which ensured that at least 2,500 single neuronal nuclei (NeuN+) were obtained from each sample. We collected tissue from two independent pools of animals (biological replicates) to calibrate inter-sample variability due to inter-individual differences.

In no case did we observe differences between individual animals or batches that were similar in magnitude to the reported cell type differences. The number of cells collected was determined by specific limitations of each data modality, and the effect of this sample size was extensively analyzed as part of the paper (Figure 6). |
| Data exclusions | Low quality cells and putative doublets were excluded based on criteria that are described in detail in the Methods.

SMART-seq:  Cells that met any one of the following criteria were removed: < 100,000 total reads, < 1,000 detected genes (CPM > 0), < 75% of reads aligned to the genome, or CG dinucleotide odds ratio > 0.5.

10x RNA-seq: For scRNA datasets, we excluded neurons with fewer than 2000 detected genes and non-neuronal cells with fewer than 1000 detected genes; for snRNA datasets, we excluded neurons with fewer than 1000 detected genes and non-neuronal cells with fewer than 500 detected genes. Doublets were identified using a modified version of the DoubletFinder algorithm and removed when doublet score > 0.3. After clustering of individual datasets, we also removed some clusters that we believed were driven by technical artifacts: clusters with strong markers indicating their regional identities outside of MOp, rare doublet clusters that were not captured by the doublet score but contain the markers from multiple highly distinct cell types (e.g. neurons and non-neuronal types), and low quality clusters showing significant gene loss, but no up-regulated genes relative to another similar cluster. Cells from these clusters were eliminated from downstream integrative analysis. A full description of the procedure for exclusion of artifactual clusters is provided in Methods in the section "Clustering individual datasets".

snATAC-seq: We excluded any single cells that had fewer than 1,000 unique fragments or a TSS enrichment of <10 for any sample sets. We used Scrublet (RRID:SCR_018098)52 to remove potential doublets for every sample set

snmC-seq: We filtered the cells based on  quality metrics: 1) The rate of bisulfite non-conversion as estimated by the rate of methylation at CCC positions (mCCC) < 0.03. mCCC rate reliably estimates the upper bound of bisulfite non-conversion rate8, 2) overall mCG rate > 0.5, 3) overall mCH rate < 0.2, 4) total final reads (combining R1 and R2) > 500,000, 5) Total mapping rate (using Bismark54) > 0.5.

In addition, during the RNA consensus clustering analysis, we excluded clusters as follows (this text is in the Methods):

Removal of low-quality and doublet-driven clusters. We performed differential gene expression analysis between every pair of clusters within each subclass. If any cluster had ≤2 up-regulated genes (fold-change>2, FDR<0.01, with additional penetrance and odds ratio criteria described in Method transcriptome analysis section) compared to another cluster, and had a substantially lower average number of detected genes per cell, we flagged the cluster as low-quality and removed it from further analysis. Next, if the up-regulated genes between any two |

clusters within a subclass were predominantly marker genes for a different subclass, and one of the clusters had significantly higher average genes detected per cell and UMI count, we flagged the cluster as a potential doublet cluster and removed it from further analysis. These criteria led to the exclusion of 8.3% of all cells, the vast majority of which came from the two 10x v3 datasets (scRNA 10X v3 A, snRNA 10X v3 B). While the 10X v3 platform boosts the gene detection for good cells, it does the same to damaged cells or debris, leading to an elevated number of clusters that were excluded for these datasets.

**Replication**

Findings in each modality were extensively compared across biological replicates (at least 2 replicates of each experiment). We did not observe any disagreement between replicates in terms of the biological conclusions, such as the identity of cell types. In addition, we extensively characterize the multimodal correspondence and concordance of the datasets, providing robust validation of cell types.

For the snmC and snATAC data, replicates from the same brain region are co-clustered compared to samples from other brain regions. Within each cluster, we calculated Pearson correlations between each replicates, and found that all replicates are highly conserved (Pearson corr. > 0.95).

**Randomization**

Not applicable. Our study does not compare treatment and control groups. Instead we focus on characterizing the cell types that are present in untreated adult mice. Single cell sequencing provides a random sample of cells derived from the source tissue.

**Blinding**

Not applicable. There were no treatment and control groups, and no pre-defined hypotheses regarding cell type identity.

# Reporting for specific materials, systems and methods

We require information from authors about some types of materials, experimental systems and methods used in many studies. Here, indicate whether each material, system or method listed is relevant to your study. If you are not sure if a list item applies to your research, read the appropriate section before selecting a response.

## Materials & experimental systems

| n/a | Involved in the study |
|---|---|
| ☒ | ☐ Antibodies |
| ☒ | ☐ Eukaryotic cell lines |
| ☒ | ☐ Palaeontology |
| ☐ | ☒ Animals and other organisms |
| ☒ | ☐ Human research participants |
| ☒ | ☐ Clinical data |

## Methods

| n/a | Involved in the study |
|---|---|
| ☒ | ☐ ChIP-seq |
| ☒ | ☐ Flow cytometry |
| ☒ | ☐ MRI-based neuroimaging |

## Animals and other organisms

Policy information about studies involving animals; ARRIVE guidelines recommended for reporting animal research

**Laboratory animals**

Salk: Mouse (mus musculus), strain C57BL6/J (RRID:IMSR_JAX:000664), both males and females, 53-63 days of postnatal age. Male C57BL/6J mice were purchased from Jackson laboratories at 8 weeks of age and maintained in the Salk animal barrier facility on 12-hr dark-light cycles with controlled temperature (20-22 Celcius range) and humidity (30-70% range), and food ad libitum for one week before dissection.

Allen Institute: All procedures were carried out in accordance with Institutional Animal Care and Use Committee protocols at the Allen Institute for Brain Science. Mice were provided food and water ad libitum and were maintained on a regular 12-h day/night cycle at no more than five adult animals per cage. Ambient temperature was set to 72°F and relative humidity was set to 40%.

Broad Institute: Animals were group housed with a 12-hour light-dark schedule and allowed to acclimate to their housing environment for two weeks post arrival. Ambient temperature was set to 70°F ± 2°F and relative humidity was set to 40% ± 10%. All rooms are on 12/12 hour light/dark cycle.

**Wild animals**

Not applicable. No wild animals were used in this study.

**Field-collected samples**

Not applicable. No field collected samples were used in this study

**Ethics oversight**

All procedures at the Allen Institute were carried out in accordance with Institutional Animal Care and Use Committee protocols at the Allen Institute for Brain Science.

All procedures involving animals at MIT were conducted in accordance with the US National Institutes of Health Guide for the Care and Use of Laboratory Animals under protocol number 1115-111-18 and approved by the Massachusetts Institute of Technology Committee on Animal Care. All procedures involving animals at the Broad Institute were conducted in accordance with the US National Institutes of Health Guide for the Care and Use of Laboratory Animals under protocol number 0120-09-16.

Experiments conducted at The Salk Institute in accordance with the US National Institutes of Health Guide for the Care and Use of Laboratory Animals under protocol number 18-00006 and approved by the Institutional Animal Care and Use Committee.

Note that full information on the approval of the study protocol must also be provided in the manuscript.

