## [Peer Review File · Nature]

Manuscript Title: A transcriptomic and epigenomic cell atlas of mouse primary motor cortex

Redactions – Third Party Material

Reviewer Comments & Author Rebuttals

Reviewer Reports on the Initial Version:

Referees' comments:

Referee #1 (Remarks to the Author):

In this paper, the authors find that the mouse primary motor cortex contains over 55 neuronal cell types that are highly replicable. They conclude that "targeted studies of individual cell types, taking advantage of the transcriptional and epigenetic signatures described here, will define their functional roles and significance in the context of neural circuits and behavior". I will not comment on methodological aspects of the paper and will take the authors at their word. Instead I will attempt to understand how this work in the current form informs understanding of what primary motor cortex does.

At no point do the authors inform us as to why they chose one defined cortical area, or primary motor cortex in particular. We are told that the boundaries of mouse primary motor cortex (MOp) were obtained from the Allen Mouse Brain Common Coordinate Framework (CCFv3). It would be of interest to general readers to be given a more precise explanation of how the boundaries are defined. After all, if it turns out that most of the cell types found in MOp are also present in neighboring premotor cortex then what does the boundary actually mean? If the cell types determine function then what does it mean to call something primary motor cortex a priori? Is there some threshold for how much cell-type overlap is allowed before a cortical area is considered the same rather than different from another?

We are provided with no theory mapping cell-type number onto function. What was expected? 2, 10, 50, 100?? Why does it matter? The answer of "55" to systems biologists and physiologists will be as unsatisfying as the answer to the meaning of the universe in *The Hitch-Hikers Guide to the Galaxy*: "42". For any computation we do not need 55 separate logical operators, so why so many cell types? When the authors refer to the functional significance of their results, they at no point explain exactly how their results will contribute to an understanding of circuits and behavior. Perhaps there are other reasons for so many cell types including evolutionary, developmental and metabolic constraints that are not related to the computations relevant to cortical control of movement.

So, what we have here is a very well collected catalogue utterly devoid of either a conceptual framework or even an idea about primary motor cortex. Thus, in effect we have been given a (surely useful) database with some English sentences sprinkled on top of it. I do not see why this data base needs to be read in a journal. The experience is like a reading a phone book despite the fact that it is full of new information.

Referee #2 (Remarks to the Author):

The authors present an atlas of the mouse primary motor cortex across multiple single-cell modalities within the transcriptional and epigenetic space. The work initially describes the transcriptomic results and dataset integration followed by epigenomic, and then integration, which is a solid flow for presenting the data. The authors then perform a rigorous assessment of cluster and cell type stability – an analysis I hope others will follow; as it provides a clear framework for transparently presenting the data without getting into the granularity debate. It is also encouraging that datasets from across a wide variety of platforms or similar platforms from different sites can be integrated and produce consensus cell type classifications. This work provides an excellent analysis path for future studies. Comments below are largely focused on specific items to improve readability / consistency / interpretability.

Line 91 – the number of sequenced molecules per cell is not purely related to sequencing depth, but largely driven by hard limitations on the molecular diversity of the libraries. The consideration of molecular diversity is key to deciding how deep a dataset should be sequenced giving diminishing returns as saturation is approached.

Line 98 – To be consistent with the descriptions of the other modalities, a “(≈[median unique nuclear fragment count])” should be added for the snATAC-seq data.

Lines 100-104 – some numbers should be added in regarding genes detected per cell. The currently language is very vague and qualitative and having parentheses with some quantitative information would sharpen the result description.

Figure 1c – The y-axis is mean mapped reads per cell – this is not entirely clear. Is it total reads sequenced? (i.e. including PCR duplicates), or unique mapped reads (which I assume is the case)? Ideally this would reflect unique, usable reads per cell. I.e UMIs for scRNA (or unique aligned for SMART), unique aligned reads for snmC, unique nuclear reads for snATAC.

Figure 2d – this panel is interesting, but hard to track the column labels to the actual bar graphs. Both due to font size and placement of the labels way at the top. I am not sure how to better represent this, but in its current for it looks pretty but is hard to actually garnish any information from without using a sheet of paper to line things up. It also may be more interpretable to move the row labels to the left so it is the first thing one sees (the RNA dataset labels).

Para at Line 204 – An interesting comparison between scRNA and snRNA would be to compare clustering resolution with different numbers of cells or depth obtained from each dataset. I.e for equivalent cell number and coverage (as in UMIs per cell, downsampled in scRNA to be comparable to snRNA) is a different resolution achieved? Also getting a sense of where these are equivalent with respect to UMIs/cell and cell number for the two modalities could give a sense for readers to know how to best design their experiments. (Note – this is somewhat done in the next results paragraph; however the downsampling analysis would be interesting as well)

Figure 2g – Would this be possible as a density plot or where the y-axis is percentage of clusters (ideally a second y-axis that is percent). As it is shown now the numbers are total and not relative to the baseline number of clusters within each of the three categories.

One note on the analysis in Fig 2h – are the clustering methods used designed primarily for the more-abundant 10X 3' type data versus the full transcript SMART seq data types? If not then that is fine, otherwise a simple note may be warranted.

Fig 3 – is there a reason tSNE was used over UMAP for the epigenetic analysis and UMAP for RNA?

Line 275 – The sentence starting on this line is misleading. Genomic coverage is not a relevant

metric when comparing something like DNA methylation (genome-wide) versus mRNA (data largely restricted to genes). For RNA transcriptome coverage is far more meaningful than genome-wide.

Line 276 – why average and not median? Median reads / cell is much more representative of the dataset

Line 280 – snATAC generates 8,800 reads per cell – I assume this is average? Again median is much more representative.

Line 320 – the statement that regulatory regions are smaller and thus more affected by sparse coverage than gene bodies assumes no enrichment for these small regions. If a majority of sequence reads fall within these regulatory regions, the size of them does not necessarily matter. The other justifications for using gene body measurements are sound though, particularly the challenge of linking distal elements to genes.

Line 354 – “highly specific” is very qualitative – adding in stats would be appropriate, as with a “large” DNA methylation valley (how large? What percentile of size with respect to identified DMVs?) – same goes for “modest” enrichment of ATAC – what fold?

Line 361 – if Lhx9 is under polycomb-mediated repression, would one not expect to see an increased ATAC signal (as stated on line 357)

Intro paragraph for section starting at 425 – this preface is important and well stated. While there may never be a consensus on how to define a cell type from single-cell molecular data, transparency and clear presentation of what was done is paramount.

Figure 6j – the points are small and hard to distinguish – could the plot be zoomed in onto the space the points occupy with the labels falling outside of the plot window?

Referee #3 (Remarks to the Author):

The authors performed single-cell measurements to assess the epigenome and transcriptome of just under a million cells/nuclei from the adult mouse primary motor cortex. These experiments appear to have been performed in different labs, at different times, and using different techniques. Still, the authors were able to integrate the data across modality and batch to find a collection of consensus cell identities that incorporate both epigenetic and transcriptional state. This represents the most comprehensive single-cell resolution brain atlas to date that integrates multiple modalities. With this rich dataset, they briefly interrogate the epigenetic regulation of transcriptional start sites for important marker genes in known cortical cell populations. Furthermore, by comparing their dataset with past single-cell mouse cortical studies, they identify a population that may correspond with a layer IV in the mouse motor cortex—a cell population that does not appear to exist by gross histology. Further investigation will reveal whether the layer IV cells are distributed among other motor cortex layers, or if there is a previously unappreciated layer IV of the adult mouse motor cortex.

Overall, the work performed in this paper represents an incredibly collaborative effort that has yielded the largest single-cell atlas that incorporates multiple measurement modalities. It will enable and accelerate future work in the field and should serve to answer long-sought questions about epigenetic regulation of cell identity in the adult mouse brain. Furthermore, it can be used to compare to other brain regions to identify which specific features, if any, differentiate motor cortex from other cortical areas.

Atlases are the basis of future insights and on that score no one has ever done such a comprehensive integration of approaches. We think that this analysis will set the gold standard, which is important. By being comprehensive the authors provide guidance of what you get for any given investment in a platform. Moreover, establishing correspondence between RNA, DNA methylation, chromatin is not a given and in that way these new data provide powerful insight. Given that people enter into a new system with little to no ground truth known, it's important to understand the limits of each assay and whether and how different analysis modalities can complement each other. Therefore, this paper re-sets the bar and provides an example of how to do it with careful examination of limitations etc.

We think that this paper will be of great interest to the broad scientific community. We have some comments and suggestions for the authors to consider.

- 1) The scope of the dataset, combined with the data quality (number of UMIS/cell, genes detected, integration with ATAC) is unprecedented. It offers motor cortex researchers genetic and epigenetic access to a vast array of cell populations that can be used to interrogate their function in the context of motor behavior. For the single-cell field, it brings us closer to understanding several things: a) the effect of single-cell vs. nuclei transcriptomic measurements, b) the effect of number of cells measured on the ability to discriminate populations in the brain, c) the amount of variation that occurs when using different techniques. In a rapidly evolving field, integration of new and past technologies is more important than ever.
- 2) The cross-validation is exceptionally detailed and robust.
- 3) Cell clusters are annotated by the known expression patterns of their marker genes.
- 4) The correspondence between DNA methylation in gene bodies, chromatin accessibility (ATAC), and transcription of known marker genes is striking (Figure 3), if expected.
- 5) The authors are extremely comprehensive and transparent with the analytical tools used.
- 6) This manuscript seems to position itself as a methods paper in the abstract and introduction (for example, lines 50-55). Yet the scope is not to develop novel integrative bioinformatic tools, but rather to stitch together experiments from diverse modalities and techniques. This is already an active goal in the bioinformatics field—where new integrative tools mine existing datasets and integrate them to create consensus atlases. As it is currently written, this manuscript represents an immensely interesting dataset with few bioinformatic advances. Why then is it framed as such?
- 7) Similarly, the extreme emphasis of the authors on their cross-validation, integration, and bioinformatic implementations means that there is little space to discuss the actual underlying biology. With such a rich dataset there are surely numerous interesting vignettes that demonstrate how useful it can be, and how the reader should interpret it. The authors briefly describe potentially novel cell populations (the layer IV cortical population that has not been observed before) yet spend very little time connecting their work with the existing motor cortex literature.
- 8) Furthermore, there is no validation of any cell population by in situ hybridization or immunohistochemistry. Given the scope of the work, it would be absurd to ask for complete validation of the results—however, the biological stories that the authors bring up could quite easily be validated. We propose that the authors validate their putative layer IV cluster with in situ hybridization. They also discuss, for instance, the expression of *Tac1* mRNA in a cluster of Pvalb-positive neurons (line 120). It is unclear what the implications of this finding are, and how the integration with epigenetic data contributes to our understanding of these cells (or the regulation of this transcript).

9) A small point, but please clarify lines 211-216. They state that Ywhaz mRNA is specifically localized to the somata, rather than the dendrites, of hippocampal neurons. The authors use this as a justification for why it is depleted from the nucleus in their measurements—this does not logically follow. It is unclear how the somatic vs. dendritic vs. axonal localization of mRNA influences the relative abundance by nuclear vs. whole cell RNA sequencing.

10) At line 139, the authors state that data was integrated using `scratch.hicat`. However, Figure 2a (to which it refers) clearly shows the Seurat integration (with CCA, presumably) across modalities. Was Seurat used to visualize, but `scratch.hicat` used to joint cluster? How did the results compare to graph-based clustering following Seurat Integration?

11) The scientific strength of this paper lies in the mixed modality epigenetic/transcriptomic measurements. There is a brief digression into methylation/accessibility /transcript expression in Figure 3, but little follow-up.

12) The authors briefly address the difference between the snRNAseq 10X V3 A and B protocols, but do not go into specific details about what separates them in a clear manner. The results of the snRNA V3 B are truly incredible—more genes recovered than SMART-seq is highly interesting—and it would be useful for the field to focus slightly more on which factors affected the data quality.

Author Rebuttals to Initial Comments:

Referees' comments:

Referee #1 (Remarks to the Author):

In this paper, the authors find that the mouse primary motor cortex contains over 55 neuronal cell types that are highly replicable. They conclude that “targeted studies of individual cell types, taking advantage of the transcriptional and epigenetic signatures described here, will define their functional roles and significance in the context of neural circuits and behavior”. I will not comment on methodological aspects of the paper and will take the authors at their word. Instead I will attempt to understand how this work in the current form informs understanding of what primary motor cortex does.

At no point do the authors inform us as to why they chose one defined cortical area, or primary motor cortex in particular. We are told that the boundaries of mouse primary motor cortex (MOp) were obtained from the Allen Mouse Brain Common Coordinate Framework (CCFv3). It would be of interest to general readers to be given a more precise explanation of how the boundaries are defined. After all, if it turns out that most of the cell types found in MOp are also present in neighboring premotor cortex then what does the boundary actually mean? If the cell types determine function then what does it mean to call something primary motor cortex a priori? Is there some threshold for how much cell-type overlap is allowed before a cortical area is considered the same rather than different from another?

We appreciate the Reviewer's observation that the choice of MOp was not sufficiently discussed in the paper. We have revised the Introduction to specifically address this. We added the following paragraph:

The primary motor cortex (MOp) is a critical region for control of voluntary movement, and its connectivity and function are well conserved across mammals. It shares many circuit motifs and cellular components with other cortical regions. Traditionally MOp is considered an agranular cortex due to the lack of a cytoarchitecturally-defined granular layer (layer 4), although neurons with Layer 4-like connectivity have been identified in MOp (Yamawaki et al. 2014). MOp is also relatively devoid of species-specific cellular structures often seen in sensory cortical areas, such as the whisker barrels in the rodent primary somatosensory cortex and the elaborate layer 4 with multiple sublayers in the primate primary visual cortex. Within the BRAIN Initiative Cell Census Network (BICCN), our goal is to obtain a census of cell types across the brain of several mammalian species by integrating multiple single-cell omics approaches. We selected MOp as the starting point for our joint efforts due to its relatively conserved structure and function across mammalian species. Our MOp atlas is a case study of the expansive potential, as well as the technical limitations, of single-cell molecular methods for comprehensive brain-wide atlasing of cell types.

We also appreciate the need to more clearly define the boundaries of MOp. Indeed, this is a non-trivial challenge, despite the relatively well-defined functional understanding of MOp and its canonical connectivity. Several other BICCN consortium papers as part of this package directly address the spatial boundaries of MOp using spatial transcriptomics (MERFISH) and a combination of histological stains and anatomical tracing. For the purposes of our paper, we adopted procedures for defining the boundaries of MOp (and other cortical regions analyzed by BICCN) using coronal sections of the mouse brain. These sections were compared with the Allen Mouse Brain Common Coordinate Framework (CCF, version 3), which defined MOp based on 23 brain-wide datasets including ISH for 5 marker genes, 12 connectivity datasets, and 5 transgenic Cre-lines¹. The collective information of marker genes and axon projections from these datasets distinguishes MOp from its adjacent regions - primary somatosensory cortex (SSp) on the lateral side and secondary motor cortex (MOs, aka premotor cortex) on the medial side. The boundaries of dissected regions were illustrated in Extended Data Fig. 8 of our original submission for the epigenomic datasets. In the revised manuscript, we have expanded this figure to include examples of the photo-documentation of MOp tissue dissections for transcriptomic data generated by both the Allen Institute and the Broad Institute.

The reviewer also raised a great point regarding whether cell types determine function and thus define a cortical region that is distinct from other regions. We believe that identifying a catalog of cell types as the “parts list” of a brain region such as MOp is an essential first step toward understanding the region’s function. The group of cell types is necessary but not sufficient to determine function. The collection of input and output connections among them as well as with other cell types in other parts of the brain is likely another necessary component underlying

function. Indeed, other recent papers, e.g. the BICCN MERFISH paper² and an Allen Institute comprehensive single-cell transcriptomic study of the entire cortex and hippocampus³, have shown that the cell types are mostly shared between MOp and its neighboring MOs, suggesting a largely similar cellular architecture. It may be that functional specificity between these two regions is defined at the connectivity level.

We are provided with no theory mapping cell-type number onto function. What was expected? 2, 10, 50, 100?? Why does it matter? The answer of "55" to systems biologists and physiologists will be as unsatisfying as the answer to the meaning of the universe in *The Hitch-Hikers Guide to the Galaxy*: "42". For any computation we do not need 55 separate logical operators, so why so many cell types? When the authors refer to the functional significance of their results, they at no point explain exactly how their results will contribute to an understanding of circuits and behavior. Perhaps there are other reasons for so many cell types including evolutionary, developmental and metabolic constraints that are not related to the computations relevant to cortical control of movement.

Of course the precise number of cell types holds no special significance in itself. However, understanding the complexity of the cellular components of brain circuits and their roles in cognitive function requires drawing category boundaries. We took a data-driven approach to address a major open question, namely what level of granularity in defining cell types is reproducibly supported by multiple biological measures and technical modalities. We agree with the reviewer that a single number (not even 42) cannot be a satisfying answer to this question; however, unlike virtually all previous single-cell studies, we offer an empirical approach to objectively define the range of plausible cell type categories that can be supported by gene expression and epigenomic profiles. Fig. 2 and Fig. 6 of our paper document multiple analyses, including novel approaches for cross-validation, as well as *MetaNeighbor*⁴, and *CONOS*⁵, which apply statistical criteria to assess the reproducibility of cell types at a range of cluster resolutions. The range of reproducible cell types that we report, from ~55 up to ~115 cell types in MOp, is consistent with recent, smaller-scale single cell studies^{6,7}, and it reconfirms the striking diversity of mammalian neuron types.

Although we do not yet know the computational significance of all these types, the fact that so many cell types have been strongly conserved through mammalian evolution⁸ suggests that they do have a consequential role for animal behavior and survival. We do not argue that all the cell types we identify are directly involved in motor control; instead, we posit that a comprehensive atlas of MOp cell types is a necessary foundation for understanding all of the biological functions of this region, including its role in motor control.

So, what we have here is a very well collected catalogue utterly devoid of either a conceptual framework or even an idea about primary motor cortex.

Thus, in effect we have been given a (surely useful) database with some English sentences sprinkled on top of it. I do not see why this data base needs to be read in a journal. The experience is like a reading a phone book despite the fact that it is full of new information.

We appreciate that the reviewer acknowledges the usefulness of the data resource we provide. However, we respectfully disagree that our study lacks a conceptual framework. Rather than approaching brain function through a narrow lens of specific hypotheses concerning individual brain regions, our study, and indeed the BICCN consortium, is committed to the idea that a multimodal, comprehensive and data-driven account of the brain's cellular components is necessary for building the next generation of biologically grounded models of neural circuit function. Whereas theoretical and computational neuroscience has traditionally limited itself to considering a small handful of neuronal cell types (and, as a rule, ignoring glia altogether), we believe that the complexity of brain cell types is not an inconvenience to be ignored but an opportunity to gain new insight.

In the context of the broader BICCN, what we uncovered is not just the number of cell types (as in a phone book), but the potential connectional and functional properties of these cell types and their relatedness to each other. Our consortium used multiple approaches to connect cell type markers with function and connectivity, including patch-seq⁹ and innovative retrograde-labeling combined with DNA methylation profiling¹⁰. In our study, we show that the ~55 neuron types in mouse MOp are organized in a hierarchical manner, first divided into glutamatergic excitatory and GABAergic inhibitory classes and then, under each major branch, further divided into multiple, more refined subclasses and types (Fig. 2). Through their gene signatures these cell types can be related to previously recognized neuron types with specific physiological and connectional properties (e.g. multiple Sst and Pvalb interneuron types, and two layer 5 cortico-subcortical projection neurons with differential target specificity). These are biologically realistic circuit components that will inform modeling and understanding of the function of the motor cortex circuit. Furthermore, we also discovered genomic regulatory elements of these cell types that can be used to build genetic tools to probe their functions.

We have added several of these points and highlighted other biological insights in the Discussion section:

Our data provide new insights into the molecular architecture of MOp cell types. The neuropeptide Substance P precursor, *Tac1*, marks a subset of Pvalb cells and is strongly upregulated in rodent MOp following motor learning^{11,12}. We found that *Tac1* is expressed in two subtypes of MOp interneurons (Pvalb_Calb1 and Pvalb_ReIn), and our epigenomic data identified a cell type-specific enhancer ~24 kb upstream of the gene promoter. We provide new evidence that MOp harbors an excitatory neuron population expressing markers of layer 4 thalamic-recipient neurons, including *Cux2*, *Rspo1* and *Rorb*¹³. The laminar distribution of these cells has been confirmed by ISH of these marker genes and in a parallel study by MERFISH². This discovery revises the traditional understanding of MOp as an agranular cortex lacking L4. We further identified networks of gene expression regulatory

elements, marked by overlapping regions of open chromatin and cell type-specific demethylation, harboring sequence motifs that identify the key transcriptional regulators. For example, by combining epigenetic and gene expression data we identified *Rfx3* as a critical factor for L2/3 IT cells. We also identified genes with non-canonical regulatory signatures, such as the enrichment of mCG in *Lhx9* specifically in L6b excitatory cells. These data show that integrated analysis can uncover epigenetic cell type signatures that are absent in the transcriptome, potentially informing about the developmental trajectory of neuron populations.

Referee #2 (Remarks to the Author):

The authors present an atlas of the mouse primary motor cortex across multiple single-cell modalities within the transcriptional and epigenetic space. The work initially describes the transcriptomic results and dataset integration followed by epigenomic, and then integration, which is a solid flow for presenting the data. The authors then perform a rigorous assessment of cluster and cell type stability – an analysis I hope others will follow; as it provides a clear framework for transparently presenting the data without getting into the granularity debate. It is also encouraging that datasets from across a wide variety of platforms or similar platforms from different sites can be integrated and produce consensus cell type classifications. This work provides an excellent analysis path for future studies. Comments below are largely focused on specific items to improve readability / consistency / interpretability.

We thank this Reviewer for the recognition of the rigorous precedent our paper has set for the field in assessing the stability and robustness of cell type classification. Indeed, a major aim of our MOp study was to set objective and rigorous standards for ongoing and future studies within the BICCN and in other groups to ensure reproducibility and progressive accumulation of knowledge about cell types across studies. We are gratified that the reviewer recognizes the importance of this investment in rigorous validation and comparative analysis across datasets and modalities.

Line 91 – the number of sequenced molecules per cell is not purely related to sequencing depth, but largely driven by hard limitations on the molecular diversity of the libraries. The consideration of molecular diversity is key to deciding how deep a dataset should be sequenced giving diminishing returns as saturation is approached.

We appreciate the Reviewer's suggested clarification, and we have revised this sentence:

The datasets we produced reflect the inherent tradeoff in single cell sequencing assays between the number of sequenced molecules per cell, which depends on cell size as well as the efficiency of RNA or DNA capture and sequencing depth, vs. the total number of cells that can be assayed for a fixed total cost.

Moreover, we have taken this opportunity to add a new analysis of the complexity of our molecular sequencing libraries. We added new columns in Supplementary Table 1 showing the median number of reads per cell, and uniquely mapped reads. We used these to estimate the number of duplicated reads in each dataset and inferred the library size, which is now shown in Fig. 1c (square symbols). Based on this calculation we estimate that we have sequenced ~30% of all unique molecules (transcripts or fragments of transcripts) in the SMART-Seq libraries, and >98% of all the unique molecules in the 10x libraries. The epigenomic datasets each cover ~60% of the available library complexity.

Line 98 – To be consistent with the descriptions of the other modalities, a “(~[median unique nuclear fragment count])” should be added for the snATAC-seq data.

We now show the median number of unique molecular fragments in Fig. 1c for snATAC-Seq and for each of the other datasets. We have clarified the figure legend to explain this. We have also added these numbers to the Results section text.

Lines 100-104 – some numbers should be added in regarding genes detected per cell. The currently language is very vague and qualitative and having parentheses with some quantitative information would sharpen the result description.

We have added the precise values for each dataset as a new column in Supplementary Table 1, and also specified some examples in the text:

Subsampling analysis of RNA-Seq datasets (Fig. 1e, Supplementary Table 1) shows that in general, scRNA-Seq detects more genes per cell (up to ~7,100 median genes/cell for 10x, 10,000 for SMART) than snRNA-Seq (4,000 for 10x, 5,800 for SMART). Moreover, the 10x v3 platform performs substantially better than 10x v2, detecting 60-100% more genes.

Figure 1c – The y-axis is mean mapped reads per cell – this is not entirely clear. Is it total reads sequenced? (i.e. including PCR duplicates), or unique mapped reads (which I assume is the case)? Ideally this would reflect unique, usable reads per cell. I.e UMIs for scRNA (or unique aligned for SMART), unique aligned reads for snmC, unique nuclear reads for snATAC.

We have clarified the figure label and legend to explain that this panel shows unique RNA or DNA fragments. For the 10x datasets this corresponds to UMIs, whereas for the SMART-Seq, ATAC and DNA methylation datasets this corresponds to unique (deduplicated) sequences.

Figure 2d – this panel is interesting, but hard to track the column labels to the actual bar graphs. Both due to font size and placement of the labels way at

the top. I am not sure how to better represent this, but in its current for it looks pretty but is hard to actually garnish any information from without using a sheet of paper to line things up. It also may be more interpretable to move the row labels to the left so it is the first thing one sees (the RNA dataset labels).

To improve the legibility of this figure, we have reorganized Fig. 2. We have increased the font size and adjusted the positions of the labels to improve the overall accessibility of the figure. We have also included the full data from this figure in tabular format in Supplementary Table 2. The revised Fig. 2c is shown here:

Para at Line 204 – An interesting comparison between scRNA and snRNA would be to compare clustering resolution with different numbers of cells or depth obtained from each dataset. Ie for equivalent cell number and coverage (as in UMIs per cell, downsampled in scRNA to be comparable to snRNA) is a different resolution achieved?

Also getting a sense of where these are equivalent with respect to UMIs/cell and cell number for the two modalities could give a sense for readers to know how to best design their experiments. (Note – this is somewhat done in the next results paragraph; however the downsampling analysis would be interesting as well)

Fig. 6 explores this question. In particular, Fig. 6a compares the number of clusters that are derived from each dataset as a function of the number of downsampled cells using a consistent cluster resolution parameter. Fig. 6d shows a statistical cross-validation analysis of cluster reproducibility as a function of downsampled cell number.

To directly address the reviewer’s question, we have added an analysis in which we sample the same number of cells and number of reads per cell for each dataset (Extended Data Figure 2h; see below). The single-cell datasets generate more clusters than single-nucleus datasets, when they are downsampled to the same number of cells and reads, and after controlling for the same sequencing protocols

(SMART-seq versus SMART-seq, 10x versus 10x).

h

Figure 2g – Would this be possible as a density plot or where the y-axis is percentage of clusters(ideally a second y-axis that is percent). As it is shown now the numbers are total and not relative to the baseline number of clusters within each of the three categories.

We have added a panel, Fig. 2e, that displays the fraction of replicable clusters instead of the number of cross-dataset clusters. We decided not to add a second y-axis with the percentage of clusters because the total number of clusters varies within classes (Glutamatergic/GABAergic/Non-Neurons) and across datasets. To convert the number of replicable clusters into a percentage of clusters, we pooled information across datasets and divided the total number of replicable clusters (clusters that belong to one of the 70 cross-dataset clusters) by the total number of clusters.

e MetaNeighbor cross-dataset

One note on the analysis in Fig 2h – are the clustering methods used designed primarily for the more-abundant 10X 3' type data versus the full transcript SMART seq data types? If not then that is fine, otherwise a simple note may be warranted.

We used three off-the-shelf clustering methods that are commonly used in the field for both 3' type data(such as 10x) and full transcript data (SMART-Seq). Seurat

and SC3 have been ranked as the top 2 methods across a variety of metrics and datasets (including 10x and Smart-Seq)¹⁴, while Monocle relies on the same clustering algorithm as Seurat with different pre-processing steps.

Fig 3 – is there a reason tSNE was used over UMAP for the epigenetic analysis and UMAP for RNA?

For consistency with the rest of the paper, we have replaced the tSNE plots in Fig. 3 with UMAP plots. The results are generally consistent, and both methods provide reasonable 2-dimensional embeddings that accurately visualize the relationships among cell types.

Line 275 – The sentence starting on this line is misleading. Genomic coverage is not a relevant metric when comparing something like DNA methylation (genome-wide) versus mRNA (data largely restricted to genes). For RNA transcriptome coverage is far more meaningful than genome-wide.

We agree that this sentence was phrased in a misleading way, and we agree that the genomic coverage is not the most relevant metric for RNA-Seq. Our intent was to compare the DNA methylation with the RNA data in terms of the number of unique sequence fragments, which gives a rough measure of the amount of information available. We have revised the sentence as follows:

“In particular, DNA methylation data provides broad genomic coverage based on a large number of unique mapped reads per cell (1.7 million unique mapped reads on average), similar to the sampling depth of SMART-seq single cell transcriptome datasets (2.1M reads/cell on average).”

Line 276 – why average and not median? Median reads / cell is much more representative of the dataset

We agree, and have replaced the mean with the median.

Line 280 – snATAC generates 8,800 reads per cell – I assume this is average? Again median is much more representative.

The reviewer is correct; 8,800 reads per cell is the average. We have updated it to median uniquely mapped reads per reviewer’s request. The median is 3,778 unique mapped reads per cell.

Line 320 – the statement that regulatory regions are smaller and thus more affected by sparse coverage than gene bodies assumes no enrichment for these small regions. If a majority of sequence reads fall within these regulatory regions, the size of them does not necessarily matter. The other justifications for using gene body measurements are sound though, particularly the

challenge of linking distal elements to genes.

For DNA methylation data, sequenced fragments are more or less uniformly distributed throughout the genome with no particular enrichment at regulatory regions. Therefore, sparseness is a critical limiting factor when analyzing small regions. The reviewer makes a good point with respect to snATAC-Seq data, for which sequenced reads are indeed enriched at open chromatin regions. However, the number of sequenced fragments per cell (median ~4,000) is far smaller than the total number of open chromatin regions. Thus, data for each cell are quite sparse. Using gene bodies overcomes this challenge because gene bodies typically include multiple enhancer and promoter regions that, together, contain a greater total amount of open chromatin and therefore are better represented in snATAC datasets.

Line 354 – “highly specific” is very qualitative – adding in stats would be appropriate, as with a “large” DNA methylation valley (how large? What percentile of size with respect to identified DMVs?) – same goes for “modest” enrichment of ATAC – what fold?

We have revised the text and added numbers to the qualitative statements. The CG and non-CG DNA methylation levels are higher in L6b cell types compared to others (CG: $p=0.0029$, fold change=1.64; non-CG: $p=0.0045$, fold change=1.97; two-sided Wilcoxon test), whereas the chromatin accessibility level does not have a significant enrichment ($p=0.099$, fold change=0.53) (extended data figure 6k). The DNA methylation valley (DMV) around Lhx9 spans 14 kilo-base pairs (chr1:138,836,000-138,850,000; browser view), which is larger than typical DMVs with a median length of 6-7 kilo-base pairs¹⁵. The revised text is:

For instance, at the Lhx9 locus, we found a highly specific enrichment of CG and non-CG DNA methylation in L6b excitatory neurons (CG: $p=0.0029$, fold change=1.64; non-CG: $p=0.0045$, fold change=1.97; Fig. 4g, Extended Data Fig. 6k). Lhx9 was covered by a large (14 kbp) DNA methylation valley (DMV) in each of the other cell types.

Line 361 – if Lhx9 is under polycomb-mediated repression, would one not expect to see an increased ATAC signal (as stated on line 357)

We do not see a significant difference in ATAC-seq signal across cell type ($p=0.099$, two-sided Wilcoxon test). We have updated extended data figure 6k to include p-values. We have also clarified this sentence in the main text: “Despite this cell type-specific epigenetic profile, we found no expression of Lhx9 RNA in any cell type and no significant enrichment of ATAC-Seq reads.”

Note that the relationship between Polycomb-mediated repression and chromatin accessibility is complex and not fully understood. A recent study using a knockout of

Suz12 to manipulate PRC2 showed that Polycomb drives chromatin compaction and reduced chromatin accessibility in mammary organoids¹⁶.

Intro paragraph for section starting at 425 – this preface is important and well stated. While there may never be a consensus on how to define a cell type from single-cell molecular data, transparency and clear presentation of what was done is paramount.

We are gratified that the reviewer appreciated this. We agree that transparency in the definition and resolution of cell types is an essential, though challenging, goal for single cell studies.

Figure 6j – the points are small and hard to distinguish – could the plot be zoomed in onto the space the points occupy with the labels falling outside of the plot window?

To improve legibility of the panel, we created a zoomed in version where points and labels are clearly visible. To highlight our main point (that there is a broad agreement between the integrative methods), we included an inset to show that we are zooming into a region of high performance.

Referee #3 (Remarks to the Author):

The authors performed single-cell measurements to assess the epigenome and transcriptome of just under a million cells/nuclei from the adult mouse primary motor cortex. These experiments appear to have been performed in different labs, at different times, and using different techniques. Still, the authors were able to integrate the data across modality and batch to find a collection of consensus cell identities that incorporate both epigenetic and transcriptional state. This represents the most comprehensive single-cell resolution brain atlas to date that integrates multiple modalities. With this rich

dataset, they briefly interrogate the epigenetic regulation of transcriptional start sites for important marker genes in known cortical cell populations. Furthermore, by comparing their dataset with past single-cell mouse cortical studies, they identify a population that may correspond with a layer IV in the mouse motor cortex—a cell population that does not appear to exist by gross histology. Further investigation will reveal whether the layer IV cells are distributed among other motor cortex layers, or if there is a previously unappreciated layer IV of the adult mouse motor cortex.

Overall, the work performed in this paper represents an incredibly collaborative effort that has yielded the largest single-cell atlas that incorporates multiple measurement modalities. It will enable and accelerate future work in the field and should serve to answer long-sought questions about epigenetic regulation of cell identity in the adult mouse brain. Furthermore, it can be used to compare to other brain regions to identify which specific features, if any, differentiate motor cortex from other cortical areas.

Atlases are the basis of future insights and on that score no one has ever done such a comprehensive integration of approaches. We think that this analysis will set the gold standard, which is important. By being comprehensive the authors provide guidance of what you get for any given investment in a platform. Moreover, establishing correspondence between RNA, DNA methylation, chromatin is not a given and in that way these new data provide powerful insight.

Given that people enter into a new system with little to no ground truth known, it's important to understand the limits of each assay and whether and how different analysis modalities can complement each other. Therefore, this paper re-sets the bar and provides an example of how to do it with careful examination of limitations etc.

We thank this Reviewer for the recognition of the comprehensive and integrative data generation and analysis described in our paper that sets the gold standard for the field.

We think that this paper will be of great interest to the broad scientific community. We have some comments and suggestions for the authors to consider.

- 1) The scope of the dataset, combined with the data quality (number of UMIS/cell, genes detected, integration with ATAC is unprecedented. It offers motor cortex researchers genetic and epigenetic access to a vast array of cell populations that can be used to interrogate their function in the context of motor behavior. For the single-cell field, it brings us closer to understanding several things: a) the effect of single-cell vs. nuclei transcriptomic measurements, b) the effect of number of cells measured on the ability to discriminate populations in the brain, c) the amount of variation that occurs

when using different techniques. In a rapidly evolving field, integration of new and past technologies is more important than ever.

2) The cross-validation is exceptionally detailed and robust.

3) Cell clusters are annotated by the known expression patterns of their marker genes.

4) The correspondence between DNA methylation in gene bodies, chromatin accessibility (ATAC), and transcription of known marker genes is striking (Figure 3), if expected.

5) The authors are extremely comprehensive and transparent with the analytical tools used.

We thank the Reviewer for the above positive comments.

6) This manuscript seems to position itself as a methods paper in the abstract and introduction (forexample, lines 50-55). Yet the scope is not to develop novel integrative bioinformatic tools, but rather to stitch together experiments from diverse modalities and techniques. This is already an active goal in the bioinformatics field—where new integrative tools mine existing datasets and integrate them to create consensus atlases. As it is currently written, this manuscript represents an immensely interesting dataset with few bioinformatic advances. Why then is it framed as such?

We have now revised the abstract and introduction to reduce the emphasis on methodology and focus on new findings and insights. The sentence highlighted by the reviewer has been revised to read: *“Although a comprehensive atlas should incorporate anatomical and physiological information, the high throughput of single cell sequencing assays currently presents the best opportunity for establishing a broad-based transcriptomic and epigenomic understanding of the diverse cellular components of a brain region.”*

7) Similarly, the extreme emphasis of the authors on their cross-validation, integration, and bioinformatic implementations means that there is little space to discuss the actual underlying biology. With such a rich dataset there are surely numerous interesting vignettes that demonstrate how useful it can be, and how the reader should interpret it. The authors briefly describe potentially novel cell populations (the layer IV cortical population that has not been observed before) yet spend very little time connecting their work with the existing motor cortex literature.

Among the set of companion and flagship papers in this BICCN paper package, the main focus of this paper is indeed on the rigorous analysis and integration of the multiple transcriptomic and epigenomic datasets. The biological implications of

identifying new cell types such as the L4 excitatory cells are investigated in greater depth in the flagship paper and other companion papers². This is why we didn't elaborate on it here. However, the reviewer has a good point, as a standalone paper we should discuss the implications of our findings here. We have now added this to the Discussion section:

Our data provide new insights into the molecular architecture of MOp cell types. The neuropeptide Substance P precursor, *Tac1*, marks a subset of Pvalb cells and is strongly upregulated in rodent MOp following motor learning^{11,12}. We found that *Tac1* is expressed in two subtypes of MOp interneurons (Pvalb_Calb1 and Pvalb_ReIn), and our epigenomic data identified a cell type-specific enhancer ~24 kb upstream of the gene promoter. We provide new evidence that MOp harbors an excitatory neuron population expressing markers of layer 4 thalamic-recipient neurons, including *Cux2*, *Rspo1* and *Rorb*¹³. The laminar distribution of these cells has been confirmed by ISH of these marker genes and in a parallel study by MERFISH². This discovery revises the traditional understanding of MOp as an agranular cortex lacking L4. We further identified networks of gene expression regulatory elements, marked by overlapping regions of open chromatin and cell type-specific demethylation, harboring sequence motifs that identify the key transcriptional regulators. For example, by combining epigenetic and gene expression data we identified *Rfx3* as a critical factor for L2/3 IT cells. We also identified genes with non-canonical regulatory signatures, such as the enrichment of mCG in *Lhx9* specifically in L6b excitatory cells. These data show that integrated analysis can uncover epigenetic cell type signatures that are absent in the transcriptome, potentially informing about the developmental trajectory of neuron populations.

8) Furthermore, there is no validation of any cell population by in situ hybridization or immunohistochemistry. Given the scope of the work, it would be absurd to ask for complete validation of the results—however, the biological stories that the authors bring up could quite easily be validated. We propose that the authors validate their putative layer IV cluster with in situ hybridization.

We have now added ISH images of all the relevant marker genes for the L4 excitatory neuron type in Extended Data Fig 5b as validation (shown below). Please also note that all the cell types described in this paper have been comprehensively validated by MERFISH and that work is described in a MERFISH focused BICCN companion paper².

They also discuss, for instance, the expression of *Tac1* mRNA in a cluster of *Pvalb*-positive neurons (line 120). It is unclear what the implications of this finding are, and how the integration with epigenetic data contributes to our understanding of these cells (or the regulation of this transcript).

We use the *Tac1* gene as an example to demonstrate our ability to correlate its mRNA expression with gene body methylation and chromatin accessibility, both within the gene body and at nearby (intergenic) distal regulatory elements. To clarify our interpretation of the findings, we have added the following to the Discussion:

The neuropeptide Substance P precursor Tac1 marks a subset of Pvalb cells and is strongly upregulated in rodent MOp following motor learning (Vruwink et al. 2001; Hertler et al. 2017). We found that Tac1 is expressed in two subtypes of MOp interneurons (Pvalb_Calb1 and Pvalb_Reln), and our epigenomic data identified a cell type-specific enhancer ~24 kb upstream of the gene promoter.

9) A small point, but please clarify lines 211-216. They state that *Ywhaz* mRNA is specifically localized to the somata, rather than the dendrites, of hippocampal neurons. The authors use this as a justification for why it is depleted from the nucleus in their measurements—this does not logically follow. It is unclear how the somatic vs. dendritic vs. axonal localization of mRNA influences the relative abundance by nuclear vs. whole cell RNA sequencing.

We have removed the statement regarding soma vs dendrite localization of this mRNA. Our intention here was to connect our findings about somatic vs. nuclear localization with other complementary data regarding dendritic vs. somatic localization. However, we agree that there is no direct logical connection between these two findings and we have removed the reference to avoid confusion.

10) At line 139, the authors state that data was integrated using *scratch.hicat*. However, Figure 2a (to which it refers) clearly shows the Seurat integration (with CCA, presumably) across modalities. Was Seurat used to visualize, but

scratch.hicat used to joint cluster? How did the results compare to graph-based clustering following Seurat Integration?

To clarify, both clustering and visualization were performed using scratch.hicat; our reference to Seurat in the legend of Fig. 2 was an error, and we apologize for this confusion. The scratch.hicat method has been described in the method section for transcriptome analysis related to Figure 2. We achieved higher cell type resolution than standard Seurat integration, and validated the conservation of cell type markers across platforms both globally (Extended Data Figure 2a) and locally (for L4/5 IT subtypes in Extended Data Figure 5, and for L5 ET subtypes in Extended Data Figure 4).

11) The scientific strength of this paper lies in the mixed modality epigenetic/transcriptomic measurements. There is a brief digression into methylation/accessibility/transcript expression in Figure 3, but little follow-up.

The epigenomic data (DNA methylation and open chromatin) are analyzed on their own in Fig. 3, followed by much more extensive integrated analysis (Fig. 4 and 6) and biological interpretation of gene regulatory networks (Fig. 5). For example, Fig. 5 analyzes epigenetically defined putative regulatory regions (enhancers), marked by differential DNA methylation (DMRs) and/or open chromatin (ATAC peaks). We highlight examples of these regulatory regions in Fig. 5g-i (and also Fig. 1f), which illustrates the rich information about intergenic regions harboring cell type-specific regulatory marks. Moreover, we performed DNA sequence motif enrichment analysis for two of the most abundant cell types (L2/3 IT and L6 CT excitatory neurons), uncovering transcription factors potentially responsible for their distinct regulation (Fig. 5f). Further in-depth analysis of the BICCN epigenomic datasets is contained in companion manuscripts describing DNA methylation¹⁷ and open chromatin¹⁸.

The authors briefly address the difference between the snRNAseq 10X V3 A and B protocols, but do not go into specific details about what separates them in a clear manner. The results of the snRNA V3 B are truly incredible—more genes recovered than SMART-seq is highly interesting—and it would be useful for the field to focus slightly more on which factors affected the data quality.

We have carefully considered the reviewer's question regarding the superior quality, in terms of genes recovered, of the Macosko lab's 10x v3 snRNA protocol. We believe there are three reasons, and that the summation of benefits imparted by the combination of these accounts for the outcome. We have now added these comments to the Methods section.

This 10x v3 snRNA-seq protocol resulted in a higher number of genes recovered compared to other snRNA-seq methods. We believe there are three

reasons, and that the summation of benefits imparted by the combination of these accounts for the outcome.

First, mouse brains are perfused with a solution emulating artificial CSF and then rapidly frozen over liquid nitrogen vapor in such a way that RNA integrity is highly preserved. The resulting bioanalyzer RIN scores of the starting brain tissues are routinely 9.8. Storage of the brains before dissection is at -80°C in the presence of a hydration sink of 1ml of OCT compound pre-frozen into the bottom of a 5ml storage tube. This prevents sublimation and subsequent desiccation-dependent RNA fragmentation.

Second, we performed expeditious sample processing. We have a well-trained group of technicians who process the mouse brain (as above) and then perform the dissociation and FACS and 10X processing (as below) in one continuous protocol without pauses. For example, each mouse is perfused and ready for dissection within minutes (10), and we limit our sample size to 6 such that no sample is waiting to move through the process.

Third, the frozen tissue snRNA Seq protocol incorporates two main features that we believe are important to quality because they prevent the nuclei from “leaking” valuable signal and simultaneously contaminating the barcoded nuclei mixture with exogenous RNA signal. Feature one is a very low level of centrifugation, which we have found to cause both loss of signal and increased exogenous signal.

Feature two is the inclusion of an excipient reagent, BASF Kollidon VA-64, as per the McCarroll Lab protocol ¹⁹.

References

1. Wang, Q. *et al.* The Allen Mouse Brain Common Coordinate Framework: A 3D Reference Atlas. *Cell* (2020) doi:10.1016/j.cell.2020.04.007.
2. Zhang, M. *et al.* Molecular, spatial and projection diversity of neurons in primary motor cortex revealed by in situ single-cell transcriptomics. *bioRxiv* 2020.06.04.105700 (2020) doi:10.1101/2020.06.04.105700.
3. Yao, Z. *et al.* A taxonomy of transcriptomic cell types across the isocortex and hippocampal formation. *bioRxiv* 2020.03.30.015214 (2020)

doi:10.1101/2020.03.30.015214.

4. Crow, M., Paul, A., Ballouz, S., Huang, Z. J. & Gillis, J. Characterizing the replicability of cell types defined by single cell RNA-sequencing data using MetaNeighbor. *Nat. Commun.* **9**, 884 (2018).
5. Barkas, N. *et al.* Wiring together large single-cell RNA-seq sample collections. *bioRxiv* 460246 (2018) doi:10.1101/460246.
6. Tasic, B. *et al.* Shared and distinct transcriptomic cell types across neocortical areas. *Nature* **563**, 72–78 (2018).
7. Saunders, A. *et al.* Molecular Diversity and Specializations among the Cells of the Adult Mouse Brain. *Cell* **174**, 1015–1030.e16 (2018).
8. Bakken, T. E., Jorstad, N. L., Hu, Q., Lake, B. B. & Tian, W. Evolution of cellular diversity in primary motor cortex of human, marmoset monkey, and mouse. *bioRxiv* (2020).
9. Scala, F. *et al.* Phenotypic variation within and across transcriptomic cell types in mouse motor cortex. *bioRxiv* 2020.02.03.929158 (2020) doi:10.1101/2020.02.03.929158.
10. Zhang, Z. *et al.* Epigenomic Diversity of Cortical Projection Neurons in the Mouse Brain. *bioRxiv* 2020.04.01.019612 (2020) doi:10.1101/2020.04.01.019612.
11. Vruwink, M., Schmidt, H. H., Weinberg, R. J. & Burette, A. Substance P and nitric oxide signaling in cerebral cortex: anatomical evidence for reciprocal signaling between two classes of interneurons. *J. Comp. Neurol.* **441**, 288–301 (2001).
12. Hertler, B., Hosp, J. A., Blanco, M. B. & Luft, A. R. Substance P signalling in primary motor cortex facilitates motor learning in rats. *PLoS One* **12**, e0189812 (2017).
13. Yamawaki, N., Borges, K., Suter, B. A., Harris, K. D. & Shepherd, G. M. G. A

- genuine layer 4 in motor cortex with prototypical synaptic circuit connectivity. *Elife* **3**, e05422 (2014).
14. Duò, A., Robinson, M. D. & Soneson, C. A systematic performance evaluation of clustering methods for single-cell RNA-seq data. *F1000Res.* **7**, 1141 (2018).
15. Mo, A. *et al.* Epigenomic Signatures of Neuronal Diversity in the Mammalian Brain. *Neuron* **86**, 1369–1384 (2015).
16. Michalak, E. M. *et al.* Canonical PRC2 function is essential for mammary gland development and affects chromatin compaction in mammary organoids. *PLoS Biol.* **16**, e2004986 (2018).
17. Liu, H. *et al.* DNA Methylation Atlas of the Mouse Brain at Single-Cell Resolution. *bioRxiv* 2020.04.30.069377 (2020) doi:10.1101/2020.04.30.069377.
18. Li, Y. E. *et al.* An Atlas of Gene Regulatory Elements in Adult Mouse Cerebrum. *bioRxiv* 2020.05.10.087585 (2020) doi:10.1101/2020.05.10.087585.
19. Bortolin, L., Goldman, M. & McCarroll, S. Extraction of Nuclei from Brain Tissue v1 (protocols.io.2srged6). doi:10.17504/protocols.io.2srged6.

Reviewer Reports on the First Revision:

Referees' comments:

Referee #1 (Remarks to the Author):

I appreciate the authors' very thoughtful responses. I still maintain that they have provided a rigorous approach to cell-typing, which will certainly serve as a springboard for future discovery-based science. I do not, however, think the paper provides any digestible new insights about the function of motor cortex. This is because no framework is provided within which to interpret the cell-type results reported.

John Krakauer

Referee #2 (Remarks to the Author):

The authors have addressed all of my comments comprehensively. I believe the revised manuscript has a number of clarifications that improve readability and interpretation and it is

suitable for publication in its current form.

Referee #3 (Remarks to the Author):

In their revised manuscript, the authors make some efforts to tie their multi-omics dataset to functionally defined cell types and novel biological insights. In some cases these efforts are successful and yield interesting new biological findings, while in others the new information presented brings up new questions that should be addressed. We also thank the authors for their detailed examination of why the 10X v3 protocols yielded such vastly different data quality. We are satisfied that it will be included in the Methods section. It is remarkable that these seemingly mundane differences are so important, and we believe that their inclusion will help improve data quality when other groups use similar techniques in novel systems. Overall, we appreciate the added focus on functional relevance and continue to believe that these datasets represent an immense resource that will be of immediate usefulness to the scientific public. We hope that the authors take our concerns seriously and provide the requisite evidence to support claims that they make in the revised manuscript.

The concerns are as follows:

1. The identification of layer 4 cells in primary motor cortex is presented in the paper as running contrary to conventional wisdom. However, as the authors briefly state, past work demonstrated that both with respect to thalamic connectivity, as well as using the genetic marker *Rorb* (indeed they use what looks like the exact same image from the Allen Atlas to show this), there does exist a layer 4 within the primary motor cortex. This should not be presented as a novel finding, as it was already shown in the past (PMID: 25525751). We previously asked for an in situ hybridization demonstrating novelty. The authors included a single-channel in situ from a previously published cell atlas, rather than performing a novel in situ showing mutual exclusion between for instance L4 and L5. To be more explicit, if the authors wish to demonstrate that their work contributes to novel information about L4 in the motor cortex, we suggest that they pick an orthogonal marker of L4 that has not been previously described and show that it indeed specifically localizes adjacent to L5 (marked by *Fezf2*). Otherwise, the authors are simply reprinting a previously published image that is suggestive but not definitive. If we are mistaken, please correct us. The following sentence in the authors' revised manuscript is grammatically correct but misleading: "We confirmed the specificity of the expression of these genes in MOp by in situ hybridization."

2. In contrast, we find the *Tac1* evidence to be highly compelling. It helps to confirm and contextualize a previous finding and could directly lead to follow-up experiments specifically examining the role of substance P in interneuron networks. It is doubly interesting that these data are also supported by the epigenomic integration!

3. We appreciate the inclusion of images in the extended data section showing dissections of motor cortex. However, in the absence of a genetic marker we worry about the troubling possibility that not all cells are derived from motor cortex. If the authors could compare their data to the next-door sensory cortex and find a few specific markers, then show (even a previously generated Allen image) an in situ, that would probably allay any fears that the other reviewers might have regarding specificity. If the analysis is performed and those markers do not exist, then that should be reported. The reasoning behind this request is as follows—when making an argument about differences/similarities between neighboring regions, one should be held to a high standard of proof.

4. The authors compare the number of 'sequenced fragments' between smart-seq and droplet-based sequencing approaches. This is a highly misleading comparison, as it vastly overstates the improvement in transcriptome coverage when comparing the two techniques. Specifically, the following sentence and Fig 1c are problematic. "By contrast, full-length transcript sequencing using SMART-Seq v4 captured a greater number of unique molecules per cell (1-2.1 million)". They may

have sequenced 1-2.1 million molecules, but they do not represent unique molecules per cell. The total RNA content of a cell is almost an order of magnitude lower than 2.1 million molecules. We point this out because while the text is technically correct (at least mostly), it will definitely be interpreted differently. The lack of UMIs in SMART-seq makes this comparison impossible, and at best the number of genes detected should be compared.

5. We urge the authors to present a slightly more detailed description of how they identified intra and extra-telencephalic identities to each cluster. We do not question the result, but they should include that Slc30a3 (and other relevant markers) were used to make these determinations.

6. Many of our concerns above (1,3,5) stem from the following: the authors do not directly discuss (or show data for) the companion MERFISH paper. We effectively reviewed that manuscript as well (on biorxiv; <https://www.biorxiv.org/content/10.1101/2020.06.04.105700v1>) and found high concordance between these two works. We think this strongly adds to the value and validity of the present work and think that several of our concerns can be addressed simply by direct reference and/or generation of figures based on some of the MERFISH data. We recognize that this becomes a delicate balancing act, but this seems like a far simpler and higher-resolution solution than independently replicating findings that have, for all intents and purposes, been demonstrated elsewhere.

Author Rebuttals to First Revision:

Referees' comments:

Referee #1 (Remarks to the Author):

I appreciate the authors' very thoughtful responses. I still maintain that they have provided a rigorous approach to cell-typing, which will certainly serve as a springboard for future discovery-based science. I do not, however, think the paper provides any digestible new insights about the function of motor cortex. This is because no framework is provided within which to interpret the cell-type results reported.

John Krakauer

Referee #2 (Remarks to the Author):

The authors have addressed all of my comments comprehensively. I believe the revised manuscript has a number of clarifications that improve readability and interpretation and it is suitable for publication in its current form.

We thank Referees #1 and #2 for their comments, which have strengthened our manuscript.

Referee #3 (Remarks to the Author):

In their revised manuscript, the authors make some efforts to tie their multi-omics dataset to functionally defined cell types and novel biological insights. In some cases these efforts are successful and yield interesting new biological findings, while in others the new information presented brings up new questions that should be addressed. We also thank the authors for their detailed examination of why the 10X v3 protocols yielded such vastly different data quality. We are satisfied that it will be included in the Methods

section. It is remarkable that these seemingly mundane differences are so important, and we believe that their inclusion will help improve data quality when other groups use similar techniques in novel systems. Overall, we appreciate the added focus on functional relevance and continue to believe that these datasets represent an immense resource that will be of immediate usefulness to the scientific public. We hope that the authors take our concerns seriously and provide the requisite evidence to support claims that they make in the revised manuscript.

We are grateful for Referee #3's appreciation of the usefulness and impact of our study and data resource. We have taken the Referee's additional concerns seriously and offer our responses below.

The concerns are as follows:

1. The identification of layer 4 cells in primary motor cortex is presented in the paper as running contrary to conventional wisdom. However, as the authors briefly state, past work demonstrated that both with respect to thalamic connectivity, as well as using the genetic marker *Rorb* (indeed they use what looks like the exact same image from the Allen Atlas to show this), there does exist a layer 4 within the primary motor cortex. This should not be presented as a novel finding, as it was already shown in the past (PMID: 25525751).

We agree with the Reviewer that Layer 4 neurons have previously been identified in MOp. The manuscript makes this clear, e.g. in the Introduction where we cite the paper mentioned by the Reviewer (line 59-61):

“Traditionally, MOp is considered an agranular cortex due to the lack of a cytoarchitecturally-defined granular layer (layer 4), although neurons with Layer 4-like connectivity have been identified in MOp [Ref to PMID: 25525751]”

We previously asked for an in situ hybridization demonstrating novelty. The authors included a single-channel in situ from a previously published cell atlas, rather than performing a novel in situ showing mutual exclusion between for instance L4 and L5. To be more explicit, if the authors wish to demonstrate that their work contributes to novel information about L4 in the motor cortex, we suggest that they pick an orthogonal marker of L4 that has not been previously described and show that it indeed specifically localizes adjacent to L5 (marked by *Fezf2*). Otherwise, the authors are simply reprinting a previously published image that is suggestive but not definitive. If we are mistaken, please correct us. The following sentence in the authors' revised manuscript is grammatically correct but misleading: “We confirmed the specificity of the expression of these genes in MOp by in situ hybridization.”

We agree with the Reviewer that the single-gene ISH images we provided (Extended Data Figure 5) are not new data but rather taken from the existing Allen Brain Atlas database. We have now clarified this in the legend for Extended Data Fig. 5. However, we did identify a novel marker gene, *Rspo1*, that is specifically associated with the transcriptomically defined L4 cell type. Please note that a previously known “L4” marker gene as the Reviewer pointed out, *Rorb*, is actually not completely specific to L4 but also

labels some L5 IT cells (Figure 2c). In ExtendedData Figure 5b, we showed side-by-side ISH images for three marker genes that are selected based on their specificity in labeling transcriptomic types: *Rspo1* for L4 IT, *Rorb* for L4/5 IT and *Fezf2* for L5 IT. Although the ISH data are taken from an existing database, we believe the message is novel, which is that we indeed have confirmed the identification of neurons with Layer 4 transcriptomic signatures (marked by *Rspo1*) in MOp and these neurons are located adjacent to L5 (marked by *Fezf2*).

To provide further definitive confirmation, as suggested by the Reviewer we have now also added a reference to Zhang et al. 2020 (<https://doi.org/10.1101/2020.06.04.105700>), a BICCN companion paper that confirms the Layer 4 population in MOp using spatial transcriptomics (MERFISH). In that paper, the authors identified 7 subtypes of L4/5 IT neurons. Two of these subtypes were localized to the neighboring SSp region (Zhang et al, Extended Data Figure 9), while the remaining five subtypes were located within MOp itself (Fig 4). The paper provides evidence that two of the clusters (L4/5 IT-1,2) express *Rorb* and *Cux2* but not *Fezf2*, while three of the clusters (L4/5 IT-3,4,5) express *Fezf2*.

We have now added a reference to these data in our manuscript (line 222-223):

“The localization of *Rorb*+/*Fezf2*– neurons within MOp has also been confirmed using a spatial transcriptomics method, MERFISH³²[ref(Zhang et al. 2020)].”

[Figure Redacted]

Fig. 4c,d,e from Zhang et al., 2020 (<https://doi.org/10.1101/2020.06.04.105700>)

2. In contrast, we find the *Tac1* evidence to be highly compelling. It helps to confirm and contextualize a previous finding and could directly lead to follow-up experiments specifically examining the role of substance P in interneuron networks. It is doubly interesting that these data are also supported by the epigenomic integration!

We appreciate the Reviewer’s positive comments. This is, indeed, an interesting finding that has emerged from the multimodal single-cell data.

3. We appreciate the inclusion of images in the extended data section showing dissections of motor cortex. However, in the absence of a genetic marker we worry about the troubling possibility that not all cells are derived from motor cortex. If the authors could compare their data to the next-door sensory cortex and find a few specific markers, then show (even a previously generated Allen image) an in situ, that would probably allay any fears that the other reviewers might have regarding specificity. If the analysis is performed and those markers do not exist, then that should be reported. The reasoning behind this request is as follows—when making an argument about differences/similarities between neighboring regions, one should be held to a high standard of proof.

It is challenging to define the boundaries of a brain region like MOp, which should

ultimately be based on connectivity and function. Moreover, we agree that there is a possibility of a small amount of contamination of cells from neighboring regions (e.g. SSp) within our dissected MOp samples. Although our paper defines the cell types in MOp, we do not specifically analyse differences between MOp and other cortical regions. However, as the Reviewer suggests, definitive evidence about the spatial distribution of cells with a given molecular signature is provided by spatial transcriptomics methods. We have therefore added a reference to the BICCN MERFISH paper by Zhang et al. (2020).

As shown in Extended Data Figures 8 and 9 of that paper (shown below), the authors found 3 types of excitatory neurons which were present in SSp (labeled L4/5 IT SSp 1 and L4/5 IT SSp 2) or in the lateral portion of MOp (L6 IT Car3) (Extended Data Fig. 9). The authors directly compared their cell type signatures with the transcriptomic clusters we identified in our MOp dataset (Extended Data Fig. 8a) and with the multimodal transcriptomic+epigenomic clusters from our SingleCellFusion analysis (Extended Data Fig. 8b). There were no MOp clusters in our analysis that mapped directly to the SSp 1 or SSp 2 clusters (highlighted red boxes). Instead, the L4/5 IT clusters we identified in MOp align with L4/5 IT 1, 2, 3, 4, 5. We did find a cluster of cells in MOp corresponding to the L6 IT Car3 population, which appears to reside in the lateral portion of MOp.

[Figure Redacted]

Zhang et al., 2020 (<https://doi.org/10.1101/2020.06.04.105700>)

[Figure Redacted]

Zhang et al., 2020 (<https://doi.org/10.1101/2020.06.04.105700>)

4. The authors compare the number of 'sequenced fragments' between smart-seq and droplet-based sequencing approaches. This is a highly misleading comparison, as it vastly overstates the improvement in transcriptome coverage when comparing the two techniques.

Specifically, the following sentence and Fig 1c are problematic. "By contrast, full-length transcript sequencing using SMART-Seq v4 captured a greater number of unique molecules per cell (1-2.1 million)". They may have sequenced 1-2.1 million molecules, but they do not represent unique molecules per cell. The total RNA content of a cell is almost an order of magnitude lower than 2.1 million molecules. We point this out because while the text is technically correct (at least mostly), it will definitely be interpreted differently. The lack of UMIs in SMART-seq makes this comparison impossible, and at best the number of genes detected should be compared.

This is an important point, and we appreciate the Reviewer's careful attention to the accuracy of this statement. We have revised it as follows (lines 100-104):

"By contrast, full-length transcript sequencing using SMART-Seq v4 captured a greater number of distinct mRNA fragments per cell (1-2.1 million), but covered fewer cells (~6,300 per dataset). Notably, SMART-Seq data lack UMIs to distinguish unique molecules; hence the sequenced fragments represent many samples of a smaller number

of transcripts.”

5. We urge the authors to present a slightly more detailed description of how they identified intra and extra-telencephalic identities to each cluster. We do not question the result, but they should include that *Slc30a3* (and other relevant markers) were used to make these determinations.

We have added the following information to the paper (lines 192-196):

“For example, we identified four clusters of excitatory neurons (expressing *Slc17a7* encoding vesicular glutamate transporter *Vglut1*) that express markers of deep layers (*Fezf2*) as well as *Fam84b* and *Bcl6*, unique markers of pyramidal tract (PT)¹ or extratelencephalic (ET) projecting neurons² (Fig. 2c). We therefore label these neurons “L5 ET 1-4”. Intra-telencephalic (IT) projecting excitatory neurons were identified by expression of *Slc30a3*’.”

6. Many of our concerns above (1,3,5) stem from the following: the authors do not directly discuss (or show data for) the companion MERFISH paper. We effectively reviewed that manuscript as well (on biorxiv; <https://www.biorxiv.org/content/10.1101/2020.06.04.105700v1>) and found high concordance between these two works. We think this strongly adds to the value and validity of the present work and think that several of our concerns can be addressed simply by direct reference and/or generation of figures based on some of the MERFISH data. We recognize that this becomes a delicate balancing act, but this seems like a far simpler and higher-resolution solution than independently replicating findings that have, for all intents and purposes, been demonstrated elsewhere.

We thank the Reviewer for this suggestion. We agree that referencing the MERFISH paper by Zhang et al. is appropriate, and helps to address many questions about the spatial distribution of MOp cells that are not answered by our dissection-based strategy alone. As explained above, we have added this reference to the paper.

1. Tasic, B. *et al.* Shared and distinct transcriptomic cell types across neocortical areas. *Nature* **563**, 72–78 (2018).
2. Hodge, R. D. *et al.* Conserved cell types with divergent features in human versus mouse cortex. *Nature* **573**, 61–68 (2019).

Reviewer Reports on the Second Revision:

Referees' comments:

Referee #3 (Remarks to the Author):

The authors present a statistically rigorous, exceptionally large cross-modality dataset profiling motor cortex diversity. The scale and usefulness of the manuscript as a resource is unquestionable to those studying the motor cortex. The authors now present several novel biological insights and a stronger framework for interpreting the results.

In their response to the last round of comments, the authors addressed the vast majority of our concerns. However, we still request several minor changes. The lack of a co-in situ that simultaneously measures the expression of Rspo1 and Fezf2 remains a confusing omission from the manuscript. Any argument about the L4/L5 identities of neurons in a 'historically agranular' cortex should be supported by mutual overlap/exclusion. This is a minor request that we made in the first round of reviews, which has still not been confirmed by in situ. If the authors insist upon not performing this experiment, then any speculation on Rspo1 should be either removed or vastly decreased (and moved to discussion). On the other hand, other conclusions are now appropriately supported by data from the companion BICCN paper, which is the equivalent of a co-in situ between these markers and is therefore valid to draw conclusions from. Other than this, we find the responses to be appropriate and complete.

Author Rebuttals to Second Revision:

Response to Reviewer #3

Referees' comments:

Referee #3 (Remarks to the Author):

The authors present a statistically rigorous, exceptionally large cross-modality dataset profiling motorcortex diversity. The scale and usefulness of the manuscript as a resource is unquestionable to thosestudying the motor cortex. The authors now present several novel biological insights and a stronger framework for interpreting the results.

We thank the Reviewer for their appreciative comments.

In their response to the last round of comments, the authors addressed the vast majority of our concerns. However, we still request several minor changes. The lack of a co-in situ that simultaneously measures the expression of Rspo1 and Fezf2 remains a confusing omission from the manuscript. Any argument about the L4/L5 identities of neurons in a 'historically agranular' cortex should be supported by mutual overlap/exclusion. This is a minor request that we made in the first round of reviews, which has still not been confirmed by in situ. If the authors insist upon not performing this experiment, then any speculation onRspo1 should be either removed or vastly decreased (and moved to discussion). On the other hand, other conclusions are now appropriately supported by data from the companion BICCN paper, which is the equivalent of a co-in situ between these markers and is therefore valid to draw conclusions from. Other than this, we find the responses to be appropriate and complete.

We agree with the Reviewer that our statements about a molecular cell type with

signatures of Layer 4 pyramidal cells based on single cell sequencing should be validated by independent, complementary data. The most direct evidence, as suggested by the reviewer, comes from simultaneous imaging of the markers *Rspo1* (marking L4 neurons) and *Fezf2* (a marker of L5). We have now addressed this using data from the BICCN companion paper Zhang et al. (bioRxiv 2020.06.04.105700), which uses spatial transcriptomics (MERFISH) to resolve the spatial distribution of these transcripts in mouse MOp. The Extended Data Figure 10 (shown below) from that paper clearly confirms the existence of a population, localized in the middle cortical layer, expressing *Rspo1* but not *Fezf2*. These L4 neurons are located above (superficial to) the *Fezf2* expressing L5 neurons. We have now added a reference to these data to our paper:

Moreover, the localization of cells with these gene markers in middle layers is further supported by spatial transcriptomics²⁰.

[Figure Readacted]

Zhang et al., 2020 (<https://doi.org/10.1101/2020.06.04.105700>)